# Generalizability of Neural Networks Minimizing Empirical Risk Based on Expressive Ability

**Lijia Yu**[1], **Yibo Miao**[2, 3], **Yifan Zhu**[2, 3], **Xiao-Shan Gao**[2, 3*], **Lijun Zhang**[1, 3, 4]

[1] Key Laboratory of System Software of Chinese Academy of Sciences
   Institute of Software, Chinese Academy of Sciences
[2] State Key Laboratory of Mathematical Sciences
   Academy of Mathematics and Systems Science, Chinese Academy of Sciences
[3] University of Chinese Academy of Sciences
[4] Institute of AI for Industries, Chinese Academy of Sciences

## ABSTRACT

The primary objective of learning methods is generalization. Classic uniform generalization bounds, which rely on VC-dimension or Rademacher complexity, fail to explain the significant attribute that over-parameterized models in deep learning exhibit nice generalizability. On the other hand, algorithm-dependent generalization bounds, like stability bounds, often rely on strict assumptions. To establish generalizability under less stringent assumptions, this paper investigates the generalizability of neural networks that minimize or approximately minimize empirical risk. We establish a lower bound for population accuracy based on the expressiveness of these networks, which indicates that with an adequate large number of training samples and network sizes, these networks, including over-parameterized ones, can generalize effectively. Additionally, we provide a necessary condition for generalization, demonstrating that, for certain data distributions, the quantity of training data required to ensure generalization exceeds the network size needed to represent the corresponding data distribution. Finally, we provide theoretical insights into several phenomena in deep learning, including robust generalization, importance of over-parameterization, and effect of loss function on generalization.

## 1 INTRODUCTION

Understanding the mechanisms behind the nice generalization ability of deep neural networks remains a fundamental challenge problem in deep learning theory. By generalization, it means that neural networks trained on finite dataset give high predict accuracy on the whole data distribution. The generalization bound serves as a critical theoretical framework for evaluating the generalizability of learning algorithms. Let $\mathcal{F}$ be a neural network, $\mathcal{D}$ the data distribution, and $L(\mathcal{F}(x), y) = \mathbb{I}(\widehat{\mathcal{F}}(x) = y)$ where $\widehat{\mathcal{F}}(x)$ is the classification result of $\mathcal{F}(x)$. For a hypothesis space $\mathbf{H}$ and any $\mathcal{F} \in \mathbf{H}$, with probability $1 - \delta$ of dataset $\mathcal{D}_{tr}$ sampled i.i.d. from $\mathcal{D}$, we have the classic uniform generalization bound (Mohri et al., 2018)

$$|\mathbb{E}_{(x,y)\sim\mathcal{D}}[L(\mathcal{F}(x), y)] - \mathbb{E}_{(x,y)\in\mathcal{D}_{tr}}[L(\mathcal{F}(x), y)]| < \sqrt{(8d\ln\frac{2eN}{d} + 8\ln\frac{4}{\delta})/N} \qquad (1)$$

where $d$ is the VC-dimension of $\mathbf{H}$ and $N = |\mathcal{D}_{tr}|$. There exist similar generalization bounds using Radermacher Complexity (Mohri et al., 2018).

In practice, the generalizability of the networks trained by SGD is desirable. For that purpose, algorithmic-dependent generalization bounds are derived. It is shown that if the data satisfy the NTK condition, two-layer networks have a small population risk after training (Jacot et al., 2018; Ji & Telgarsky, 2020). Stability generalization bounds are also obtained by assuming the convexity and Lipschitz properties of the total loss (Hardt et al., 2016; Kuzborskij & Lampert, 2018).

---

*Corresponding author

Unfortunately, uniform generalization bounds fail to explain the important phenomenon that over-parameterized models exhibit nice generalizability (Belkin et al., 2019), as pointed out by Nagarajan & Kolter (2019). For example, the VC-dimension is equal to the product of the number of parameters and the depth for ReLU networks (Bartlett et al., 2021), which renders the bound in equation 1 useless for over-parameterized models. Most of the algorithmic-dependent generalization bounds make strong and unrealistic assumptions. For example, the NTK condition is used to reduce the training to a convex optimization (Ji & Telgarsky, 2020) and the strong smoothness and convexity of the empirical loss are used to measure the effect in each training epoch (Hardt et al., 2016).

In order to give generalization conditions under more relaxed assumptions, we will study the generalization of networks that minimize or approximately minimize the empirical risk, that is, the networks $\mathcal{F} \in \mathbf{M} = \arg \min_{\mathcal{G} \in \mathbf{H}} \sum_{(x,y) \in \mathcal{D}_{tr}} L(\mathcal{G}(x), y)$. The approach is reasonable because most practical training will lead to a very small empirical risk.

In this paper, we consider two-layer networks, like many previous theoretical works (Ba et al., 2020; Ji & Telgarsky, 2020; Zeng & Lam, 2022). From the perspective of expressive ability, we obtain a new type of sample complexity bound: when the number of training data and the size of the network are independently large enough, the network has generalizability. The sample complexity bound depends only on the cost required for the network to express the data distribution, as shown below.

**Theorem 1.1** (Informal, Corollary 4.5). *Let data distribution $\mathcal{D}$ satisfy the condition that a two-layer network with width $W_0$ can reach accuracy 1 over $\mathcal{D}$. Then with high probability of $\mathcal{D}_{tr} \sim \mathcal{D}^N$, if $N \geq \Omega(W_0^2)$ and width$(\mathcal{F}) \geq \Omega(W_0)$ for $\mathcal{F} \in \mathbf{M}$, then $\mathcal{F}$ has high population accuracy.*

From this result, we can determine the amount of training data and the size of the network that can ensure generalizability. Because the requirements for $N$ and width$(\mathcal{F})$ are independent, our bounds can be used to explain the nice generalizability of over-parameterized models (Belkin et al., 2019). The above result is extended to networks that approximately minimize the empirical risk.

We also give a lower bound for the sample complexity. For some data distribution, to ensure the generalizability of network which minimizes the empirical risk, the required number of data must be greater than the size of neural networks required to express such a distribution, as shown below.

**Theorem 1.2** (Informal, Section 5). *For some data distribution $\mathcal{D}$, if the width required for a two-layer network with* ReLU *activation function to express $\mathcal{D}$ is at least $W_0$, then with high probability for a dataset with fewer than $O(W_0)$ elements, any network that minimizes the empirical risk over $\mathcal{D}$ has poor generalization.*

Finally, while deep neural networks exhibit good generalization, numerous classical experimental results indicate that these networks encounter problems such as robustness and generalization. We provide some interpretability for these problems based on our theoretical results. Let $\mathcal{D}_{tr}$ be a dataset and $\mathcal{F} \in \mathbf{M}$. Then, three phenomena of deep learning are discussed with our theoretical results.

**Robustness Generalization.** (Section 6.1) It is known that robust memorization for a dataset $\mathcal{D}_{tr}$ is more difficult than memorization for $\mathcal{D}_{tr}$ (Park et al., 2021; Li et al., 2022; Yu et al., 2024a). We further show that when robust memorization of $\mathcal{D}_{tr}$ is much more difficult than memorization of $\mathcal{D}_{tr}$, then the robustness accuracy of $\mathcal{F}$ over $\mathcal{D}$ has an upper bound which may be low, or $\mathcal{F}$ has no robustness generalization over $\mathcal{D}$.

**Importance of over-parameterization.** (Section 6.2) It is recognized that over-parameterized networks have nice generalizaility (Belkin et al., 2019; Bartlett et al., 2021). In this regard, we show that when the network is large enough, a small empirical loss leads to high test accuracy. In contrast, when the network $\mathcal{F}$ is not large enough, there exist networks that achieve good generalization but cannot be found by minimizing the empirical risk.

**Loss function.** (Section 6.3) We show that for some loss function, generalization may not be achieved. If the loss function has reached its minimum value or is a strictly decreasing concave function, then the network may have poor generalization.

## 2 RELATED WORK

**Generalization bound.** Generalization bound is the central issue of learning theory and has been studied extensively (Valle-Pérez & A. Louis, 2022). The algorithm-independent generalization

bounds usually depend on the VC-dimension or the Rademacher complexity (Mohri et al., 2018). In (Harvey et al., 2017; Bartlett et al., 2019; Yang et al., 2023), the VC-dimension has been accurately calculated in terms of width, depth, and number of parameters. In (Wei & Ma, 2019; Arora et al., 2018; Li et al., 2018), tighter generalization bounds were given based on Radermacher complexity. Generalization bounds were also studied for netwoks with special structures: Long & Sedghi (2019); Ledent et al. (2021); Li et al. (2018) gave the generalization bound of CNN, Vardi et al. (2022) gave the sample complexity of small networks, Brutzkus & Globerson (2021) studied the generalization bound of maxpooling networks, Trauger & Tewari (2024); Li et al. (2023) gave the generalization bound of transformers, and Ma et al. (2018); Luo & Yang (2020); Ba et al. (2020) studied two-layer networks. Under some assumptions for the networks, Neyshabur et al. (2017); Barron & Klusowski (2018); Dziugaite & Roy (2017); Bartlett et al. (2017); Valle-Pérez & A. Louis (2022) gave the upper bounds of the generalization error. Generalization bounds based on information theory and Bayesian theory were also given (Alquier, 2024; Hellström et al., 2023; Tolstikhin & Seldin, 2013). Bayesian generalization bounds do not use VC-dimension or Radermacher complexity, but they hold only for most of the networks. Unfortunately, Nagarajan & Kolter (2019) show that the uniform generalization bound cannot explain the generalizability for deep learning.

Algorithm-dependent generalization bounds were established in the algorithmic stability setting (Bousquet & Elisseeff, 2002; Elisseeff et al., 2005; Shalev-Shwartz et al., 2010). Under some assumptions, Hardt et al. (2016); Wang & Ma (2022); Kuzborskij & Lampert (2018); Lei (2023); Bassily et al. (2020) gave the stability bounds under SGD. For small networks, Ji & Telgarsky (2020); Taheri & Thrampoulidis (2024); Li et al. (2020) proved the generalization of networks under some assumptions. Farnia & Ozdaglar (2021); Xing et al. (2021); Xiao et al. (2022); Wang et al. (2025); Allen-Zhu & Li (2022) gave stability generalization bounds for adversarial training under SGD. Regatti et al. (2019); Sun et al. (2023) gave stability generalization bounds under asynchronous SGD. However, these algorithmic-dependent generalization bounds always impose strong assumptions on the training process or dataset. Generalization bounds for memorization networks were given in Yu et al. (2024b). However, minimizing empirical risk for cross-entropy loss does not necessarily lead to memorization, so our assumption is weaker than memorization.

**Interpretability for Deep Learning.** Interpretability is dedicated to providing reasonable explanations for phenomena that occur in deep learning. In (Zhang et al., 2021) it was pointed out that interpretability is not always needed, but it is important for some prediction systems that are required to be highly reliable. For adversarial samples, it was shown that for certain data distributions and networks, there must be a trade-off between accuracy and adversarial accuracy (Shafahi et al., 2019; Bastounis et al., 2021). In (Yu et al., 2023), it was proven that a small perturbation of the network parameters will lead to low robustness. In (Allen-Zhu & Li, 2022), it was shown that the generation of adversarial samples after training is due to dense mixtures in the hidden weights. In (Yu et al., 2024a; Li et al., 2022), it was shown that ensuring generalization requires more parameters. For overfitting, long-term training has been shown to lead to a decrease in generalization (Xiao et al., 2022; Xing et al., 2021). In (Roelofs et al., 2019), comprehensive analysis of overfitting was given. In (Belkin et al., 2019; Bartlett et al., 2021), the importance of over-parameterized interpolation networks is mentioned, and in Arora et al. (2019); Cao & Gu (2019); Ji & Telgarsky (2020), the training and generalization of DNNs in the over-parameterized regime were studied. In this paper, we explain these phenomena from the perspective of the expressive ability of networks.

## 3 NOTATION

In this paper, for any $A \in \mathbb{R}$, $O(A)$ means a real number no more than $cA$ for some $c > 0$ and $\Omega(A)$ means a real number not less than $cA$ for some $c > 0$. By saying that for all $(x, y) \sim \mathcal{D}$ there is an event $A$, we mention $\mathbb{P}_{(x,y)\sim\mathcal{D}}(A) = 1$.

### 3.1 NEURAL NETWORK

In this paper, we consider two-layer neural networks $\mathcal{F} : \mathbb{R}^n \to \mathbb{R}$ that can be written as:

$$F(x) = \sum_{i=1}^{W} a_i \sigma(W_i x + b_i) + c,$$

where $\sigma$ is the activation function, $W_i \in \mathbb{R}^{1 \times n}$ is the transition matrix, $b_i \in \mathbb{R}$ is the bias part, $W$ is the width of the network, and $a_i, c \in \mathbb{R}$. Denote $\mathbf{H}_W^\sigma(n)$ as the set of all two-layer neural networks

with input dimension $n$, width $W$, activation function $\sigma$, and all parameters are in $[-1,1]$. To simplify the notation, we denote $\mathbf{H}_W^{\text{ReLU}}(n)$ by $\mathbf{H}_W(n)$ when using the ReLU activation function.

## 3.2 DATA DISTRIBUTION

In this paper, we consider binary classification problems. To avoid extreme cases, we focus primarily on the data distribution defined below.

**Definition 3.1.** For $n \in \mathbb{Z}_+$, $\mathcal{D}(n)$ is the set of distributions $\mathcal{D}$ over $[0,1]^n \times \{-1,1\}$ that have a *positive separation bound:* $\inf_{(x_1,y_1),(x_2,y_2)\sim\mathcal{D} \text{ and } y_1\neq y_2} ||x_1 - x_2||_2 > 0$.

*Remark* 3.2. Equivalently, a distribution that has positive separation bound means that there exists a $c > 0$, such that $||x_1 - x_2||_2 > c$ for all $(x_1,y_1),(x_2,y_2) \sim \mathcal{D}$ where $y_1 \neq y_2$.

The accuracy of a network $\mathcal{F}$ on a distribution $\mathcal{D}$ is defined as $A_\mathcal{D}(\mathcal{F}) = \mathbb{P}_{(x,y)\sim\mathcal{D}}(\text{Sgn}(\mathcal{F}(x)) = y)$, where Sgn is the sign function. We use $\mathcal{D}_{\text{tr}} \sim \mathcal{D}^N$ to mean that $\mathcal{D}_{\text{tr}}$ is a dataset of $N$ samples drawn i.i.d. according to $\mathcal{D}$.

## 3.3 MINIMUM EMPIRICAL RISK

Consider the loss function $L(\mathcal{F}(x),y) = \ln(1 + e^{-\mathcal{F}(x)y})$, which is the cross-entropy loss for binary classification problems. For a dataset $\mathcal{D}_{tr} \subset [0,1]^n \times \{-1,1\}$ and a hypothesis space $\mathbf{H}_W^\sigma(n)$. To learn the features of the data in $D_{tr}$, the general method is empirical risk minimization (ERM), which minimizes empirical risk on the training dataset $\sum_{(x,y)\in\mathcal{D}_{tr}} L(\mathcal{F}(x),y)$ of the network $\mathcal{F}$.

In this paper, we mainly consider networks $\mathcal{F} \in \mathbf{H}_W^\sigma(n)$ that can minimize empirical risk, that is, networks in

$$\mathbf{M}_W^\sigma(\mathcal{D}_{tr}, n) = \arg\min_{\mathcal{G}\in\mathbf{H}_W^\sigma(n)} \sum_{(x,y)\in\mathcal{D}_{tr}} L(\mathcal{G}(x),y). \tag{2}$$

It should be noted that such networks exist in most cases, as shown below.

**Proposition 3.3.** *Let $\mathcal{D}_{tr} \subset [0,1]^n \times \{-1,1\}$ and $\sigma$ be a continuous function. Then for any $W \in \mathbb{Z}^+$, there exists an $\mathcal{F} \in \mathbf{H}_W^\sigma(n)$ such that $\mathcal{F} \in \mathbf{M}_W^\sigma(\mathcal{D}_{tr}, n)$.*

*Proof.* Since $\sigma$ is a continuous function, the empirical risk $\sum_{(x,y)\in\mathcal{D}_{tr}} L(\mathcal{F}(x),y) = \sum_{(x,y)\in\mathcal{D}_{tr}} \ln e^{-y(\sum_{i=1}^W a_i\sigma(W_ix+b_i)+c)}$ is a continuous function of the network parameters $a_i, b_i, c$ and $W_i$. The proposition now comes from the fact that continuous functions have reachable upper and lower bounds on a closed domain $[-1,1]^{W_g}$ of the parameters, where $W_g = W(n+2)+1$ is the number of parameters of $\mathcal{F}$.

$\square$

# 4 GENERALIZATION BASED ON NEURAL NETWORK EXPRESSIVE ABILITY

In this section, we demonstrate that, based on the expressive ability of neural networks, the generalizability of the network that minimizes the empirical risk can be established. Specifically, in Section 4.1, we establish the relationship between expressive ability and generalizability. In Section 4.2, we extend our conclusion to networks that approximately minimize empirical risk. In Section 4.3, we compare our generalization bounds with existing generalization bounds, showcasing the superiority of our bound.

## 4.1 A LOWER BOUND FOR ACCURACY BASED ON THE EXPRESSIVE ABILITY

We first define the expressive ability of neural networks to classify the data distribution.

**Definition 4.1.** We say that a distribution $\mathcal{D}$ over $[0,1]^n \times \{-1,1\}$ can be **expressed** by $\mathbf{H}_W^\sigma$ with confidence $c$, if there exists an $\mathcal{F} \in \mathbf{H}_W^\sigma$ such that

$$\mathbb{P}_{(x,y)\sim\mathcal{D}}(y\mathcal{F}(x) \geq c) = 1.$$

*Remark* 4.2. The smallest $W$ that $\mathcal{D}$ can be expressed by $\mathbf{H}_W^\delta$ with confidence $c$ can be considered as a complexity measure of distribution $\mathcal{D}$ under confidence $c$ and activation function $\delta$.

For any distribution $\mathcal{D} \in \mathcal{D}(n)$, we can always find some activation function $\sigma$, such that $\mathcal{D}$ can be expressed by $\mathbf{H}_W^\sigma(n)$ with confidence $c$ for some $W$ and $c$. Therefore, this definition is reasonable. For example, if $\sigma = \text{ReLU}$, according to the universal approximation theorem of neural networks (Cybenko, 1989), any $\mathcal{D} \in \mathcal{D}(n)$ can be represented by a network with $\text{ReLU}$ as activation function, as shown by the following proposition. The proof is given in Appendix A.

**Proposition 4.3.** *For any distribution $\mathcal{D} \in \mathcal{D}(n)$, there exist $W \in \mathbb{N}_+$ and $c > 0$ such that $\mathcal{D}$ can be expressed by $\mathbf{H}_W(n)$ with confidence $c$.*

We have the following relationship between expressive ability and generalization ability. The proof is given in Appendix B.

**Theorem 4.4.** *Let $\sigma$ be a continuous function with Lipschitz constant $L_p$ and $\mathcal{D} \in \mathcal{D}(n)$ be expressed by $\mathbf{H}_{W_0}^\sigma$ with confidence $c$. Then for any $W \geq W_0 + 1$, $N \in \mathbb{N}_+$, $\delta \in (0,1)$, with probability at least $1 - \delta$ of $\mathcal{D}_{tr} \sim \mathcal{D}^N$, the following bound stands for any $\mathcal{F} \in \mathbf{M}_W^\sigma(\mathcal{D}_{tr}, n)$:*

$$A_\mathcal{D}(\mathcal{F}) \geq 1 - O(\frac{W_0}{cW} + \frac{nL_p(W_0+c)\sqrt{\log(4n)}}{c\sqrt{N}} + \sqrt{\frac{\ln(2/\delta)}{N}}). \tag{3}$$

**Proof Idea.** There are two main steps in the proof. The first step tries to estimate the minimum value of the empirical risk, which mainly uses the assumption: $\mathcal{D}$ can be expressed by $\mathbf{H}_{W_0}^\sigma$ with confidence $c$. The minimum value is based on $W_0, c, W$. Then, we use the minimum value to estimate the performance of the network on $\mathcal{D}_{tr}$. In the second step, we can use the result in the first step and the classic generalization bound to estimate the performance of the network across the entire distribution and get the result. The core idea of this step is that the minimum value of empirical risk does not depend on $N$, but the Radermacher complexity becomes smaller when increasing $N$. Then, when $N$ is large enough, the performance of networks in $\mathcal{D}$ and $\mathcal{D}_{tr}$ is similar.

Some experimental results used to verify Theorem 4.4 are included in Appendix L. Formula (3) differs from the classical generalization limits in that both the number of samples $N$ and the size of the network $W$ are in the denominator, and therefore increasing $N$ and $W$ independently leads to better test accuracy. As a consequence, the generalization ability of over-parameterized networks can be explained. Since the values of $N$ and $W$ to ensure generalization are only influenced by the size required for the network to express the data distribution, we can infer the following corollary.

**Corollary 4.5.** *With probability $1 - \delta$ of $\mathcal{D}_{tr} \sim \mathcal{D}^N$, it holds $A_\mathcal{D}(\mathcal{F}) \geq 1 - \epsilon$ for any $\mathcal{F} \in \mathbf{M}_W^\sigma(\mathcal{D}_{tr}, n)$, when $W \geq \Omega(W_0/(c\epsilon))$ and $N \geq \Omega(\frac{L_p(W_0+c)n\sqrt{\log(4n)}}{c\epsilon})^2 + \Omega(\frac{\ln(2/\delta)}{\epsilon^2})$.*

The above condition for generalization is different from traditional sample complexity in that besides the requirements on the number of samples, a new requirement on the size of networks is given independently, which is the reason to explain the generalization ability of over-parameterized networks.

*Remark* 4.6. For networks with depth larger than two, we can show that if the depth and width of the network and the number of data exceed a distribution-dependent threshold, then with high probability, the network minimizing the empirical risk can ensure generalization, as demonstrated in Appendix K. However, due to the complexity of deep networks, accurately determining the required depth, width, and data volume remains a challenge.

### 4.2 GENERALIZATION OF NETWORKS APPROXIMATELY MINIMIZING EMPIRICAL RISK

In practice, it is often challenging to find the networks that accurately minimize empirical risk. In this section, we show that for networks that approximately minimize empirical risk, its generalization can also be guaranteed if the value of the empirical risk is small. We define such a set of networks.

**Definition 4.7.** For any $q \geq 1$ and dataset $\mathcal{D}_{tr}$, we say $\mathcal{F} \in \mathbf{H}_W^\sigma(n)$ is a $q$-approximation of minimizing empirical risk if

$$\sum_{(x,y)\in\mathcal{D}_{tr}} L(\mathcal{F}(x), y) \leq q \min_{f\in\mathbf{H}_W^\sigma(n)} \sum_{(x,y)\in\mathcal{D}_{tr}} L(f(x), y).$$

For all $q$-approximation networks, we have the following result. The proof is given in Appendix C.

**Theorem 4.8.** *Let $\sigma$ be a continuous function with Lipschitz constant $L_p$ and $\mathcal{D} \in \mathcal{D}(n)$ be expressed by $\mathbf{H}^\sigma_{W_0}$ with confidence $c$. Then for any $W \geq W_0 + 1$, $N \in \mathbb{N}_+$, $q \geq 1$ and $\delta \in (0, 1)$, with probability at least $1 - \delta$ of $\mathcal{D}_{tr} \sim \mathcal{D}^N$, we have*

$$A_{\mathcal{D}}(\mathcal{F}) \geq 1 - O\left(\frac{qW_0}{cW} + \frac{nL_p(W_0 + c)\sqrt{\log(4n)}}{c\sqrt{N}} + \sqrt{\frac{\ln(2/\delta)}{N}}\right),$$

*for any $q$-approximation $\mathcal{F} \in \mathbf{H}^\sigma_W(n)$ to minimize the empirical risk.*

The conditions of the above theorem can be achieved much easier than those of Theorem 4.4, because we do not need $\mathcal{F}$ to achieve any local or global minimum point, but only need to have a small empirical risk.

### 4.3 COMPARISON WITH CLASSICAL CONCLUSIONS

In this section, we compare our generalization bounds with previous ones. Compared to algorithm-independent generalization bounds, our bound performs better when the data size is not significantly larger than the network size. Compared to algorithm-dependent generalization bounds, our bound does not require overly strong assumptions as prerequisites.

**Compare with the algorithm-independent generalization bound.** When the scale of the network is bounded, a general generalization bound can be calculated by the VC-dimension.

**Theorem 4.9** (P.217 of (Mohri et al., 2018), Informal). *Let $\mathcal{D}_{tr} \sim \mathcal{D}^N$ be the training set. For the hypothesis space $\mathbf{H} = \{\mathrm{Sgn}(\mathcal{F}(x)) \,\|\, \mathcal{F}(x) : \mathbb{R}^n \to \mathbb{R}\}$ and $\delta \in \mathbb{R}_+$, with probability at least $1 - \delta$, for any $\mathrm{Sgn}(\mathcal{F}(x)) \in \mathbf{H}$, we have*

$$|A_{\mathcal{D}}(\mathrm{Sgn}(\mathcal{F})) - \mathbb{E}_{(x,y) \in \mathcal{D}_{tr}}[I(\mathrm{Sgn}(\mathcal{F}(x)) = y)]| \leq O\left(\sqrt{\frac{\mathrm{VC}(\mathbf{H}) + \ln(1/\delta)}{N}}\right). \tag{4}$$

*When considering the local VC-dimension, we have the following result (Zhivotovskiy & Hanneke, 2018). Under Massart's bounded noise condition, let $\mathcal{F}_{\mathcal{D}_{tr}}$ be the network that has the highest accuracy over $\mathcal{D}_{tr}$. Then with probability $1 - \delta$ of $\mathcal{D}_{tr} \sim \mathcal{D}^N$, it holds*

$$A_{\mathcal{D}}(\mathcal{F}_{\mathcal{D}_{tr}}) \geq 1 - \Omega\left(\frac{\mathrm{VC}(\mathbf{H})\log(N/\mathrm{VC}(\mathbf{H})) + \log(1/\delta)}{N}\right).$$

Theorem 4.9 points out that when the number of data is much more than the VC-dimension of the network hypothesis space, generalization can be ensured. Since the VC-dimension is generally larger than the number of parameters of the network (Bartlett et al., 2019), Theorem 4.9 means that to ensure generalization, the number of data must be greater than the number of parameters of the network, which is contradictory to the fact that over-parameterized models have nice generalizability (Belkin et al., 2019; Bartlett et al., 2021). Similar results hold for the generalization bound based on Radermacher complexity, due to the observation that the Radermacher complexity for deep networks is close to 1 (Zhang et al., 2017). On the other hand, our generalization bounds in Theorem 4.4 can be used to explain the fact that over-parameterized models have nice generalizability.

**Compare with the algorithm-dependent generalization bound.** In the study of algorithm-dependent generalization bound, some works derive generalization bounds based on gradient descent under strong assumptions not met by neural networks.

**Theorem 4.10** (Ji & Telgarsky (2020)). *Let $\epsilon \in (0, 1)$, $\delta \in (0, 1/4)$ and distribution $\mathcal{D}$ over $[0, 1]^n$ satisfy the NTK conditions with constant $\gamma$. Let $\lambda = \frac{\sqrt{2\ln(4n/\delta)} + \ln(4/\epsilon)}{\gamma/4}$ and $M = \frac{4096\lambda^2}{\gamma^6}$. If the two-layer network with width $W > M$ and training step $\eta \leq 1$, with probability $1 - 4\delta$ of $\mathcal{D}_{tr} \sim \mathcal{D}^N$ and training initiation point, after at most $\frac{2\lambda^2}{\eta\epsilon}$ times gradient descent on $\mathcal{D}_{tr}$, it holds for the trained network $\mathcal{F}$*

$$A_{\mathcal{D}}(\mathcal{F}) \geq 1 - 2\epsilon - 16\frac{\sqrt{2\ln(4N/\delta)} + \ln(4/\epsilon)}{\gamma^2\sqrt{N}} - 6\sqrt{\frac{\ln(2/\delta)}{N}}.$$

Theorem 4.10 requires NTK conditions for distribution. These conditions can lead the training approach to convex optimization, which is an overly strong condition. Theorem 4.4 only requires that a network interpolates the positive separation distribution, and it stands for any distribution $\mathcal{D} \in \mathcal{D}(n)$ as mentioned in Proposition 4.3. Stability bounds represent another algorithm-dependent approach to generalization bound, as shown below:

**Theorem 4.11** (Theorem 3.7 in Hardt et al. (2016)). *Assume that for every sample $(x, y)$, $L(\mathcal{F}_\theta(x), y)$ as a function based on $\theta$ is $\beta$-smooth, convex and $L$-Lipschitz. Let $F^*$ be the network obtained by training on dataset $\mathcal{D}_{tr}$ by using SGD $T$ times and each step size $\alpha_t < 2/\beta$. Then we have*

$$\mathbb{E}_{\mathcal{D}_{tr} \sim \mathcal{D}^N, \text{SGD}} |\frac{1}{N} \sum_{(x,y) \in \mathcal{D}_{tr}} L(\mathcal{F}^*(x), y) - E_{(x,y) \sim \mathcal{D}}[L(F^*(x), y)]| \leq \frac{2L^2 \sum_{i=1}^T \alpha_i}{N}.$$

Theorem 4.11 requires convex and smooth conditions for the loss function which are not satisfied by neural networks. Also, the Lipschitz constant is directly related to the network size, which cannot explain the over-parameterization phenomenon. In Theorem 4.4, there is no such problem because the network size $W$ is in the denominator.

**Compare with the PAC-Bayes-KL Bound.** Consider the following Bayesian generalization bound:

**Theorem 4.12** (Tolstikhin & Seldin (2013)). *For any fixed distribution $\pi$ over hypothesis space $\mathbf{H}$ and a distribution $\rho$ over $\mathbf{H}$, with probability at least $1 - \delta$ of $\mathcal{D}_{tr} \sim \mathcal{D}^N$, we have*

$$\mathbb{E}_{\mathcal{F} \sim \rho}[\mathbb{E}_{(x,y) \sim \mathcal{D}}[L(\mathcal{F}(x), y)]] - \mathbb{E}_{\mathcal{F} \sim \rho}[\mathbb{E}_{(x,y) \in \mathcal{D}_{tr}}[L(\mathcal{F}(x), y)]] \leq \frac{\text{KL}(\rho \| \pi) + O(\ln(N/\delta))}{N}$$

*where* KL *is the Kullback-Leibler divergence.*

The result estimates the overall generalization bound of networks in the hypothesis space based on a distribution. In contrast, we attempt to provide estimates for networks minimizing the empirical risk.

# 5 LOWER BOUND FOR SAMPLE COMPLEXITY BASED ON EXPRESSIVE ABILITY

In this section, we consider the lower bound of data complexity necessary for generalization, similar to Section 4, the lower bound of data complexity which we are looking for should only rely on the distribution itself but not rely on the hypothesis space, such as the result in (Wainwright, 2019).

## 5.1 UPPER BOUND FOR ACCURACY WITHOUT ENOUGH DATA

This section illustrates that in the worst-case scenario, the minimum number of data needed to guarantee accuracy is constrained by the VC-dimension of the smallest hypothesis space necessary to represent a distribution. We give a definition first.

**Definition 5.1.** For a hypothesis space $\mathbf{H} \subset \mathbb{R}^n \to \mathbb{R}$, $\text{VC}(\mathbf{H})$ is the maximum number of data in $[0, 1]^n$ that $\mathbf{H}$ can shatter. Precisely, there exist $\text{VC}(\mathbf{H})$ samples $\{x_i\}_{i=1}^{\text{VC}(\mathbf{H})} \subset [0, 1]^n$, such that for any $\{y_i\}_{i=1}^{\text{VC}(\mathbf{H})} \in \{-1, 1\}$, there is an $\mathcal{F} \in \mathbf{H}$ such that $\text{Sgn}(\mathcal{F}(x_i)) = y_i$ for all $i \in [\text{VC}(\mathbf{H})]$. But there do not exist $\text{VC}(\mathbf{H}) + 1$ such samples.

We have the following theorem. The proof is given in Appendix D.

**Theorem 5.2.** *For any $n, W, W_0 \in \mathbb{N}_+$ and activation function $\sigma$, there is a $\mathcal{D} \in \mathcal{D}(n)$ that satisfies the following properties.*

*(1) There is an $\mathcal{F} \in \mathbf{H}_{W_0}^\sigma(n)$ such that $A_{\mathcal{D}}(\mathcal{F}) = 1$.*

*(2) For any given $\epsilon, \delta \in (0, 1)$, if $N \leq \text{VC}(\mathbf{H}_{W_0}^\sigma(n))(1 - 4\epsilon - \delta)$, then with probability $1 - \delta$ of $\mathcal{D}_{tr} \sim \mathcal{D}^N$, we have $A_{\mathcal{D}}(\mathcal{F}) < 1 - \epsilon$ for some $\mathcal{F} \in \mathbf{M}_W^\sigma(\mathcal{D}_{tr}, n)$.*

The theorem indicates that for distributions that require networks with width $W_0$ to express, some of them require at least $\Omega(\text{VC}(\mathbf{H}_{W_0}^\sigma(n)))$ training data to ensure generalization. It is worth mentioning that this conclusion is true for any given $W$ in the theorem. It is easy to see that a larger $W_0$ makes $\text{VC}(\mathbf{H}_{W_0}^\sigma(n))$ larger, so as the cost of expression increases, generalization becomes difficult. However, it is difficult to accurately calculate $\text{VC}(\mathbf{H}_{W_0}^\sigma(n))$ for general $\sigma$. If we focus on $\text{ReLU}$ networks, by the result in (Bartlett et al., 2019), we have

**Corollary 5.3.** *For any given $n, W, W_0 \in \mathbb{N}_+$, there is a $\mathcal{D} \in \mathcal{D}(n)$ that satisfies the following properties.*

*(1) There is an $\mathcal{F} \in \mathbf{H}_{W_0}(n)$ such that $A_{\mathcal{D}}(\mathcal{F}) = 1$.*

*(2) For any given $\epsilon, \delta \in (0, 1)$, if $N \leq O(nW_0(1 - 4\epsilon - \delta))$, then for all $\mathcal{D}_{tr} \sim \mathcal{D}^N$, it holds $A_{\mathcal{D}}(\mathcal{F}) < 1 - \epsilon$ for some $\mathcal{F} \in \mathbf{M}_W^{\sigma}(\mathcal{D}_{tr}, n)$.*

Besides, for any distribution, we can show that if the parameters required to express a distribution tend to infinity, the required number of data to ensure generalization for such a distribution must also tend to infinity. As shown in the following theorem. The proof is given in Appendix E.

**Theorem 5.4.** *Suppose $\mathcal{D} \in \mathcal{D}(n)$, $W_0 \geq 2^{n+1}$, and $A_{\mathcal{D}}(\mathcal{F}) \leq 1 - \epsilon$ for any $\epsilon$ and $\mathcal{F} \in \mathbf{H}_{W_0}(n)$. If $N \leq W_0^{\frac{1}{n+1}}(n + 1)/e$, then for any $\mathcal{D}_{tr} \sim \mathcal{D}^N$ and $W \in \mathbb{N}_+$, there is an $\mathcal{F} \in \mathbf{M}_W(\mathcal{D}_{tr}, n)$ such that $A_{\mathcal{D}}(\mathcal{F}) \leq 1 - \epsilon$.*

However, since Theorem 5.4 is correct for all distributions and datasets, it can only provide a relatively loose bound. If the distribution is given, we can calculate the relationship between the minimum number of data required and the minimum number of parameters required to fit it, as shown in the following section.

## 5.2 APPROPRIATE NETWORK MODEL HELPS WITH GENERALIZATION

As mentioned in the previous sections, expressive ability and generalization ability are closely related. Section 4.1 demonstrates that simpler expressions facilitate generalization; Section 5.1 reveals that, in the worst-case scenario, the amount of data required to guarantee generalization is at least the VC-dimension of the hypothesis space that can express the distribution.

Therefore, for a given distribution, selecting an appropriate network model that can fit the distribution easily may help facilitate better expression with fewer data and network size, ultimately leading to improved generalization. In this section, we illustrate that selecting an appropriate activation function for the neural network according to the target distribution enhances generalization.

To better explain this conclusion, let us examine the following distribution.

**Definition 5.5.** *Let $\mathcal{D}_n$ be a distribution defined over $\{(\frac{i}{n}\mathbf{1}, \mathbb{I}(i))\}_{i=1}^n$, where $\mathbf{1}$ is the vector with all one entries in $\mathbb{R}^n$, $\mathbb{I}(x) = 1$ if $x$ is odd and $\mathbb{I}(x) = -1$ if $x$ is even, and the probability of each point is the same.*

As shown below, ReLU networks need $\Omega(n)$ width to express this distribution and require $\Omega(n)$ data to ensure generalization. The proof is given in Appendix F.

**Proposition 5.6.** *(1) For any $n$, $A_{\mathcal{D}_n}(\mathcal{F}) < 1$ for any $\mathcal{F} \in \mathbf{H}_W(n)$ when $W < n/2$;*

*(2) If $N \leq \delta n$ where $\delta \in (0, 1)$, then for all $\mathcal{D}_{tr} \sim \mathcal{D}_n^N$ and $W \in \mathbb{N}_+$, it holds $A_{\mathcal{D}}(\mathcal{F}) \leq 0.5 + 2\delta$ for some $\mathcal{F} \in \mathbf{M}_W(\mathcal{D}_{tr}, n)$.*

But if we use the activation function $\sigma(x) = \sin(\pi x)$, the networks only need $O(1)$ width to express such a distribution and require fewer data to ensure generalization. The proof is given in Appendix G.

**Proposition 5.7.** *(1) For any $n$, $\mathcal{D}_n$ can be expressed by $\mathbf{H}_1^{\sigma}(n)$ with confidence 1;*

*(2) For any $W \geq 2, n > 2$, $\delta \in (0, 1)$ and $N \geq 4\frac{\ln(\delta/2)}{\ln(0.5+1/n)}$, with probability $1 - \delta$ of $\mathcal{D}_{tr} \sim \mathcal{D}_n^N$, it holds $A_{\mathcal{D}}(\mathcal{F}) = 1$ for all $\mathcal{F} \in \mathbf{M}_W^{\sigma}(\mathcal{D}_{tr}, n)$.*

As shown in the above example, using $\sigma(x) = \sin(\pi x)$ as the activation function, it only requires $O(\ln(\delta))$ samples and $O(1)$ width to ensure generalization, but ReLU networks require at least $\Omega(n)$ samples and width to ensure generalization. This demonstrates the crucial role of selecting the appropriate network models.

*Remark* 5.8. It is worth mentioning that for some very simple distributions like the Bernoulli distribution, the performance of various activation functions is similar, so we cannot provide a general conclusion for any distribution.

# 6 INTERPRETABILITY OF SOME PHENOMENA IN DEEP NEURAL NETWORK

Although networks minimizing empirical risk are good for generalization, many classic experimental results have shown that networks still have problems. In this section, we will provide interpretability for some classic experimental results based on our theoretical results.

## 6.1 WHY DO GENERAL NETWORKS LACK ROBUSTNESS?

Experiments show that deep neural networks can easily lead to low robustness accuracy (Szegedy et al., 2013). In this section, we provide some explanations for this fact.

The *robustness accuracy* of network $\mathcal{F}$ under distribution $\mathcal{D}$ and robust radius $\epsilon$ is defined as

$$\mathrm{Rob}_{\mathcal{D},\epsilon}(\mathcal{F}) = \mathbb{P}_{(x,y)\sim\mathcal{D}}(\mathbb{I}(\widehat{\mathcal{F}}(x') = y), \forall x' \in \mathbb{B}(x,\epsilon) \cap [0,1]^n).$$

The robustness accuracy requires not only correctness on the samples but also correctness within a neighborhood of the sample. We introduce a notation.

**Definition 6.1.** For a dataset $\mathcal{D}_{tr} = \{(x_i, y_i)\}_{i=1}^N$ and an $\epsilon > 0$, define

$$R(\mathcal{D}_{tr}, \epsilon) = \{\mathcal{D}_r \,\|\, \mathcal{D}_r = \mathcal{D}_{tr} \cup \{(x_i + \epsilon_i, y_i)\}_{i=1}^N, \text{ for some } ||\epsilon_i|| \leq \epsilon\}\}.$$

It is easy to see that $R(\mathcal{D}_{tr}, \epsilon)$ contains all the data formed by adding a perturbation with budget $\epsilon$ to $\mathcal{D}_{tr}$. In the above section, we mainly discussed the network expression ability in distribution. On the other hand, there are also some studies on the network expression ability on dataset such as memorization. Moreover, previous studies (Park et al., 2021; Li et al., 2022; Yu et al., 2024a) have shown that robustly memorizing a dataset may be much more difficult than memorizing a dataset. So, for a given hypothesis space $\mathbf{H}$ that can express a normal dataset well, it may not be able to express the dataset after disturbance. In this case, in order to minimize the empirical risk, the network will prioritize simple features that are easy to fit, but will ignore the complex robust features, which leads to low robustness, as shown in the following theorem. The proof is given in Appendix H.

**Theorem 6.2.** *Let $\mathcal{D} \in \mathcal{D}(n)$ and $L_p$ be the Lipschitz constant of activation function $\sigma$. If $N_0, W_0 \in \mathbb{N}_+$ and $\epsilon, \delta, c_0, c_1 > 0$ satisfy that with probability $1 - \delta$ of $\mathcal{D}_{tr} \sim \mathcal{D}^{N_0}$, it holds*

*(1) there exists an $\mathcal{F} \in \mathbf{H}_{W_0}^\sigma(n)$ such that $y\mathcal{F}(x) \geq c_0$ for all $(x, y) \in \mathcal{D}_{tr}$;*

*(2) there exists a $\mathcal{D}_r \in R(\mathcal{D}_{tr}, \epsilon)$, such that $\sum_{(x,y)\in\mathcal{D}_r} \frac{y\mathcal{F}(x)}{|\mathcal{D}_r|} \leq c_1$ for any $\mathcal{F} \in \mathbf{H}_{W_0}^\sigma(n)$.*

*Then, for any $W \geq W_0 + 1$, with probability $1 - O(\delta)$ of $\mathcal{D}_{tr} \sim \mathcal{D}^{N_0}$ and $\mathcal{F} \in \mathbf{M}_W^\sigma(\mathcal{D}_{tr}, n)$, we have $\mathrm{Rob}_{\mathcal{D},\epsilon}(\mathcal{F}) \leq 1 - \Omega(\frac{c_0 - 2c_1}{nL_pW_0} - \frac{c_1}{nL_pW_0}(\frac{W_0}{W} + \frac{1}{W_0}) - \sqrt{\frac{\ln(n/\delta)}{N_0}})$.*

This theorem implies that if the dataset after adding perturbations becomes more difficult to fit, the network may have a low robustness generalization. Please note that $\epsilon$ affects the conclusion implicitly, because $c_1$ is related to $\epsilon$.

*Remark* 6.3. Conditions (1) and (2) required in the theorem are reasonable. It is obvious that as $\epsilon$ increases, $c_1$ will decrease and when $\epsilon$ is large enough, we have $c_0 \gg c_1 \approx 0$. Hence, in some situation, a small $\epsilon$ is also enough to make $c_0 \gg c_1$, such as the example given in the proof of Theorem 4.3 in (Li et al., 2022).

## 6.2 IMPORTANCE OF OVER-PARAMETERIZED NETWORKS

In the above section, we mainly consider $\mathcal{F} \in \mathbf{M}_W(\mathcal{D}_{tr}, n)$. But what we really need is $\mathcal{F} \in \arg\max_{\mathcal{G}\in\mathbf{H}_W(n)} A_\mathcal{D}(\mathcal{G})$. By Theorem 4.4, it is easy to show that when the number of data and the size of the network are large enough, the generalization of $\mathcal{F} \in \mathbf{M}_W(\mathcal{D}_{tr}, n)$ and $\mathcal{F} \in \arg\max_{\mathcal{G}\in\mathbf{H}_W(n)} A_\mathcal{D}(\mathcal{G})$ are close, as shown below. Following Theorem 4.4, we have

**Corollary 6.4.** *For all $\mathcal{F}_1 \in \mathbf{M}_W(\mathcal{D}_{tr}, n)$ and $\mathcal{F}_2 \in \arg\max_{\mathcal{G}\in\mathbf{H}_W(n)} A_\mathcal{D}(\mathcal{G})$, it holds $|A_\mathcal{D}(\mathcal{F}_2) - A_\mathcal{D}(\mathcal{F}_1)| \leq O(\frac{W_0}{cW} + \frac{nL_p(W_0+c)\sqrt{\log(4n)}}{c\sqrt{N}} + \sqrt{\frac{\ln(2/\delta)}{N}})$.*

*Proof.* Since $1 \geq A_{\mathcal{D}}(\mathcal{F}_2) \geq A_{\mathcal{D}}(\mathcal{F}_1)$, we have $|A_{\mathcal{D}}(\mathcal{F}_2) - A_{\mathcal{D}}(\mathcal{F}_1)| \leq 1 - A_{\mathcal{D}}(\mathcal{F}_1)$, and by Theorem 4.4, we obtain the result. □

The above corollary shows that if the size of the network is large enough, the gap will be small. In the following, we point out that for some distribution $\mathcal{D}$, if the size of network is too small, even with enough data, it may lead to a large gap of $A_{\mathcal{D}}(\mathcal{F}_2) - A_{\mathcal{D}}(\mathcal{F}_1)$. This emphasizes the importance of over-parameterization, as shown below. The proof is given in the Appendix I.

**Proposition 6.5.** *For some distribution $\mathcal{D} \in \mathcal{D}(n)$, there is a $W_0 > 0$, such that*

*(1) There exists an $\mathcal{F} \in \mathbf{H}_{W_0}(n)$ such that $A_{\mathcal{D}}(\mathcal{F}) \geq 0.99$.*

*(2) For any $\delta > 0$, if $N \geq \Omega(n^2 \ln(n/\delta))$, with probability $1 - O(\delta)$ of $\mathcal{D}_{tr} \sim \mathcal{D}^N$, we have $A_{\mathcal{D}}(\mathcal{F}) \leq 0.6$ for all $\mathcal{F} \in \mathbf{M}_{W_0}(\mathcal{D}_{tr}, n)$.*

*Remark* 6.6. In Proposition 6.5, 0.99 can be changed to any real number in $(0, 1)$ and 0.6 can be changed to any real number in $(0.5, 1)$, and the result is still correct.

By Corollary 6.4, a large width does not make (2) in Proposition 6.5 true. So, the above conclusion indicates that for some distributions, when the network is not large enough, even if there exist networks with high accuracy, they cannot be found by minimizing the empirical risk. The distribution considered here contains some outliers. In order to fit these outliers, the small network must reduce generalization.

## 6.3 THE IMPACT OF LOSS FUNCTION

In order to ensure generalizability of the network after empirical risk minimization, it is necessary to choose an appropriate loss function because minimizing some types of loss function is not good for generalization. In the previous sections, we mainly discussed the crossentropy loss function. In this section, we point out that not all loss functions can reach conclusions similar to Theorem 4.4.

**Definition 6.7.** We say that the loss function $L_b : \mathbb{R}^2 \to \mathbb{R}$ is bad if (1) or (2) is valid.

(1) There exist $x_{-1}, x_1 \in \mathbb{R}$ such that $L_b(x_{-1}, -1) = \min_{x \in \mathbb{R}} L_b(x, -1)$ and $L_b(x_1, 1) = \min_{x \in \mathbb{R}} L_b(x, 1)$.

(2) $L_b(\mathcal{F}(x), y) = \phi(y\mathcal{F}(x))$, where $\phi$ is a strictly decreasing concave function.

Condition (1) in the definition means that the loss function can reach its minimum value and condition (2) means that the loss function is a concave function. Some commonly used loss functions, such as the MSE loss $L_{\mathrm{MSE}}(\mathcal{F}(x), y) = ||\mathcal{F}(x) - y||_2$, or $L_q(\mathcal{F}(x), y) = -y\mathcal{F}(x)$, are all bad loss functions.

For such bad loss functions, we have

**Theorem 6.8.** *For any $n$ and bad loss function $L_b$, there is a distribution $\mathcal{D} \in \mathcal{D}(n)$ satisfying the following property. For any $N \geq 0$, there is a $W_0 \geq 0$, such that if $W \geq W_0$, then with probability $0.99$ of $\mathcal{D}_{tr} \sim \mathcal{D}^N$, we have $A_{\mathcal{D}}(\mathcal{F}) \leq 0.5$ for some $\mathcal{F} \in \arg\min_{\mathcal{G} \in \mathbf{H}_W(n)} \sum_{(x,y) \in \mathcal{D}_{tr}} L_b(\mathcal{G}(x), y)$.*

This theorem means that to ensure generalizability, it is important to choose the appropriate loss function. The proof is given in the Appendix J.

## 7 CONCLUSION

In this paper, we give a lower bound for the population accuracy of the neural networks that minimize the empirical risk, which implies that as long as there exist enough training data and the network is large enough, generalization can be achieved. The data and network sizes required only depend on the size required for the network to represent the target data distribution. Furthermore, we show that if the scale required for the network to represent a data distribution increases, the amount of data required to achieve generalization on that distribution will also inevitably increase. Finally, the results are used to explain some phenomena in deep learning.

**Limitation and future work.** Although considering 2 layer networks is quite common in theoretical analysis of deep learning, it is still desirable to extend the result to deep neural networks. Preliminary results for deep neural networks are given in Appendix K, which need to be further studied. A more accurate estimate of the cost required to represent a given data distribution is needed.

## ACKNOWLEDGMENTS

This work is supported by CAS Project for Young Scientists in Basic Research, Grant No.YSBR-040, ISCAS New Cultivation Project ISCAS-PYFX-202201, and ISCAS Basic Research ISCAS-JCZD-202302. This work is also supported by NSFC grant No.12288201 and No.92270001, and grant GJ0090202. The authors thank anonymous referees for their valuable comments.

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

## A  PROOF OF PROPOSITION 4.3

A function $\sigma$ is sigmoidal if $\mathrm{limit}_{x \to -\infty} \sigma(x) = 0$ and $\mathrm{limit}_{x \to \infty} \sigma(x) = 1$. Then, we have

**Theorem A.1** (Theorem 1 in Cybenko (1989)). *For any continuous sigmoidal activation function $\sigma$, $\epsilon \in (0, 1)$ and continuous function $f : [0, 1]^n \to \mathbb{R}$, there exist $W \geq 0$ and $F \in \mathbf{H}_W^\sigma(n)$ such that $|f(x) - F(x)| \leq \epsilon$.*

We prove Proposition 4.3 by using the above Theorem.

*Proof.* It is easy to see that $\sigma(x) = \mathrm{ReLU}(x + 1) - \mathrm{ReLU}(x)$ is a continuous sigmoidal activation function.

Denote $Z_W^\sigma(n)$ as the set of all two-layer neural networks with input dimension $n$, width $W$, and activation function $\sigma$. For simplicity, $Z_W(n)$ means $Z_W^{\mathrm{ReLU}}(n)$.

Firstly, it is easy to see that $Z_W^\sigma(n) \subset Z_{2W}(n)$ for any $W \in \mathbb{N}_+$.

Then, because $\mathcal{D}$ has a positive separation distance with a different label, there is a continuous function $f$ such that: $f(x) = 1$ if $x$ has label 1 in distribution $\mathcal{D}$; $f(x) = -1$ if $x$ has label -1 in distribution $\mathcal{D}$.

Finally, by Theorem A.1, there exist a $W$ and a $\mathcal{F} \in Z_W^\sigma(n)$ such that $|\mathcal{F}(x) - f(x)| \leq 0.1$ for all $x \in [0, 1]^n$. Thus, $\mathcal{F} \in Z_W^\sigma(n) \subset Z_{2W}(n)$ and $P_{(x,y) \sim \mathcal{D}}(y\mathcal{F}(x) \geq 0.9) = 1$.

Let the maximum of the absolute value of the parameters of $\mathcal{F}$ be $A$. If $A \leq 1$, then $\mathcal{F}$ is what we want. If $A > 1$, then we write $\mathcal{F} = a\mathrm{ReLU}(Wx + b) + c$, let $\mathcal{F}_A = (a/A)\mathrm{ReLU}((W/A)x + b/A) + c/A^2$, then there are $\mathcal{F}_A = a\mathrm{ReLU}(Wx + b)/A^2 + c/A^2 = \mathcal{F}/A^2$, so $\mathcal{F}_A$ is a network whose parameter is in $[-1, 1]$ and $\mathcal{F}_A = \mathcal{F}/A^2$. Hence, there are $\mathcal{F}_A \in \mathbf{H}_{2W}(n)$ and $P_{(x,y) \sim \mathcal{D}}(y\mathcal{F}(x) \geq 0.9/A^2) = 1$. The proposition is proved. $\square$

## B  PROOF OF THEOREM 4.4

### B.1  PREPARATORY RESULTS

We give some definitions of the hypothesis space.

**Definition B.1.** For a network $\mathcal{F} : \mathbb{R}^n \to \mathbb{R}$ and an $a > 0$, let $\mathcal{F}_{-a,a}(x) = \min\{\max\{-a, \mathcal{F}(x)\}, a\}$, that is, clamp $\mathcal{F}$ in $[-a, a]$. Then for any hypothesis space $\mathbf{H}$, let $\mathbf{H}_{-a,a} = \{\mathcal{F}_{-a,a} \| \mathcal{F} \in H\}$.

We define the Radermacher complexity.

**Definition B.2.** For a hypothesis space $\mathbf{H}$ and dataset $\mathcal{D}$, the Radermacher complexity of $\mathbf{H}$ under dataset $\mathcal{D}$ is:

$$\mathrm{Rad}_{\mathbf{H}}(\mathcal{D}) = \mathbb{E}_{(q_i)_{i=1}^{|\mathcal{D}|}} \left[ \sup_{\mathcal{F} \in \mathbf{H}} \frac{\sum_{x_i \in \mathcal{D}} q_i \mathcal{F}(x_i)}{|\mathcal{D}|} \right]$$

where $q_i$ satisfies that $P(q_i = 1) = P(q_i = -1) = 0.5$ and $q_i$ are i.i.d.

Here are some results about the Radermacher complexity:

**Lemma B.3.** *For any hypothesis space $\mathbf{H}$, let $\mathbf{H}_{+a} = \{\mathcal{F} + a \| \mathcal{F} \in \mathbf{H}\}$, where $a \in \mathbb{R}$. Then for any hypothesis space $\mathbf{H}$, $a \in \mathbb{R}$ and dataset $\mathcal{D}$, there are $\mathrm{Rad}_{\mathbf{H}}(\mathcal{D}) = \mathrm{Rad}_{\mathbf{H}_{+a}}(\mathcal{D})$.*

Let the $L_{1,\infty}$ norm of a matrix $W$ be the maximum value of the $L_1$ norm for each row of the matrix $W$.

**Lemma B.4.** *Let $\mathcal{F}_{n,d,(L_i),(c_i)} : \mathbb{R}^n \to \mathbb{R}$ be a network with $d$ hidden layers, $L_i$ Lipschitz-continuous activation function for $i$-th activation function, and the output layer does not contain an activation function. Let $w_i$ be the $i$-th transition matrix and $b_i$ be the $i$-th bias vector. Let the $L_{1,\infty}$ norm of $w_i$ plus the $L_{1,\infty}$ norm of $b_i$ be not more than $c_i$. Let $\mathbf{H}_{n,d,L_i,c_i} = \{\mathcal{F}_{n,d,L_i,c_i}\}$. Then when $L_i \geq 1$, $c_i \geq 1$, for any $\{x_i\}_{i=1}^N \subset [0, 1]^n$, there are:*

$$\mathrm{Rad}_{H_{n,d,L_i,c_i}}(\{x_i\}_{i=1}^N) \leq \frac{\Pi_{i=1}^d L_i \Pi_{i=1}^{d+1} c_i}{\sqrt{N}} \left( \sqrt{(d+3)\log(4)} + \sqrt{2\log(2n)} \right).$$

The above lemma is an obvious corollary of Theorem 1 in (Wen et al., 2021). By the above two lemmas we can calculate the Radermacher complexity of $\mathbf{H}_W^\sigma(n)_{-a,a}$.

**Lemma B.5.** *Let $\sigma$ be a $L_p$ Lipschitz-continuous activation function and $L_p \geq 1$, and let $\mathbf{H} = \{F(x,y) : F(x,y) = y\mathcal{F}(x), \mathcal{F}(x) \in \mathbf{H}_W^\sigma(n)_{-a,a}\}$ where $a > 0$ is given in Definition B.1. Then for any $S = \{(x_i, y_i)\}_{i=1}^N \subset [0,1]^n \times \{-1, 1\}$, there are*

$$\mathrm{Rad}_{\mathbf{H}}(S) \leq \frac{2L_p(n+1)(W+1+a)}{\sqrt{N}}(\sqrt{5\log(4)} + \sqrt{2\log(2n)}).$$

*Proof.* First, we have

$$\mathrm{Rad}_{\mathbf{H}}(S) = \mathrm{Rad}_{\mathbf{H}}(\{(x_i, y_i)\}_{i=1}^N) = \mathbb{E}_{(q_i)_{i=1}^N}\Big[\sup_{f \in \mathbf{H}_W^\sigma(n)_{-a,a}} \frac{\sum_{i=1}^N q_i y_i f(x_i)}{|\mathcal{D}|}\Big].$$

Taking into account the definition of $q_i$ in definition B.2, there are $\mathrm{Rad}_{\mathbf{H}}(\{(x_i, y_i)\}_{i=1}^N) = \mathbb{E}_{(q_i)_{i=1}^N}[\sup_{f \in \mathbf{H}_W^\sigma(n)_{-a,a}} \frac{\sum_{i=1}^N q_i f(x_i)}{|D|}] = \mathrm{Rad}_{\mathbf{H}_W^\sigma(n)_{-a,a}}(\{x_i\}_{i=1}^N)$.

So, we just need to calculate $\mathrm{Rad}_{\mathbf{H}_W^\sigma(n)_{-a,a}}(\{x_i\}_{i=1}^N)$.

First, for any function $f$ and $a > 0$, $k \in N^+$, we have

$$
\begin{aligned}
& f_{-a,a}(x) \\
= & \mathrm{ReLU}(f(x) + a) - \mathrm{ReLU}(f(x) - a) - a \\
= & \sum_{i=1}^k (\mathrm{ReLU}(f(x)/k + a/k) - \mathrm{ReLU}(f(x)/k - a/k)) - a
\end{aligned}
$$

On the other hand, let $H_{+a} = \{f + a \| f \in \mathbf{H}_W^\sigma(n)_{-a,a}\}$. Then for any $F \in \mathbf{H}_{+a}$, there are $F = f_{-a,a}(x) + a$ for some $f \in \mathbf{H}_W^\sigma(n)$. Then by the above form of expression, take $k = [W/2]$, $F$ and write it as a network with:

(1): Depth 3. Because $f$ has depth 2, after adding a $\mathrm{ReLU}$ activation function, it was depth 3.

(2): The first layer has an $L_p$ Lipschitz-continuous activation function; the second layer has a 1 Lipschitz-continuous activation function, that is, $\mathrm{ReLU}$.

(3): The $L_{1,\infty}$ norm of the three transition matrices plus bias vectors are $n + 1$, $\frac{W+1+a}{[W/2]}$ and $2[W/2]$, as shown in the below.

The first transition matrices: this layer is the same as the first layer of $f$. Consider the bound of values of parameters of $f$ is not more than 1, and the first transition matrix of has $n$ weights in each row, and with the bias added, there are a total of $n + 1$ weights, so its norm is $n + 1$.

The second transition matrices: Let the second transition matrices of $f$ be $W_f$ and bias be $c$. Then the second transition matrices of $\mathcal{F}$ is $W_f/k$, bias is $c/k + a/k$ or $c/k - a/k$. Using the bound of value of parameters of $f$, the width of $W_f$ is $W + 1$, and the value of $k$, so we get the result.

The third transition matrix: It is $(1, 1, 1, \ldots, 1, 1, -1, -1, -1, \ldots, -1, -1)$, where there are $k$ number of 1 and $k$ number of -1 in it, and we get the result.

So, by Lemmas B.4 and B.3, there are $\mathrm{Rad}_{\mathbf{H}_{+a}}(\{x_i\}_{i=1}^N) = \mathrm{Rad}_{\mathbf{H}_W^\sigma(n)_{-a,a}}(\{x_i\}_{i=1}^N) = \frac{2L_p(n+1)(W+1+a)}{\sqrt{N}}(\sqrt{5\log(4)} + \sqrt{2\log(2n)})$. The theorem is proved. $\qquad \square$

Another important Theorem is required.

**Theorem B.6** (Theorem in Mohri et al. (2018)). *Let $H = \{F : \mathbb{R}^n \to [-a, a]\}$, and $\mathcal{D}$ be a distribution, then with probability $1 - \delta$ of $\mathcal{D}_{tr} \sim \mathcal{D}^N$, there are:*

$$|\mathbb{E}_{x \sim \mathcal{D}}[F(x)] - \sum_{x \in \mathcal{D}_{tr}} \frac{F(x)}{N}| \leq 2\mathrm{Rad}_H(\mathcal{D}_{tr}) + 6a\sqrt{\frac{\ln(2/\delta)}{2N}},$$

*for any $F \in H$.*

We give a simple lemma below.

**Lemma B.7.** *(1): When $0 < x \le e$, there are $\ln(1+x) \ge x/(e+1)$.*

*(2): When $x > 0$, there are $xe^{-x} \le 1/e$.*

*Proof.* For (1): Consider $f(x) = \ln(1+x) - x/(e+1)$, there are $f'(x) = 1/(1+x) - 1/(1+e) \ge 0$, so $f(x) \ge f(0) = 0$, which means that $\ln(1+x) - x/(e+1) \ge 0$.

For (2): Consider $f(x) = xe^{-x}$, there are $f'(x) = e^{-x}(1-x)$, it is easy to see that $f'(x)$ become positive then negative when $x$ from 0 to $\infty$, and $f'(1) = 0$, so $f(x) \le f(1) = 1/e$. $\qquad\square$

### B.2 PROOF OF THEOREM 4.4

*Proof.* Let $\mathcal{D}_{tr} \sim \mathcal{D}^N$ and $\mathcal{F}$ be a network in $\mathbf{M}_W^\sigma(\mathcal{D}_{tr}, n)$. We prove Theorem 4.4 in four parts:

**Part one:** We have $\sum_{(x,y)\in\mathcal{D}_{tr}} L(\mathcal{F}(x), y) \le N\ln(1 + e^{-c[\frac{W}{W_0+1}]})$.

Because $\mathcal{D}$ can be expressed by $\mathbf{H}_{W_0}^\sigma(n)$ with confidence $c$, so there is a network $\mathcal{F}_0 = \sum_{i=1}^{W_0} a_i\sigma(W_i x + b_i) + c_1$ such that $y\mathcal{F}(x) \ge c$ for all $(x,y) \sim \mathcal{D}$. Moreover, we can write such network as $\mathcal{F}_0 = \sum_{i=1}^{W_0+1} a_i\sigma(W_i x + b_i)$, where $a_{W_0+1} = \mathrm{Sgn}(c_1)$, $W_{W_0+1} = 0$, $b_{W_0+1} = |c_1|$.

Now, we consider the following network in $\mathbf{H}_W^\sigma(n)$:

$$\mathcal{F}_W = \sum_{i=1}^{(W_0+1)[\frac{W}{W_0+1}]} a_{i\%(W_0+1)}\sigma(W_{i\%(W_0+1)}x + b_{i\%(W_0+1)}),$$

Here, we stipulate that $i\%(W_0+1) = W_0 + 1$ when $W_0+1|i$. Then we have $\mathcal{F}_W(x) = [\frac{W}{W_0+1}]\mathcal{F}_0(x)$ and $\mathcal{F}_W(x) \in \mathbf{H}_W^\sigma(n)$. Moreover, there are $y\mathcal{F}_W(x) = y[\frac{W}{W_0+1}]\mathcal{F}_0(x) \ge [\frac{W}{W_0+1}]c$ for all $(x,y) \sim \mathcal{D}$, so $\sum_{(x,y)\in\mathcal{D}_{tr}} L(\mathcal{F}_W(x), y) \le N\ln(1 + e^{-c[\frac{W}{W_0+1}]})$. So for any $\mathcal{F} \in \arg\min_{f\in\mathbf{H}_W^\sigma(n)} \sum_{(x,y)\in\mathcal{D}_{tr}} L(f(x), y)$, there are $\sum_{(x,y)\in\mathcal{D}_{tr}} L(\mathcal{F}(x), y) \le \sum_{(x,y)\in\mathcal{D}_{tr}} L(\mathcal{F}_W(x), y) \le N\ln(1 + e^{-c[\frac{W}{W_0+1}]})$.

**Part Two:** Let $k = [\frac{W}{W_0+1}]$, by the assumption in Theorem, there is $k \ge 1$. We will show that $|\{(x,y) : (x,y) \in \mathcal{D}_{tr}, y\mathcal{F}(x) \le kc/2\}| \le Ne^{-kc/2+2}$.

Let $S = \{(x,y) : (x,y) \in \mathcal{D}_{tr}, y\mathcal{F}(x) \le kc/2\}$, then according to part one, there are: $|S|\ln(1 + e^{-kc/2}) \le \sum_{(x,y)\in S} L(\mathcal{F}(x), y) \le \sum_{(x,y)\in\mathcal{D}_{tr}} L(\mathcal{F}(x), y) \le N\ln 1 + e^{-kc} \le Ne^{-kc}$. So, there are $|S|\ln 1 + e^{-kc/2} \le Ne^{-kc}$.

By Lemma B.7, there are $|S|e^{-kc/2}/(e+1) \le |S|\ln 1 + e^{-kc/2} \le Ne^{-kc}$, so $|S| \le Ne^{-kc/2}(e+1) < Ne^{-kc/2+2}$.

**Part Three:** By Definition B.1, let network $g = \mathcal{F}_{-kc/2,kc/2}$, we show that, with high probability, $\mathbb{E}_{(x,y)\sim\mathcal{D}}yg(x)$ has a lower bound.

Firstly, by part two, there are $\sum_{(x,y)\in\mathcal{D}_{tr}} yg(x) \ge N(kc(1 - e^{-kc/2+2})/2 - kce^{-kc/2+2}/2) = Nkc(1 - 2e^{-kc/2+2})/2$.

Then, let $H = \{y\mathcal{F}(x) : \mathcal{F}(x) \in \mathbf{H}_W^\sigma(n)_{-kc/2,kc/2}\}$, by Lemma B.5, there are $Rad_H(\mathcal{D}_{tr}) \le \frac{2(n+1)(W+1+kc/2)L_p}{\sqrt{N}}(\sqrt{5\log(4)} + \sqrt{2\log(2n)})$, $Rad_H(\mathcal{D}_{tr})$ is defined in definition B.2.

So, considering that $yg(x) \in H$ and by Theorem B.6, with probability $1 - \delta$ of $\mathcal{D}_{tr}$, there are

$$\mathbb{E}_{(x,y)\sim\mathcal{D}}yg(x)$$
$$\ge \frac{1}{N}\sum_{(x,y)\in\mathcal{D}_{tr}} yg(x) - 2\mathrm{Rad}([\mathbf{H}_W^\sigma(n)]_{-kc/2,kc}) - 3kc\sqrt{\frac{\ln(2/\delta)}{2N}}$$
$$\ge kc(1 - 2e^{-kc/2+2})/2 - \frac{2(n+1)L_p(W+1+kc/2)}{\sqrt{N}}(\sqrt{5\log(4)} + \sqrt{2\log(2n)}) - 3kc\sqrt{\frac{\ln(2/\delta)}{2N}}.$$

**Part Four:** Now, we prove Theorem 4.4.

Firstly, there are $A_{\mathcal{D}}(g) = \mathbb{P}_{(x,y)\sim\mathcal{D}}(yg(x) > 0) \geq \mathbb{E}_{(x,y)\sim\mathcal{D}}[yg(x)]/(kc/2)$, we use $|g(x)| \leq kc/2$ in here. So, by part three, with probability of $\mathcal{D}_{tr}$, there are

$$A_{\mathcal{D}}(g) \geq 1 - 2e^{-kc/2+2} - \frac{4(n+1)L_p(W+1+kc/2)}{\sqrt{N}kc}(\sqrt{5\log(4)} + \sqrt{2\log(2n)}) - 6\sqrt{\frac{\ln(2/\delta)}{2N}}.$$

By Lemma B.7 and $k = [W/(W_0+1)] \geq \frac{W}{2W_0}$ which is because $[W/(W_0+1)] = k \geq 1$ and $W_0 \geq 2$, there are $2e^{-kc/2+2} \leq \frac{4e}{kc} = \frac{4e}{c[\frac{W}{W_0+1}]} \leq \frac{8eW_0}{Wc}$; and it is easy to see that $\frac{4(n+1)L_p(W+1+kc/2)}{\sqrt{N}kc} \leq \frac{4(n+1)WL_p(2+kc/2W)}{\sqrt{N}kc} \leq \frac{8nWL_p(2+c/2W_0)}{\sqrt{N}[W/(W_0+1)]c} \leq \frac{8nL_p(4W_0+c)}{\sqrt{N}c}$, the last inequality uses $[W/(W_0+1)] \geq \frac{W}{2W_0}$.

The last step uses $k = [W/(W_0+1)] \geq \frac{W}{2W_0}$. And $\sqrt{5\log(4)} + \sqrt{2\log(2n)} \leq (\sqrt{5}+\sqrt{2})\sqrt{\log(4n)}$. So there are:

$$A_{\mathcal{D}}(g) \geq 1 - \frac{8eW_0}{Wc} - \frac{8nL_p(1+4\frac{W_0}{c})}{\sqrt{N}}(\sqrt{5}+\sqrt{2})\sqrt{\log(4n)} - 6\sqrt{\frac{\ln(2/\delta)}{2N}}.$$

Lastly, because $A_{\mathcal{D}}(g) = A_{\mathcal{D}}(\mathcal{F})$, we have $A_{\mathcal{D}}(\mathcal{F}) \geq 1 - O(\frac{W_0}{Wc} + \frac{nL_p(W_0+c)\sqrt{\log(4n)}}{\sqrt{N}c} + \sqrt{\frac{\ln(2/\delta)}{N}})$.

The theorem is proved. $\qquad\square$

## C    PROOF OF THEOREM 4.8

The proof is similar to the proof of Theorem 4.4, so we just follow the proof of Theorem 4.4.

*Proof.* Let $\mathcal{D}_{tr} \sim \mathcal{D}^N$, $\mathcal{F}$ be a network in $\mathbf{M}_W^\sigma(\mathcal{D}_{tr}, n)$, and $\mathcal{F}_q$ be a network that is a $q$ approximation of the empirical risk minimization.

We prove Theorem 4.8 in four parts below.

**Part one:** It holds $\sum_{(x,y)\in\mathcal{D}_{tr}} L(\mathcal{F}(x), y) \leq N\ln(1 + e^{-c[\frac{W}{W_0+1}]})$. This is the same as in Part one in the proof of Theorem 4.4

**Part Two:** Let $k = [\frac{W}{W_0+1}] \geq 1$. Then, $|\{(x,y) : (x,y) \in \mathcal{D}_{tr}, y\mathcal{F}_q(x) \leq kc/2\}| \leq qNe^{-kc/2+2}$.

Let $S = \{(x,y) : (x,y) \in \mathcal{D}_{tr}, y\mathcal{F}_q(x) \leq kc/2\}$, then according to part one, there are: $|S|\ln(1 + e^{-kc/2}) \leq \sum_{(x,y)\in S} L(\mathcal{F}_q(x), y) \leq q\sum_{(x,y)\in\mathcal{D}_{tr}} L(\mathcal{F}(x), y) \leq qN\ln 1 + e^{-kc} \leq qNe^{-kc}$. So, there are $|S|\ln 1 + e^{-kc/2} \leq qNe^{-kc}$.

By Lemma B.7, there are $|S|e^{-kc/2}/(e+1) \leq |S|\ln 1 + e^{-kc/2} \leq qNe^{-kc}$, so $|S| \leq qNe^{-kc/2}(e+1) < qNe^{-kc/2+2}$.

**Part Three:** By Definition B.1, let network $g = (\mathcal{F}_q)_{-kc/2,kc/2}$. We will show that, with high probability, $\mathbb{E}_{(x,y)\sim\mathcal{D}}yg(x)$ has a lower bound.

Firstly, by part two, we have $\sum_{(x,y)\in\mathcal{D}_{tr}} yg(x) \geq N(kc(1 - qe^{-kc/2+2})/2 - qkce^{-kc/2+2}/2) = Nkc(1 - 2qe^{-kc/2+2})/2$.

So, with probability $1 - \delta$ of $\mathcal{D}_{tr}$, it holds that

$$\mathbb{E}_{(x,y)\sim\mathcal{D}}yg(x)$$
$$\geq \frac{1}{N}\sum_{(x,y)\in\mathcal{D}_{tr}} yg(x) - 2\text{Rad}([\mathbf{H}_W^\sigma(n)]_{-kc/2,kc}) - 3kc\sqrt{\frac{\ln(2/\delta)}{2N}}$$
$$\geq kc(1 - 2qe^{-kc/2+2})/2 - \frac{2(n+1)L_p(W+1+kc/2)}{\sqrt{N}}(\sqrt{5\log(4)} + \sqrt{2\log(2n)}) - 3kc\sqrt{\frac{\ln(2/\delta)}{2N}}.$$

**Part Four:** Now, we prove Proposition 4.8.

Firstly, there are $A_{\mathcal{D}}(g) = \mathbb{P}_{(x,y)\sim\mathcal{D}}(yg(x) > 0) \geq \mathbb{E}_{(x,y)\sim\mathcal{D}}[yg(x)]/(kc/2)$. So, by part three, with $1 - \delta$ probability of $\mathcal{D}_{tr}$, there are

$$A_{\mathcal{D}}(g) \geq 1 - 2qe^{-kc/2+2} - \frac{4(n+1)L_p(W + 1 + kc/2)}{\sqrt{N}kc}(\sqrt{5\log(4)} + \sqrt{2\log(2n)}) - 6\sqrt{\frac{\ln(2/\delta)}{2N}}.$$

Then, similar as part four in proof of Theorem 4.4, there are

$$A_{\mathcal{D}}(g) \geq 1 - \frac{8qeW_0}{Wc} - \frac{8nL_p(1 + 4\frac{W_0}{c})}{\sqrt{N}}(\sqrt{5} + \sqrt{2})\sqrt{\log(4n)} - 6\sqrt{\frac{\ln(2/\delta)}{2N}},$$

which is what we want. $\qquad\square$

## D    PROOF OF THEOREM 5.2

*Proof.* Assume that Theorem 5.2 is wrong, then there exist $n$, $W$ and $W_0$ such that for given $\epsilon, \delta \in (0, 1)$, if $\mathcal{D} \in \mathcal{D}(n)$ and $N \geq \mathrm{VC}(\mathbf{H}^{\sigma}_{W_0}(n))(1 - 4\epsilon - \delta)$, with probability $1 - \delta$ of $\mathcal{D}_{tr}$, we have $A_{\mathcal{D}}(\mathcal{F}) \geq 1 - \epsilon$ for all $\mathcal{F} \in \arg\min_{f \in \mathbf{H}_W(n)} \sum_{(x,y)\in\mathcal{D}_{tr}} L(f(x), y)$.

We will derive contradictions on the basis of this conclusion.

**Part 1: Find some points and values.**

For a simple expression, let $k = \mathrm{VC}(\mathbf{H}^{\sigma}_{W_0}(n))$, and $\{u_i\}_{i=1}^k$ be $k$ points that can be shattered by $\mathrm{VC}(\mathbf{H}^{\sigma}_{W_0}(n))$. Let $q = \mathrm{VC}(\mathbf{H}^{\sigma}_{W_0}(n))(1 - 4\epsilon - \delta)$.

Now, we consider the following types of distribution $\mathcal{D}$:

(c1): $\mathcal{D}$ is a distribution in $\mathcal{D}(n)$ and $\mathbb{P}_{(x,y)\sim\mathcal{D}}(x \in \{u_i\}_{i=1}^k) = 1$.

(c2): $\mathbb{P}_{(x,y)\sim\mathcal{D}}(x = u_i) = \mathbb{P}_{(x,y)\sim\mathcal{D}}(x = u_j) = 1/k$ for any $i, j \in [k]$.

Let $S$ be the set that contains all such distributions, and it is easy to see that for any $\mathcal{D} \in S$, it can be expressed by $\mathbf{H}^{\sigma}_{W_0}(n)$.

**Part 2: Some definition.**

Moreover, for $\mathcal{D} \in S$, we define $S(\mathcal{D})$ as the following set:

$Z \in S(\mathcal{D})$ if and only if $Z \in [k]^q$ is a vector satisfying: Define $D(Z)$ as $D(Z) = \{(u_{z_i}, y_{z_i})\}_{i=1}^q$, then $A_{\mathcal{D}}(\mathcal{F}) \geq 1 - \epsilon$ for all $\mathcal{F} \in \arg\min_{f \in \mathbf{H}_W(n)} \sum_{(x,y)\in D_Z} L(f(x), y)$, where $z_i$ is the $i$-th weight of $Z$ and $y_{z_i}$ is the label of $u_{z_i}$ in distribution $\mathcal{D}$.

It is easy to see that if we i.i.d. select $q$ samples in distribution $\mathcal{D}$ to form a dataset $\mathcal{D}_{tr}$, then by $c2$, with probability 1, $\mathcal{D}_{tr}$ only contain the samples $(u_j, y_j)$ where $j \in [k]$.

Now for any $\mathcal{D}_{tr}$ selected from $\mathcal{D}$, we construct a vector in $[k]^q$ as follows: the index of $i$-th selected samples as the $i$-th component of the vector. Then each selection situation corresponds to a vector in $[k]^q$ which is constructed as before. Then by the definition of $S(\mathcal{D})$, we have $A_{\mathcal{D}}(\mathcal{F}) \geq 1 - \epsilon$ for all $\mathcal{F} \in \arg\min_{f \in \mathbf{H}_W(n)} \sum_{(x,y)\in\mathcal{D}_{tr}} L(f(x), y)$ if and only if the corresponding vector of $\mathcal{D}_{tr}$ is in $S(\mathcal{D})$.

By the above result and by the assumption at the beginning of the proof, for any $\mathcal{D} \in S$ we have $\frac{|S(\mathcal{D})|}{q^k} \geq 1 - \delta$.

**Part 3: Prove the theorem.**

Let $S_s$ be a subset of $S$, and $S_s = \{\mathcal{D}_{i_1, i_2, \ldots, i_k}\}_{i_j \in \{-1, 1\}, j \in [k]} \subset S$, where the distribution $\mathcal{D}_{i_1, i_2, \ldots, i_k}$ satisfies the label of $u_j$ is $i_j$, where $j \in [k]$.

We will show that there exists at least one $\mathcal{D} \subset S_s$, such that $|S(\mathcal{D})| < (1 - \delta)q^k$, which is contrary to the inequality $\frac{|S(\mathcal{D})|}{q^k} \geq 1 - \delta$ as shown in the above. To prove that, we only need to prove that $\sum_{\mathcal{D}\in S_s} |S(\mathcal{D})| < (1 - \delta)2^k q^k$, use $|S_s| = 2^k$ here.

To prove that, for any vector $Z \in [k]^q$, we estimate how many $\mathcal{D} \in S_s$ make $Z$ included in $S(\mathcal{D})$.

**Part 3.1, situation of a given vector $Z$ and a given distribution $\mathcal{D}$.**

For a $Z = (z_i)_{i=1}^q$ and $\mathcal{D}$ such that $Z \in S(\mathcal{D})$, let $\text{len}(Z) = \{c \in [k] : \exists i, c = z_i\}$. We consider the distributions in $S_s$ that satisfy the following condition: for $i \in \text{len}(Z)$, the label of $u_i$ is equal to the label of $u_i$ in $\mathcal{D}$. Obviously, we have $2^{k-|\text{len}(Z)|}$ distributions that can satisfy the above condition in $S_s$. Let such distributions make up a set $S_{ss}(\mathcal{D}, Z)$. Now, we estimate how many distributions $\mathcal{D}_s$ in $S_{ss}(\mathcal{D}, Z)$ satisfy $Z \in S(\mathcal{D}_s)$.

It is easy to see that if $\mathcal{D}_s \in S_{ss}(\mathcal{D}, Z)$ such that there are more than $[2k\epsilon]$ of $i \in [k]$, $\mathcal{D}_s$ and $D$ have different labels of $u_i$, then $\min\{A_\mathcal{D}(\mathcal{F}), A_{\mathcal{D}_s}(\mathcal{F})\} < 1 - \epsilon$ for any $\mathcal{F}$. So considering $A_\mathcal{D}(\mathcal{F}) \geq 1 - \epsilon$ for all $\mathcal{F} \in \arg\min_{f \in \mathbf{H}_W(n)} \sum_{(x,y) \in D_Z} L(f(x), y)$, by the above result, such kind of $\mathcal{D}_s$ is at most $\sum_{i=0}^{[2k\epsilon]} C_{k-|\text{len}(Z)|}^i$. So, we have that: There are at most $\sum_{i=0}^{[2k\epsilon]} C_{k-|\text{len}(Z)|}^i$ numbers of distributions $\mathcal{D}_s$ in $S_{ss}(\mathcal{D}, Z)$ satisfy $Z \in S(\mathcal{D}_s)$.

**Part 3.2, for any vector $Z$ and distribution $\mathcal{D}$.**

For any distribution $\mathcal{D} \in S_s$, let $y(\mathcal{D})_i$ be the label of $u_i$ in distribution $\mathcal{D}$.

Firstly, for a given $Z$, we have at most $2^{|len(Z)|}$ different $\mathcal{S}_{ss}(\mathcal{D}, Z)$ for $\mathcal{D} \in \mathcal{D}_S$. Because when $\mathcal{D}_1$ and $\mathcal{D}_2$ satisfy $y(\mathcal{D}_1)_i = y(\mathcal{D}_2)_i$ for any $i \in \text{len}(Z)$, we have $\mathcal{D}_{ss}(\mathcal{D}_1, Z) = \mathcal{D}_{ss}(\mathcal{D}_2, Z)$, and $2^{|\text{len}(Z)|}$ situations of label of $u_i$ where $i \in \text{len}(Z)$, so there exist at most $2^{|\text{len}(Z)|}$ different $\mathcal{S}_{ss}(\mathcal{D}, Z)$.

Then, by part 3.1, for an $\mathcal{S}_{ss}(\mathcal{D}, Z)$, at most $\sum_{i=0}^{[2k\epsilon]} C_{k-|\text{len}(Z)|}^i$ of $\mathcal{D}_s \in S_{ss}(\mathcal{D}, Z)$ satisfies $Z \in S(\mathcal{D}_s)$. So by the above result and consider that $\mathcal{D}_s = \cup_{\mathcal{D} \in \mathcal{D}_s} \mathcal{S}_{ss}(\mathcal{D}, Z)$, at most $2^{|\text{len}(Z)|} \sum_{i=0}^{[2k\epsilon]} C_{k-|\text{len}(Z)|}^i$ number of $\mathcal{D}_s \in S_s$ such that $Z \in S(\mathcal{D}_s)$.

And there exist $q^k$ different $Z$, so $\sum_{\mathcal{D} \in S_s} |S(\mathcal{D})| = \sum_Z \sum_{\mathcal{D} \in S_s} I(Z \in S(\mathcal{D})) \leq \sum_Z 2^{|\text{len}(Z)|} \sum_{i=0}^{[2k\epsilon]} C_{k-|\text{len}(Z)|}^i \leq \sum_Z 2^k(1 - \delta) = q^k 2^k(1 - \delta)$. For the last inequality, we use $\sum_{i=0}^{[2k\epsilon]} C_{k-|\text{len}(Z)|}^i < 2^{k-|\text{len}(Z)|}(1 - \delta)$, which can be shown by $|\text{len}(Z)| \leq q \leq k(1 - 4\epsilon - \delta)$ and Lemma D.1.

This is what we want. We proved the theorem. $\square$

A required lemma is given.

**Lemma D.1.** *If $\epsilon, \delta \in (0, 1)$ and $k, x \in \mathbb{Z}_+$ satisfy that: $x \leq k(1 - 2\epsilon - \delta)$, then $2^x(\sum_{j=0}^{[k\epsilon]} C_{k-x}^j) < 2^k(1 - \delta)$.*

*Proof.* We have

$$2^x(\textstyle\sum_{j=0}^{[k\epsilon]} C_{k-x}^j) \leq 2^x 2^{k-x} \frac{[k\epsilon]}{k-x} \leq 2^k \frac{k\epsilon}{k-x} < 2^k(1 - \delta).$$

The first inequality sign uses $\sum_{j=0}^m C_n^m \leq m2^n/n$ where $m \leq n/2$, and by $x \leq k(1 - 2\epsilon - \delta)$, so $[k\epsilon] \leq (k - x)/2$. The third inequality sign uses the fact $x \leq k(1 - 2\epsilon - \delta)$. $\square$

## E PROOF OF THEOREM 5.4

We give the proof of Theorem 5.4.

*Proof.* Let $\mathcal{D}_{tr} \sim \mathcal{D}^N$ and $\mathcal{D}_{tr} = \{(x_i, y_i)\}_{i=1}^N$. For any given $W$, let $\mathcal{F}$ be a network in $\mathbf{M}_W^\sigma(\mathcal{D}_{tr}, n)$ and $\mathcal{F} = \sum_{i=1}^W a_i \text{ReLU}(W_i x + b_i) + c$.

Then, we consider another network $F_f$ that is constructed in the following way:

(1): For a $v \in \{-1, 1\}^N$, we say $i \in S_v$ if: $\text{ReLU}(W_i x_j + b_i) \geq 0$ for all $j$ such that $v_j = 1$; $\text{ReLU}(W_i x_j + b_i) < 0$ for all $j$ such that $v_j = -1$.

(2): For any $v \in \{-1, 1\}^N$, if $S_v \neq \phi$, let $\mathbb{P}_v = \sum_{i \in S_v} a_i W_i/|S_v|$ and $Q_v = \sum_{i \in S_v} a_i b_i/|S_v|$.

(3): Define $F_f$ as: $F_f(x) = \sum_{v \in \{-1,1\}^N, S_v \neq \phi} \sum_{i=1}^{|S_v|} \text{ReLU}(\mathbb{P}_v x + Q_v) + c$.

Then we have the following result:

(r1): $F_f \in \arg\min_{f \in \mathbf{H}_W^\sigma(n)} \sum_{(x,y) \in \mathcal{D}_{tr}} L(f(x), y)$.

Firstly, it is easy to see that each parameter of $F_f$ is in $[-1, 1]$, because for any $v$, $||\mathbb{P}_v||_\infty = ||\sum_{i \in S_v} \frac{a_i W_i}{|S_v|}||_\infty \leq \sum_{i \in S_v} \frac{||a_i W_i||_\infty}{|S_v|} \leq |S_v| \frac{1}{|S_v|} = 1$, and $||Q_v||_\infty = ||\sum_{i \in S_v} \frac{a_i b_i}{|S_v|}||_\infty \leq \sum_{i \in S_v} \frac{||a_i b_i||_\infty}{|S_v|} \leq |S_v| \frac{1}{|S_v|} = 1$.

Then, $F_f$ has width $W$, because for each $i$, there is only one $v$ such that $i \in S_v$, so $\sum_{v \in \{-1,1\}^N, S_v \neq \phi} \sum_{i=1}^{|S_v|} 1 = W$, which implies that $F_f$ has width $W$.

Finally, there are $\mathcal{F}_f(x_i) = \mathcal{F}(x_i)$ for all $(x_i, y_i) \in \mathcal{D}_{tr}$. We just need to show that for $x_1$, others are similar.

There are $\mathcal{F}(x_1) = \sum_{i=1}^W a_i \text{ReLU}(W_i x_1 + b_i) + c = \sum_{i \in [W], W_i x_1 + b_i \geq 0} a_i(W_i x_1 + b_i) + c$. Hence, letting $V1 = \{v : v \in \{-1,1\}^N, v_1 = 1\}$, then there is $F_f(x_1) = \sum_{v \in \{-1,1\}^N, S_v \neq \phi} \sum_{i=1}^{|S_v|} \text{ReLU}(\mathbb{P}_v x_1 + Q_v) + c = \sum_{v \in V1, S_v \neq \phi} \sum_{i=1}^{|S_v|} (\mathbb{P}_v x_1 + Q_v) + c$.

Consider that $\{i \in [W], W_i x_1 + b_i \geq 0\} = \{i : i \in S_v, v \in V1\}$, so:

$$
\begin{aligned}
&\mathcal{F}(x_1) \\
=& \sum_{i \in [W], W_i x_1 + b_i > 0} a_i(W_i x_1 + b_i) + c \\
=& \sum_{i : i \in S_v, v \in V1} a_i(W_i x_1 + b_i) + c \\
=& \sum_{v \in V1, S_v \neq \phi} \sum_{i \in S_v} a_i(W_i x_1 + b_i) + c \\
=& \sum_{v \in V1, S_v \neq \phi} |S_v|(\mathbb{P}_v x_1 + b_v) + c \\
=& \mathcal{F}_f(x_1).
\end{aligned}
$$

By such three points and considering $\mathcal{F} \in \arg\min_{f \in \mathbf{H}_W^\sigma(n)} \sum_{(x,y) \in \mathcal{D}_{tr}} L(f(x), y)$, so there are $F_f \in \arg\min_{f \in \mathbf{H}_W^\sigma(n)} \sum_{(x,y) \in \mathcal{D}_{tr}} L(f(x), y)$.

(r2): $A_\mathcal{D}(F_f) \leq 1 - \delta$ when $N \leq W_0^{\frac{1}{n+1}}(n+1)/e$, where $W_0$ is defined in Theorem. This is what we want.

Firstly, we show that $|\{v : S_v \neq \phi\}| \leq max\{2^{n+1}, \frac{eN}{n+1}^{n+1}\}$, just by Lemma E.1.

Secondly, consider the network $F_{f1} = \sum_{v \in \{-1,1\}^N, S_v \neq \phi} \text{ReLU}(|S_v|\mathbb{P}_v x_1 + |S_v|Q_v) + c$. By the assumption of $\mathcal{D}$ and $|\{v : S_v \neq \phi\}| \leq max\{2^{n+1}, \frac{eN}{n+1}^{n+1}\}$, then we know that, when $N \leq W_0^{\frac{1}{n+1}}(n+1)/e$, there are $A_\mathcal{D}(F_{f1}) \leq 1 - \delta$.

Moreover, there are $F_{f1}(x) = \sum_{v \in \{-1,1\}^N, S_v \neq \phi} \text{ReLU}(|S_v|\mathbb{P}_v x + |S_v|Q_v) + c = \sum_{v \in \{1,1\}^N, S_v \neq \phi} \sum_{i=1}^{|S_v|} \text{ReLU}(\mathbb{P}_v x + Q_v) + c = F_f(x)$, so $A_\mathcal{D}(\mathcal{F}_f) = A_\mathcal{D}(F_{f1}) \leq 1 - \delta$, this is what we want. $\square$

A required lemma is given:

**Lemma E.1.** *For any $S = \{x_i\}_{i=1}^N \subset \mathbb{R}^n$, let $\Pi(S) = \{(\text{Sgn}(W x_i + b))_{i=1}^n : W \in \mathbb{R}^n, b \in \mathbb{R}\}$. Then $|\Pi(S)| \leq max\{2^{n+1}, \frac{eN}{n+1}^{n+1}\}$.*

*Proof.* It is easy to see that $|\Pi(S)| \leq 2^N$ because $\text{Sgn}(W x_i + b) \in \{-1, 1\}$. So, when $N \leq n + 1$, it is obviously correct.

When $N > n + 1$. Consider that the VC-dim of the linear space is $n + 1$, and $\Pi(S) = \{(\text{Sgn}(W x_i + b))_{i=1}^n : W \in \mathbb{R}^n, b \in \mathbb{R}\}$ is the growth function of linear space under dataset $S$. So by Theorem 1 of (Sauer, 1972), we have $|\Pi(S)| \leq \sum_{i=0}^{n+1} C_N^i$.

Moreover, there are $\sum_{i=0}^{n+1} C_N^i \leq \frac{eN}{n+1}^{n+1}$ as shown in (Sauer, 1972), this is what we want. $\square$

## F  PROOF OF PROPOSITION 5.6

We give the proof of Proposition 5.6.

*Proof.* Firstly, it is easy to show that $\mathcal{D}_n$ cannot be expressed by $\mathbf{H}_W(n)$ when $W < n/2$ by Lemma F.1, so we have proved (1) of Proposition 5.6.

Let $\mathcal{D}_{tr} \sim \mathcal{D}_n^N$ and $N \leq n\delta$, for any given $W$, let $\mathcal{F}$ be a network in $\mathbf{M}_W^\sigma(\mathcal{D}_{tr}, n)$, and $\mathcal{F} = \sum_{i=1}^W a_i \mathrm{ReLU}(W_i x + b_i) + c$.

Now we prove (2) of Proposition 5.6. Let $\mathcal{D}_{tr} = \{(\frac{x_i}{n}\mathbf{1}, \mathbb{I}(x_i))\}_{i=1}^N$ where $x_i \in [n]$ be selected from the distribution, without loss of generality, let $x_i < x_{i+1}$ for any $i \in [N]$.

We will divide $[W]$ into several subsets based on the intersection of the plane $W_j x + b$ and the line $-\infty\mathbf{1} \to \infty\mathbf{1}$, let $[W] = \cup_{i=1}^{2N} s_i$, and:

1. For any $i \in [N-1]$: if $j \in [W]$ such that $\frac{x_i}{n}W_j\mathbf{1} + b_i < 0$ and $\frac{x_{i+1}}{n}W_j\mathbf{1} + b_j \geq 0$, then $j \in s_i$;

2. If $j \in [W]$ such that $\frac{x_i}{n}W_j\mathbf{1} + b_i < 0$ for any $i \in [N]$, then $j \in s_N$;

3. For any $i \in \{N+1, N+2, \ldots, 2N-1\}$: if $j \in [W]$ such that $\frac{x_{i-N}}{n}W_j\mathbf{1} + b_i \geq 0$ and $\frac{x_{i-N+1}}{n}W_j\mathbf{1} + b_j < 0$, then $j \in s_i$;

4. If $j \in [W]$ such that $\frac{x_i}{n}W_j\mathbf{1} + b_i \geq 0$ for any $i \in [N]$, then $j \in s_{2N}$.

Now, by such $2N$ subset, we consider another network $\mathcal{F}_f$ that is defined as:

For any $i \in [2N]$, if $S_i \neq \phi$, define $P_i = \sum_{j \in S_i} a_i W_i / |S_i|$ and $Q_i = \sum_{j \in S_i} a_i b_i / |S_i|$. Then $F_f = \sum_{i \in [2N], S_i \neq \phi} \sum_{j=1}^{|S_i|} \mathrm{ReLU}(P_i x + Q_i) + c = \sum_{i \in [2N], S_i \neq \phi} |S_i|\mathrm{ReLU}(P_i x + Q_i) + c$.

Because there is only one intersection point between a straight line and a plane, each $j \in [W]$ is only in one subset $s_i$. So, $\mathcal{F}_f \in \mathbf{H}_W^\sigma(n)$. Moreover, we show that $\mathcal{F}_f(x) = \mathcal{F}(x)$ for any $(x, y) \in \mathcal{D}_{tr}$, which implies $\mathcal{F}_f \in \arg\min_{f \in \mathbf{H}_W^\sigma(n)} \sum_{(x,y) \in \mathcal{D}_{tr}} L(f(x), y)$.

For any $j \in [N]$, by the definition of $s_i$, we know that $\frac{x_j}{n}W_i\mathbf{1} + b_i \geq 0$ if and only if $i \in \{1, 2, \ldots, j-1\} \cup \{N+j, N+j+1, \ldots, 2N\}$, so:

$$
\begin{aligned}
&\mathcal{F}_f(x_j) \\
=\ &\sum_{i \in [2N], S_i \neq \phi} \sum_{j=1}^{|S_i|} \mathrm{ReLU}(P_i x_j + Q_i) + c \\
=\ &\sum_{i \in \{1,2,\ldots,j-1\} \cup \{N+j, N+j+1, \ldots, 2N\}, S_i \neq \phi} \sum_{j=1}^{|S_i|}(P_i x_j + Q_i) + c \\
=\ &\sum_{k \in \underset{i \in \{1,2,\ldots,j-1, N+j, N+j+1, \ldots, 2N\}}{\cup s_i}} (W_k x_j + b_k) + c \\
=\ &\sum_{k \in [W]} \mathrm{ReLU}(W_k x_j + b_k) + c \\
=\ &\mathcal{F}(x_j)
\end{aligned}
$$

This is what we want. At last, by $\mathcal{F}_f = \sum_{i \in [2N], S_i \neq \phi} |S_i|\mathrm{ReLU}(P_i x + Q_i) + c$ has width at most $2N$ and Lemma F.1, and consider that $N \leq n\delta$, we have that: $A_\mathcal{D}(\mathcal{F}_f) \leq 0.5 + 2\delta$, this is what we want.

$\square$

A required lemma is given below.

**Lemma F.1.** *If $x_1 < x_2 < x_3 < \cdots < x_N$, and $x_i$ has label $y_i = 1$ when $i$ is odd, or $x_i$ has label $y_i = -1$. We consider dataset $S = \{(x_i\mathbf{1}(n), y_i)\}$, where $\mathbf{1}$ is all-one vector in $\mathbb{R}^n$. Then: For any two-layer network width $M$, this network can correctly classify at most to $M + \frac{N}{2}$ samples in $S$.*

*Proof.* Let $\mathcal{F} = \sum_{i=1}^M a_i \mathrm{ReLU}(W_i x + b_i) + c$. Let $W_i x + b_i$ and the line $-\infty\mathbf{1}(n) \to \infty\mathbf{1}(n)$ intersect at one point $P_i\mathbf{1}(n)$. Let $P_i \leq P_j$ if $i \leq j$. Let $P_{M+1} = \infty$.

Then it is easy to see that in the line segment $P_i\mathbf{1}(n) \to P_{i+1}\mathbf{1}(n)$, $\mathcal{F}(x)$ is a linear function. So, there is $P_{i+0.5} \in (P_i, \mathbb{P}_{i+1})$ such that $\mathcal{F}$ maintains the positive and negative polarity unchanged in $P_i\mathbf{1}(n) \to P_{i+0.5}\mathbf{1}(n)$ and $P_{i+0.5}\mathbf{1}(n) \to P_{i+1}\mathbf{1}(n)$.

So if $P_i \leq x_u < x_{u+1} < \cdots < x_{u+k} < P_{i+0.5}$, $F$ gives the same label to $x_u\mathbf{1}(n), x_{u+1}\mathbf{1}(n), \ldots, x_{u+k}\mathbf{1}(n)$, which means that $\mathcal{F}$ can classify at most $[\frac{(k+1)+1}{2}]$ samples in them. Similar to when $P_{i+0.5} \leq x_u < x_{u+1} < \cdots < x_{u+k} < P_{i+1}$.

Let $q_i = |\{j : P_{i/2} \leq x_j < P_{i/2+0.5}\}|$ where $i \in [2M]$. Consider that each sample in $S$ is appeared in a $P_i\mathbf{1}(n) \to P_{i+0.5}\mathbf{1}(n)$ or $P_{i+0.5}\mathbf{1}(n) \to P_{i+1}\mathbf{1}(n)$, so $\sum_{i=1}^{2M} q_i = N$.

So, the whole network can classify at most $\sum_{i=1}^{2M}[\frac{1+q_i}{2}] \leq \sum_{i=1}^{2M}\frac{1+q_i}{2} = M + \frac{N}{2}$. $\qquad\square$

## G   PROOF OF PROPOSITION 5.7

*Proof.* Proof of (1): Let $\mathbf{1}$ be the all one vector, $\sum x = \sum_{i=1}^{n} x_i$ where $x_i$ is the $i$-th weight of $x$. We show that $\mathcal{F} = \sigma(\mathbf{1}x - 0.5) \in \mathbf{H}_1^\sigma(n)$ is what we want. Because if $\sum x$ is odd, then $\sigma(\mathbf{1}x - 0.5) = \sigma(\sum x - 0.5) = sin(\pi(\sum x - 0.5)) = 1$; if $\sum x$ is even, then $\sigma(\mathbf{1}x - 0.5) = \sigma(\sum x - 0.5) = sin(\pi(\sum x - 0.5)) = -1$.

Proof of (2): we will prove it into three parts:

**Part one:** For any $W$ and $\mathcal{D}_{tr} \sim \mathcal{D}_n^N$, let $\mathcal{F} \in \mathbf{M}_W^\sigma(\mathcal{D}_{tr}, n)$ and $\mathcal{F} = \sum_{i=1}^{W} \sigma(W_i x + b_i) + c$. Then there are: for any $(x, y) \in \mathcal{D}_{tr}$, there are $y\sigma(W_i x + b_i) = 1$ for any $i \in [W]$.

If not, without loss of generality, assume that $y\sigma(W_1 x + b_1) < 1$ for some $(x, y) \in \mathcal{D}_{tr}$. According to the proof of (1), there are $W_0$ and $b_0$ such that $y\sigma(W_0 x + b_0) = 1$ for any $(x, y) \in \mathcal{D}_{tr}$. Now we consider the network $\mathcal{F}_c(x) = \sum_{i=2}^{W} \sigma(W_i x + b_i) + \sigma(W_0 x + b_0) + c$, then we have that:

Firstly, it is easy to see that $\mathcal{F}_c \in \mathbf{H}_W^\sigma(n)$.

Secondly, we show that $y\mathcal{F}(x) \leq y\mathcal{F}_c(x)$ for any $(x, y) \in \mathcal{D}_{tr}$ and $y\mathcal{F}(x) < y\mathcal{F}_c(x)$ for some $(x, y) \in \mathcal{D}_{tr}$.

By the definition of $\mathcal{F}$ and $\mathcal{F}_c$, for any $(x, y) \in \mathcal{D}_{tr}$, there are $y\mathcal{F}_c(x) - y\mathcal{F}(x) = y(\sigma(W_0 x + b_0) - \sigma(W_1 x + b_1)) = 1 - y\sigma(W_1 x + b_1) \geq 0$, and by the assumption, there is a $(x, y) \in \mathcal{D}_{tr}$ such that $1 > y\sigma(W_1 x + b_1))$, then $y\mathcal{F}_c(x) - y\mathcal{F}(x) > 0$ for such $(x, y) \in \mathcal{D}_{tr}$, this is what we want.

By the above two results, and considering that $L(\mathcal{F}(x), y)$ is a strictly decreasing function about $y\mathcal{F}(x)$, there are $\sum_{(x,y)\in\mathcal{D}_{tr}} L(\mathcal{F}(x), y) > \sum_{(x,y)\in\mathcal{D}_{tr}} L(\mathcal{F}_c(x), y)$, which is contradictory to $\mathcal{F} \in \arg\min_{f\in\mathbf{H}_W^\sigma(n)} \sum_{(x,y)\in\mathcal{D}_{tr}} L(f(x), y)$. So we prove part one.

**Part Two.** For any $j \in Z$, let $x_j = \frac{j}{n}\mathbf{1}$ and $y_j = \mathbb{I}(j)$, where $\mathbb{I}(x)$ is defined in the definition of distribution $\mathcal{D}_n$. If $i_j \in \mathbb{Z}$ where $j \in [4]$ such that $i_1 - i_2$ and $i_3 - i_4$ are co-prime, then there are: if $W_0 \in [-1, 1]^n$ and $b_0 \in [-1, 1]$ such that $y_{i_j}\sigma(W_0 x_{i_j} + b_0) = 1$ for any $j \in [4]$, then $y_p\sigma(W_0 x_p + b_0) = 1$ for all $p \in Z$.

When there is $y_{i_j}\sigma(W_0 x_{i_j} + b_0) = y_{i_j}sin(\pi(W_0 x_{i_j} + b_0)) = y_{i_j}sin(\pi(<W_0, \mathbf{1}> i_j/n + b_0)) = 1$, consider that $y_{i_j} \in \{-1, 1\}$, then there is $<W_0, \mathbf{1}> i_j/n + b_0 = m_{i_j} - 0.5$ for $m_{i_j} \in Z$, moreover, $m_{i_j}$ and $i_j$ are same parity.

Now consider $(W_0 x_{i_1} + b_0) - (W_0 x_{i_2} + b_0)$ and $(W_0 x_{i_3} + b_0) - (W_0 x_{i_4} + b_0)$, there are $<W_0, \mathbf{1}> (i_1 - i_2)/n = m_{i_1} - m_{i_2}$ and $<W_0, \mathbf{1}> (i_3 - i_4)/n = m_{i_3} - m_{i_4}$. So, there are $\frac{i_1-i_2}{i_3-i_4} = \frac{m_{i_1}-m_{i_2}}{m_{i_3}-m_{i_4}}$.

By $i_1 - i_2$ and $i_3 - i_4$ are co-prime, and $|m_{i_1} - m_{i_2}| = |<W_0, \mathbf{1}> (i_1 - i_2)/n| \leq |i_1 - i_2|$, $|m_{i_3} - m_{i_5}| = |<W_0, \mathbf{1}> (i_3 - i_4)/n| \leq |i_3 - i_4|$, there are $<W_0, \mathbf{1}> /n = 1$ or $<W_0, \mathbf{1}> /n = -1$.

Hence, by $m_{i_j} - i_j = <W_0, \mathbf{1}> i_j/n + b_0 + 0.5 - i_j$ and $<W_0, \mathbf{1}> /n = 1$ or $<W_0, \mathbf{1}> /n = -1$, consider that $m_{i_j}$ and $i_j$ are the same parity, so $b = -0.5$.

So for any $p \in \mathbb{Z}$, there are $y_p\sigma(W_0 x_p + b_0) = y_p sin(\pi(<W_0, \mathbf{1}> p/n + b_0)) = y_p sin(\pi(p - 0.5)) = 1$, this is what we want.

**Part Three,** if $\mathcal{D}_{tr} \sim \mathcal{D}_n^N$ and $N \geq 4\frac{\ln(\delta/2)}{\ln(0.5+1/n)}$, with probability $1 - \delta$, there are four samples $(x_i, y_i)$ where $i \in [4]$ in $\mathcal{D}_{tr}$, such that $x_i = \frac{m_i}{n}\mathbf{1}$, $m_1 - m_2$ and $m_3 - m_4$ are co-prime.

By the definition of $\mathcal{D}_n$, it is equivalent to: repeatable randomly select $N \geq 4\frac{\ln(\delta/2)}{\ln(0.5+1/n)}$ points from $[n]$, with probability $1 - \delta$, there are four samples $m_i$ such that $m_1 - m_2$ and $m_3 - m_4$ are co-prime.

By Lemma G.1, when $N \geq 4\frac{\ln(\delta/2)}{\ln(0.5+1/n)}$, with probability at least $1 - (0.5 + 1/n)^{\frac{\ln(\delta/2)}{\ln(0.5+1/n)}}/(0.5 + 1/n) = 1 - \delta/(1 + 2/n) \geq 1 - \delta$. This is what we want.

**Part Four,** we prove the result.

Let $\mathcal{D}_{tr} \sim \mathcal{D}_n^N$. For any $W$, let $\mathcal{F} \in \arg\min_{f \in \mathbf{H}_W^\sigma(n)} \sum_{(x,y) \in \mathcal{D}_{tr}} L(f(x), y)$ and $\mathcal{F} = \sum_{i=1}^W \sigma(W_i x + b_i) + c$.

Firstly, with probability $1 - \delta$, there are four samples in $\mathcal{D}_{tr}$ satisfying part three. Then, according to part one, there are $y\sigma(W_i x + b_i) = 1$ for such four samples. Finally, in part two, there are $y\sigma(W_i x + b_i) = y\sigma(\sum x) = 1$ for any $(x, y) \sim \mathcal{D}_n$. So, $y\mathcal{F}(x) \geq W - 1 > 0$ for any $(x, y) \sim \mathcal{D}_n$, we prove the result. $\qquad\square$

A required lemma is given.

**Lemma G.1.** *Randomly select $N$ points from $[n]$, where $n \geq 3$ and $N \geq 4$. With probability $1 - (0.5 + 1/n)^{N/4-1}$, there are four samples $m_i$ such that $m_1 - m_2$ and $m_3 - m_4$ are co-prime.*

*Proof.* Firstly, we consider the situation that $N = 4$, let $\{m_i\}_{i=1}^4$ are the selected number. Then we have

$$
\begin{aligned}
& P((m_1 - m_2, m_3 - m_4) = 1) \\
=\ & P(m_1 - m_2 \neq 0, m_3 - m_4 \neq 0) - P((m_1 - m_2, m_3 - m_4) \neq 1, m_1 - m_2 \neq 0, m_3 - m_4 \neq 0) \\
=\ & (1 - 1/n)^2(1 - P((|m_1 - m_2|, |m_3 - m_4|) \neq 1 | m_1 - m_2 \neq 0, m_3 - m_4 \neq 0)) \\
\geq\ & (1 - 1/n)^2(1 - \sum_{q \in Prime} P(q | (|m_1 - m_2|, |m_3 - m_4|) | m_1 - m_2 \neq 0, m_3 - m_4 \neq 0)) \\
\geq\ & (1 - 1/n)^2(1 - \sum_{q \in Prime} \frac{1}{q^2}) \\
\geq\ & 0.5(1 - 1/n)^2 \geq 0.5 - 1/n
\end{aligned}
$$

where $Prime$ is the set of all primes. For the second inequality sign, we use

$$
\begin{aligned}
& P(q | m_1 - m_2 \,|\, m_1 - m_2 \neq 0) \\
=\ & \sum_{i=1}^{n-1} P(q | i, i = |m_1 - m_2| \,|\, m_1 - m_2 \neq 0) \\
=\ & [(n-1)/q] * \frac{1}{n-1} \\
\leq\ & 1/q.
\end{aligned}
$$

Similar for $m_3 - m_4$. For the last inequality sign, we use $P(2) = \sum_{i \in Prime} \frac{1}{i^2} < 0.5$, where $P$ is Riemann function.

So, when we select $N$ samples, it contains $[N/4] > N/4 - 1$ pairs of four independent samples randomly selected. So, with probability $1 - (0.5 + 1/n)^{N/4-1}$, there are four samples $m_i$ such that $m_1 - m_2$ and $m_3 - m_4$ are co-prime. $\qquad\square$

## H    PROOF OF THEOREM 6.2

Now, we prove Theorem 6.2.

*Proof.* we prove the proposition into three parts.

**Part One,** with probability $1 - 2\delta$ of $\mathcal{D}_{tr} \sim D^{N_0}$, there are $\mathbb{E}_{x \sim \mathcal{D}}[y\mathcal{F}(x)] \geq c_0 N_0[\frac{W}{W_0+1}] - 2\frac{L_p(W+1)(n+1)(\sqrt{4\log(4)}+\sqrt{2\log(2n)})}{\sqrt{N_0}} - 6\mathcal{F}_{\max}\sqrt{\frac{\ln(2/\delta)}{2N_0}}$ for all $\mathcal{F} \in \mathbf{M}_W^\sigma(\mathcal{D}_{tr}, n)$, where $\mathcal{F}_{\max} = \max_{x+\delta \in [0,1]^n} |\mathcal{F}(x + \delta)|$.

Firstly, we show that there are $\sum_{(x,y) \in \mathcal{D}_{tr}} y\mathcal{F}(x) \geq N_0[\frac{W}{W_0+1}]c_0$ for all $\mathcal{F} \in \mathbf{M}_W^\sigma(\mathcal{D}_{tr}, n)$ when $\mathcal{D}_{tr} \sim \mathcal{D}^{N_0}$ satisfies the conditions of the proposition.

Because $\mathcal{D}_{tr}$ can be expressed in the network space $\mathbf{H}^{\sigma}_{W_0}(n)$ with confidence $c_0$, there is a network $\mathcal{F}_0 = \sum_{i=1}^{W_0} a_i \sigma(W_i x + b_i) + c$ such that $y\mathcal{F}(x) \geq c_0$ for all $(x, y) \in \mathcal{D}_{tr}$. Moreover, we can write such networks as: $\mathcal{F}_0 = \sum_{i=1}^{W_0+1} a_i \sigma(W_i x + b_i)$, where $a_{W_0+1} = \text{Sgn}(c)$, $W_{W_0+1} = 0$, $b_{W_0+1} = |c|$.

Now, we consider the following network in $\mathbf{H}^{\sigma}_{W}(n)$:

$$\mathcal{F}_W = \sum_{i=1}^{(W_0+1)[\frac{W}{W_0+1}]} a_{i\%(W_0+1)} \sigma(W_{i\%(W_0+1)} x + b_{i\%(W_0+1)}),$$

Here, we stipulate that $i\%(W_0+1) = W_0+1$ when $W_0+1|i$. Then we have $\mathcal{F}_W(x) = [\frac{W}{W_0+1}]\mathcal{F}_0(x)$ and $\mathcal{F}_W(x) \in \mathbf{H}^{\sigma}_W(n)$. Moreover, there are $y\mathcal{F}_W(x) = y[\frac{W}{W_0+1}]\mathcal{F}_0(x) \geq [\frac{W}{W_0+1}]c_0$ for all $(x, y) \in \mathcal{D}_{tr}$, so $\sum_{(x,y) \in \mathcal{D}_{tr}} L(\mathcal{F}_W(x), y) \leq N_0 \ln(1 + e^{-c_0[\frac{W}{W_0+1}]})$.

Then, because $\ln 1 + e^x$ is a convex function, so that:

$$
\begin{aligned}
& N_0 \ln 1 + e^{-\frac{\sum_{(x,y) \in \mathcal{D}_{tr}} y\mathcal{F}(x)}{N}} \\
\leq\ & \sum_{(x,y) \in \mathcal{D}_{tr}} \ln 1 + e^{-y\mathcal{F}(x)} \\
=\ & \sum_{(x,y) \in \mathcal{D}_{tr}} L(\mathcal{F}(x), y) \\
\leq\ & \sum_{(x,y) \in \mathcal{D}_{tr}} L(\mathcal{F}_W(x), y) \\
\leq\ & N_0 \ln(1 + e^{-c_0[\frac{W}{W_0+1}]})
\end{aligned}
$$

So $\sum_{(x,y) \in \mathcal{D}_{tr}} y\mathcal{F}(x) \geq c_0 N[\frac{W}{W_0+1}]$.

Hence, by Lemma B.4 and Theorem B.6, with probability $1 - \delta$ of $\mathcal{D}_{tr}$, there are:

$$|\mathbb{E}_{x \sim \mathcal{D}}[\mathcal{F}(x)] - \sum_{x \in \mathcal{D}_{tr}} \frac{\mathcal{F}(x)}{N_0}| \leq 2\frac{L_p(W+1)(n+1)(\sqrt{4\log(4)} + \sqrt{2\log(2n)})}{\sqrt{N_0}} + 6\mathcal{F}_{\max}\sqrt{\frac{\ln(2/\delta)}{2N_0}},$$

for all $\mathcal{F} \in \mathbf{H}_W(n)$.

Finally, combining the above two results, with probability $1 - 2\delta$, there is $\mathbb{E}_{x \sim \mathcal{D}}[\mathcal{F}(x)] \geq c_0 N_0[\frac{W}{W_0+1}] - 2\frac{L_p(W+1)(n+1)(\sqrt{4\log(4)} + \sqrt{2\log(2n)})}{\sqrt{N_0}} - 6\mathcal{F}_{\max}\sqrt{\frac{\ln(2/\delta)}{2N_0}}$.

**Part Two**, there is an upper bound of $\mathbb{E}_{(x,y) \sim \mathcal{D}}[\min_{||\delta|| \leq \epsilon} y\mathcal{F}(x + \delta)]$, if $\mathcal{D}_{tr}$ satisfies Part One.

For any $\mathcal{F} \in \mathbf{H}_W(n)$, we can write $\mathcal{F} = \sum_{i=0}^{\lceil \frac{W}{W_0} \rceil - 1} \sum_{j=1}^{W_0} \text{ReLU}(W_{iW_0+j} x + b_{iW_0+j}) + c$, which is a representation of the sum of $\lceil \frac{W}{W_0} \rceil$ small networks with width of $W_0$. So by part one and by the assumption in the theorem, with probability $1 - \delta$ of $\mathcal{D}_{tr} \sim \mathcal{D}^N$, there is a $\mathcal{D}_r \in R(\mathcal{D}_{tr}, \epsilon)$ such that $\sum_{(x,y) \in \mathcal{D}_r} y\mathcal{F}_1(x) \leq 2N_0 c_1$ for all $\mathcal{F}_1 \in \mathbf{H}_{W_0}(n)$. Then we have $\sum_{(x,y) \in \mathcal{D}_r} y\mathcal{F}(x) \leq 2N_0 c_1 \lceil \frac{W}{W_0} \rceil$, by the definition of $\mathcal{D}_r$, which implies that $\sum_{(x,y) \in \mathcal{D}_{tr}} \min_{||\delta|| \leq \epsilon} y\mathcal{F}(x+\delta) + y\mathcal{F}(x) \leq 2N_0 c_1 \lceil \frac{W}{W_0} \rceil$.

And then, by McDiarmid inequality, with probability $1 - \delta$ of $\mathcal{D}_{tr} \sim \mathcal{D}^{N_0}$, there are $|\mathbb{E}_{(x,y) \sim \mathcal{D}}[\min_{||\delta|| \leq \epsilon} y\mathcal{F}(x + \delta) + y\mathcal{F}(x)] - \frac{1}{N_0} \sum_{(x,y) \in \mathcal{D}_{tr}} \min_{||\delta|| \leq \epsilon} y\mathcal{F}(x + \delta) + y\mathcal{F}(x)| \leq 2\mathcal{F}_{\max}\sqrt{\frac{\ln 1/\delta}{2N_0}}$. So if there are $\mathbb{E}_{(x,y) \sim \mathcal{D}}[\min_{||\delta|| \leq \epsilon} y\mathcal{F}(x + \delta) + y\mathcal{F}(x)] > 2c_1 \lceil \frac{W}{W_0} \rceil + 2\mathcal{F}_{\max}\sqrt{\frac{\ln 1/\delta}{2N_0}}$, according to McDiarmid inequality, with probability $1 - \delta$ of $\mathcal{D}_{tr} \sim \mathcal{D}^{N_0}$, $\sum_{(x,y) \in \mathcal{D}_{tr}} \min_{||\delta|| \leq \epsilon} y\mathcal{F}(x + \delta) + y\mathcal{F}(x) > 2N_0 c_1 \lceil \frac{W}{W_0} \rceil$ stand, which is a contradiction with the above result.

So there must be $\mathbb{E}_{(x,y) \sim \mathcal{D}}[\min_{||\delta|| \leq \epsilon} y\mathcal{F}(x + \delta) + y\mathcal{F}(x)] \leq 2c_1 \lceil \frac{W}{W_0} \rceil + 2\mathcal{F}_{\max}\sqrt{\frac{\ln 1/\delta}{2N_0}}$. Finally, considering the result in Part one, we have that:

$$
\begin{aligned}
& \mathbb{E}_{(x,y) \sim \mathcal{D}}[\min_{||\delta|| \leq \epsilon} y\mathcal{F}(x + \delta)] \\
\leq\ & 2c_1 \lceil \frac{W}{W_0} \rceil - c_0[\frac{W}{W_0+1}] + 2\frac{L_p(W+1)(n+1)(\sqrt{4\log(4)} + \sqrt{2\log(2n)})}{\sqrt{N_0}} + 8\mathcal{F}_{\max}\sqrt{\frac{\ln(2/\delta)}{2N_0}}
\end{aligned}
$$

**Part Three,** Now we can get the result.

By Lemma H.1 and part two, there are $Rob_{\mathcal{D},\epsilon}(\mathcal{F}) \leq 1 - \frac{c_0\lceil\frac{W}{W_0+1}\rceil - 2c_1\lceil\frac{W}{W_0}\rceil}{\mathcal{F}_{\max}} + 8\sqrt{\frac{\ln 2/\delta}{2N_0}} + 2\frac{L_p(W+1)(n+1)(\sqrt{4\log(4)}+\sqrt{2\log(2n)})}{\sqrt{N_0}\mathcal{F}_{\max}}$, and we consider that each parameter of $\mathcal{F}$ is not greater than 1 and Lipschitz constant of $\sigma$ is not more than $L_p$, so $\mathcal{F}_{\max} = \max_{x+\delta\in[0,1]^n}|\mathcal{F}(x+\delta)| = \max_{x\in[0,1]^n}|\mathcal{F}(x)| \leq L_p W(n+1) + 1$.

Let $T = [\frac{W}{W_0+1}]$, by $L_p, n, W_0 \geq 1$ and $W \geq W_0 + 1$, there are:

$$\frac{c_0\lceil\frac{W}{W_0+1}\rceil - 2c_1\lceil\frac{W}{W_0}\rceil}{L_p W(n+1)+1}$$
$$\geq \frac{c_0 T - 2c_1(\frac{(T+1)(W_0+1)}{W_0}+1)}{L_p(T+1)(W_0+1)(n+1)+1}$$
$$= \frac{c_0 T - 2c_1 T}{L_p(T+1)(W_0+1)(n+1)+1} - \frac{4c_1}{L_p(T+1)(W_0+1)(n+1)+1} - \frac{2c_1}{L_p W_0(W_0+1)(n+1)+1}$$
$$\geq \frac{c_0 - 2c_1}{8L_p W_0 n} - \frac{4c_1}{L_p W_0 n}\left(\frac{1}{W/W_0} + \frac{1}{W_0}\right)$$

and

$$\frac{L_p(W+1)(n+1)(\sqrt{4\log(4)}+\sqrt{2\log(2n)})}{\sqrt{N_0}\mathcal{F}_{\max}}$$
$$\geq \frac{L_p(W+1)(n+1)(\sqrt{4\log(4)}+\sqrt{2\log(2n)})}{2\sqrt{N_0}L_p(W+1)(n+1)}$$
$$= 2\frac{\sqrt{4\log(4)}+\sqrt{2\log(2n)}}{\sqrt{N_0}}$$

So, there are $\text{Rob}_{\mathcal{D},\epsilon}(\mathcal{F}) \leq 1 - \frac{c_0-2c_1}{8L_p W_0 n} + \frac{4c_1}{L_p W_0 n}\left(\frac{1}{W/W_0} + \frac{1}{W_0}\right) + 2\sqrt{\frac{\ln 2/\delta}{2N_0}} + 4\frac{\sqrt{4\log(4)}+\sqrt{2\log(2n)}}{\sqrt{N_0}}$. Merge some items and ignore constants, this is what we want. $\qquad\square$

A required lemma is given below.

**Lemma H.1.** *If $\mathcal{F} : \mathbb{R}^n \to \mathbb{R}$ and the distribution $\mathcal{D} \in [0,1]^n \times \{-1,1\}$ satisfy $\mathbb{E}_{(x,y)\sim\mathcal{D}}[y\mathcal{F}(x)] \leq A$ and $\max_{x\in[0,1]^n}|\mathcal{F}(x)| \leq B$, then $A_{\mathcal{D}}(\mathcal{F}) \leq 1 + \frac{A}{B}$.*

*Proof.* There are $\mathbb{E}_{(x,y)\sim\mathcal{D}}[y\mathcal{F}(x)] \geq -(\max_{x\in[0,1]^n}|\mathcal{F}(x)|)\mathbb{P}_{(x,y)\sim\mathcal{D}}(y \neq \text{Sgn}(\mathcal{F}(x))) = -B(1 - A_{\mathcal{D}}(\mathcal{F}))$, so $A \geq -B + BA_{\mathcal{D}}(\mathcal{F})$, that is, $A_{\mathcal{D}}(\mathcal{F}) \leq 1 - \frac{A}{B}$. $\qquad\square$

# I PROOF OF PROPOSITION 6.5

*Proof.* We take a $c > 0$ such that $\ln(1 + e^{-c}) \geq \ln 2 - \ln 2/800$, $1 - (1/e)^{4c} < 0.1$. Then take an $n$ such that $\ln(1 + e^{-n/2+2c}) < \ln 2/2$. Let $N$ satisfy $(\frac{4(n+1)(\sqrt{5\log(4)}+\sqrt{2log2n})}{\sqrt{98N/200}} + 6(n + 2)\sqrt{\frac{\ln(2/\delta)}{2N}}) < \ln 2/800$.

We consider the following distribution $\mathcal{D}$:

(c1): Let $s_1 = \{(x,1) : x \in [0,1], \sum x = n/2+c, ||x||_{-\infty} \geq 2c/n\}$, $||x||_{-\infty}$ mean the minimum of the weight of $|x|$; $s_2 = \{(x,-1) : x \in [0,1], \sum x = n/2 - c, ||x||_\infty \leq 1 - 2c/n\}$; $s_3 = \{(x,-1) : x \in [0,1], \sum x = n - c\}$;

(c2): $\mathbb{P}_{(x,y)\sim\mathcal{D}}(\sum x = n - c) = 1/100$, and $\mathcal{D}$ is a uniform distribution in $s_3$;

(c3): $\mathbb{P}_{(x,y)\sim\mathcal{D}}(\sum x = n/2 + c) = \mathbb{P}_{(x,y)\sim\mathcal{D}}(\sum x = n/2 - c) = 99/200$, and $\mathcal{D}$ is a uniform distribution in $s_1 \cup s_2$.

Let $W_0 = 1$, then we show this distribution and $W_0$ are what we want.

(1) in Theorem: Let $\mathcal{F}_1 = Relu(\mathbf{1}x) - c/2 \in \mathbf{H}_1(n)$. Then $\mathcal{F}_1(x) > 0$ for all $x$ such that $\sum x = c$, and $\mathcal{F}_1(x) < 0$ for all $x$ such that $\sum x = -c$, so $A_{\mathcal{D}}(\mathcal{F}_1) \geq 0.99$.

(2) in Theorem: We use the following parts to show the (2) in the Theorem.

**Part One.** With probability at least $1 - 3e^{-2N/200^2}$ of $\mathcal{D}_{tr} \sim \mathcal{D}^N$, there are at least $N/200$ points in $\mathcal{D}_{tr} \cap s_3$, and at least $98/200N$ points with label 1 in $\mathcal{D}_{tr}$, at least $98/200N$ points with label -1 in $\mathcal{D}_{tr}$.

Using the Hoeffding inequality and $\mathbb{P}_{(x,y)\sim\mathcal{D}}(\sum x = n - c) = 1/100$, we know that with probability at least $1 - e^{-2N/200^2}$ of $\mathcal{D}_{tr}$, there are at least $N/200$ points in $s_3$. Using also the Hoeffding inequality and $\mathbb{P}_{(x,y)\sim\mathcal{D}}(y = 1) = 99/200$, we know that with probability at least $1 - e^{-2N(99/200 - 98/200)^2}$ of $\mathcal{D}_{tr}$, there are at least $98/200N$ points with label 1 in $\mathcal{D}_{tr}$; similar, with probability at least $1 - e^{-2N(101/200 - 98/200)^2}$ of $\mathcal{D}_{tr}$, there are at least $98/200N$ points with label -1 in $\mathcal{D}_{tr}$. Adding them, we get the result.

**Part Two.** For a $\mathcal{D}_{tr}$ that satisfies Part One, if $\mathcal{F} \in \arg\min_{f \in \mathbf{H}_W(n)} \sum_{(x,y)\in\mathcal{D}_{tr}} L(f(x), y)$, then there is $\sum_{(x,y)\in\mathcal{D}_{tr}} L(\mathcal{F}(x), y) \leq \frac{199\ln 2 + \ln(1 + e^{-n/2 + 2c})}{200} N$.

We just consider the following network $\mathcal{F}_1 \in \mathbf{H}_1(n)$: $\mathcal{F}_1 = -\text{ReLU}(\mathbf{1}x - (n/2 + c))$, then $\sum_{(x,y)\in\mathcal{D}_{tr}} L(\mathcal{F}_1(x), y) = \ln 2|\mathcal{D}_{tr}/s_3| + \ln 1 + e^{-n/2 + 2c}|\mathcal{D}_{tr} \cap s_3| \leq \frac{199\ln 2 + \ln(1 + e^{-n/2 + 2c})}{200}$. Hence, for any $\mathcal{F} \in \arg\min_{f \in \mathbf{H}_W(n)} \sum_{(x,y)\in\mathcal{D}_{tr}} L(f(x), y)$, there must be $\sum_{(x,y)\in\mathcal{D}_{tr}} L(\mathcal{F}(x), y) \leq \sum_{(x,y)\in\mathcal{D}_{tr}} L(\mathcal{F}_1(x), y) \leq \frac{199\ln 2 + \ln(1 + e^{-n/2 + 2c})}{200} N$, which is what we want.

**Part Three.** If $\mathcal{F} \in \mathbf{H}_1(n)$ such that $\mathcal{F}(x) \geq 0$ for all $(x, -1) \in s_3$. Then $\mathbb{E}_{(x,y)\sim\mathcal{D}}[L(\mathcal{F}(x), y)] \geq 99/100 \ln 1 + e^{-c} + 1/100 \ln 2$.

Consider that for any $(x_1, 1) \in s_1$, there must be $(x_1 - 2c\mathbf{1}/n, -1) \in s_2$; on the other hand, if $(x_2, -1) \in s_2$, there must be $(x_2 + 2c\mathbf{1}/n, 1) \in s_1$. So we can match the points in $s_1$ and $s_2$ one by one by adding or subtracting a vector $2c\mathbf{1}/n$.

Moreover, for any $x \in [0, 1]$ and $x \in \mathbf{H}_1(n)$, there are $|\mathcal{F}(x) - \mathcal{F}(x - 2c\mathbf{1}/n)| \leq 2c$, which implies $L(\mathcal{F}(x), 1) + L(\mathcal{F}(x - 2c\mathbf{1}/n), -1) = \ln(1 + e^{-\mathcal{F}(x)}) + \ln(1 + e^{\mathcal{F}(x - 2c\mathbf{1}/n)}) \geq 2\ln 1 + e^{-c}$. So for a $(x_1, 1) \in s_1$ and $(x_2, -1) \in s_2$ where $x_2 = x_1 - 2c\mathbf{1}/n$, there must be $L(\mathcal{F}(x_1), 1) + L(\mathcal{F}(x_2), -1) \geq 2\ln 1 + e^{-c}$.

Hence, by $\mathcal{F}(x) > 0$ for all $(x, -1) \in s_3$, $\mathbb{E}_{(x,y)\sim\mathcal{D}}[L(\mathcal{F}(x), y)] \geq 99/200 \ln(1 + e^{-c}) + \ln 2/100$.

**Part Four.** For any network $\mathcal{F} \in \mathbf{H}_1(n)$ such that $\mathcal{F}(x) < 0$ for a $x \in s_3$, then $A_\mathcal{D}(\mathcal{F}) < 60\%$.

Firstly, we show that if $z_1, z_2, z_3$ are collinear, without loss of generality, assuming $z_2$ is between $z_1$ and $z_3$, then $\mathcal{F}(z_1) \geq \mathcal{F}(z_2) \geq \mathcal{F}(z_3)$ or $\mathcal{F}(z_1) \leq \mathcal{F}(z_2) \leq \mathcal{F}(z_3)$. Consider that $z_1, z_2, z_3$ are collinear, so $z_2 = \lambda z_1 + (1 - \lambda)z_3$ for some $\lambda \in (0, 1)$. So let $f(k) = \text{ReLU}(k(Wz_1 + b) + (1 - k)(Wz_3 + b))$, there are $f(0) = \text{ReLU}(Wz_3 + b)$, $f(1) = \text{ReLU}(Wz_1 + b)$ and $f(\lambda) = \text{ReLU}(\lambda(Wz_1 + b) + (1 - \lambda)(Wz_3 + b)) = \text{ReLU}(Wz_2 + b)$. Consider that $\text{ReLU}(\cdot)$ is a monotonic function, so that $f(k)$ is also an monotonic function about $k \in \mathbb{R}$, so we get the result.

Secondly, for any $(z, -1) \in s_2$, let $x_z$ satisfy: $(x_z, 1) \in s_1$ and $x, x_z, z$ are collinear. Then we have that:

(1): For any $(z, -1) \in s_2$, $\mathcal{F}$ must give the wrong label to $x_z$ or $z$. If not, there are $\mathcal{F}(x) < 0$, $\mathcal{F}(x_z) > 0$ and $\mathcal{F}(z) < 0$. By the above result, it is not possible.

(2): Let $S = \{x_z : (z, 1) \in s_2\} \subset s_1$, then $\mathbb{P}_{(x,y)\sim\mathcal{D}}(x \in S | x \in s_1) \geq (1 - 4c/n)^{n-1}$. Because for any $(z, 1) \in s_2$, $\frac{||x - x_z||_2}{||x - z||_2} = \frac{\sum(x - x_z)}{\sum(x - z)} = \frac{n/2 - 2c}{n/2}$, which is a constant value, where $\sum x$ means the sum of the weights of $x$, so $S$ is a proportional scaling of $s_1$ with the ratio $\frac{n - 4c}{n}$, we get the result.

So, there are: $A_\mathcal{D}(\mathcal{F}) \leq \max\{\mathbb{P}_{(x,y)\sim\mathcal{D}}((x,y) \in s_2), \mathbb{P}_{(x,y)\sim\mathcal{D}}((x,y) \in S)\} + \mathbb{P}_{(x,y)\sim\mathcal{D}}((x,y) \in s_3) + \mathbb{P}_{(x,y)\sim\mathcal{D}}(s_1/S) \leq \frac{101 + 99(1 - (1 - 4c/n)^{n-1})}{200} \leq 101/200 + 99/200 * (1 - (1/e)^{4c}) \leq 0.6$, use the definition of $c$.

**Part Five.** Prove the Theorem.

We show that with probability $1 - 3e^{-2N/200^2} - \delta$ of $\mathcal{D}_{tr}$, for any $\mathcal{F} \in \arg\min_{f \in \mathbf{H}_W(n)} \sum_{(x,y) \in \mathcal{D}_{tr}} L(f(x), y)$, $\mathcal{F}$ must give the correct label to some points in $s_3$. Then by part four, we can get the result.

By part one, with probability at least $1 - 3e^{-2N/200^2}$ of $\mathcal{D}_{tr}$, there are at least $N/200$ points in $\mathcal{D}_{tr} \cap s_3$, and at least $98N/200(98N/200)$ points has label 1(-1). Hence, by Lemma I.1 and Theorem 4.9, we know that, with probability $1 - \delta$ of $\mathcal{D}_{tr}$, there are $|\sum_{(x,y) \in \mathcal{D}_{tr}} L(\mathcal{F}(x), y)/N - \mathbb{E}_{(x,y) \sim \mathcal{D}}[L(\mathcal{F}(x), y)]| \geq \frac{4(n+1)(\sqrt{5 \log(4)} + \sqrt{2 log 2n})}{\sqrt{98N/200}} + 6(n+2)\sqrt{\frac{\ln(2/\delta)}{2N}})$. So, with probability $1 - 3e^{-2N/200^2} - \delta$, $\mathcal{D}_{tr}$ satisfies the above two conditions.

For such a $\mathcal{D}_{tr}$, assume that $\mathcal{F} \in \arg\min_{f \in \mathbf{H}_W(n)} \sum_{(x,y) \in \mathcal{D}_{tr}} L(f(x), y)$, and $\mathcal{F}$ must give the correct label to some points in $s_3$.

If not, by part two, we know that $\sum_{(x,y) \in \mathcal{D}_{tr}} L(\mathcal{F}(x), y) \leq \frac{199 \ln 2 + \ln(1 + e^{-n/2+2c})}{200} N$.

Then, by part three, $\mathbb{E}_{(x,y) \sim \mathcal{D}} L(\mathcal{F}(x), y) \geq 99/100 \ln 1 + e^{-c} + 1/100 \ln 2$. Hence, by the definition of $\mathcal{D}_{tr}$, there are $\sum_{(x,y) \in \mathcal{D}_{tr}} L(\mathcal{F}(x), y) \geq N(99/100 \ln 1 + e^{-c} + 1/100 \ln 2) - N(\frac{4(n+1)(\sqrt{5 \log(4)} + \sqrt{2 log 2n})}{\sqrt{98N/200}} - 6(n+2)\sqrt{\frac{\ln(2/\delta)}{2N}})$.

By the definition of $c, n$ and $N$, there are $\sum_{(x,y) \in \mathcal{D}_{tr}} L(\mathcal{F}(x), y)/N \geq (99/100 \ln 1 + e^{-c} + 1/100 \ln 2) - (\frac{4(n+1)(\sqrt{5 \log(4)} + \sqrt{2 log 2n})}{\sqrt{98N/200}} + 6(n+2)\sqrt{\frac{\ln(2/\delta)}{2N}}) \geq \frac{199.5 \ln 2}{200} > \frac{199 \ln 2 + \ln(1 + e^{-n/2+2c})}{200} \geq \sum_{(x,y) \in \mathcal{D}_{tr}} L(\mathcal{F}(x), y)/N$, which leads to contradiction. And we prove the result. $\square$

A required lemma is given below.

**Lemma I.1.** *For any given $D = \{(x_i, y_i)\}_{i=1}^N$, if there are at least $K$ samples have label 1 in it and there are at least $K$ samples have label -1 in it, then there are:*

$$\mathbb{E}_{\sigma_i}[\max_{\mathcal{F} \in \mathbf{H}_1(n)} \frac{1}{N} \sum_{i=1}^N \sigma_i L(\mathcal{F}(x_i), y_i)] \leq \frac{4(n+1)(\sqrt{5 \log(4)} + \sqrt{2 \log(2n)})}{\sqrt{K}},$$

*where $\sigma_i$ are i.i.d and $P(\sigma_i = 1) = P(\sigma_i = -1) = 0.5$.*

*Proof.* We have

$$\begin{aligned} &\mathbb{E}_{\sigma_i}[\max_{\mathcal{F} \in \mathbf{H}_1(n)} \frac{1}{N} \sum_{i=1}^N \sigma_i L(\mathcal{F}(x_i), y_i)] \\ = \ &\mathbb{E}_{\sigma_i}[\max_{\mathcal{F} \in \mathbf{H}_1(n)} \frac{1}{N} \sum_{i=1}^N \sigma_i \ln 1 + e^{y_i \mathcal{F}(x_i)}] \\ \leq \ &\mathbb{E}_{\sigma_i}[\max_{\mathcal{F} \in \mathbf{H}_1(n)} \frac{1}{|D_1|} \sum_{x \in D_1} \sigma_i \ln 1 + e^{\mathcal{F}(x)}] + \mathbb{E}_{\sigma_i}[\max_{\mathcal{F} \in \mathbf{H}_1(n)} \frac{1}{|D_2|} \sum_{x \in D_2} \sigma_i \ln 1 + e^{\mathcal{F}(x)}] \end{aligned}$$

Hence, see $2 \ln(1 + e^x)$ as an activation of the second layer, and the output layer is $\mathcal{F}_2(x) = x/2$. By Lemma B.4, we have $\mathbb{E}_{\sigma_i}[\max_{\mathcal{F} \in \mathbf{H}_1(n)} \frac{1}{|D_1|} \sum_{x \in D_1} \sigma_i \ln 1 + e^{\mathcal{F}(x)}] \leq \frac{2(n+1)(\sqrt{5 \log(4)} + \sqrt{2 \log(2n)})}{\sqrt{|D_1|}}$. Similar for an other part, so we get the result. $\square$

## J  PROOF OF THEOREM 6.8

At first, we give the proof when the loss function $L_b$ satisfies condition (1) in Definition 6.7.

*Proof.* We first define some symbols.

Let the loss function $L_b$ be a bad loss function that satisfies (1) in Definition 6.7. Let $L_b(z_1, 1) = \min_{x \in \mathbb{R}} L_b(x, 1)$ and $L_b(z_{-1}, -1) = \min_{x \in \mathbb{R}} L_b(x, -1)$, assume $|z_1| + |z_{-1}| = z$. For any given $x \in \mathbb{R}^n$, let $x_t = (x_2, x_3, \ldots, x_n) \in \mathbb{R}^{n-1}$, where $x_i$ is the $i$-the weight of $x$; let $x^t = (0, x_1, x_2, x_3, \ldots, x_n) \in \mathbb{R}^{n+1}$.

Then we prove the Theorem in three parts:

**Part One:** We construct the following distribution $\mathcal{D}_b \in [0,1]^n \times \{-1,1\}$:

(1): $\mathcal{D}_b$ is defined on $\{x : x \in [0,1]^n, 0.6 \leq x_1 \text{ or } x_1 \leq 0.4\} \times \{-1,1\}$, where $x_1$ is the first weight of $x$.

(2): $x$ has label 1 if and only if $x_1 \geq 0.6$, or $x$ has label -1.

(3): The marginal distribution about $x$ of $\mathcal{D}_b$ is an uniform distribution.

**Part Two:** For any $\mathcal{D}_{tr} \sim \mathcal{D}_b^N$, we consider the following network $\mathcal{F}_{\mathcal{D}_{tr}}$.

Let $\mathcal{D}_{tr-t} = \{(x_t, y) \| (x,y) \in \mathcal{D}_{tr}\}$. By Lemma J.2, with probability 0.99, there is a $\mathcal{F}_t$ with width $W$ not greater than $O(zN^5n^2)$ such that: if $(x_t, 1) \in \mathcal{D}_{tr-t}$, there are $\mathcal{F}_t(x_t) = z_1$; if $(x_t, -1) \in \mathcal{D}_{tr-t}$, there are $\mathcal{F}_t(x_t) = z_{-1}$. Let $\mathcal{F}_t(x) = \sum_{i=1}^{W} a_i \mathrm{ReLU}(W_i x + b_i) + c$.

Then, we construct $\mathcal{F}_{\mathcal{D}_{tr}} : \mathbb{R}^n \to \mathbb{R}$ as $\mathcal{F} = \sum_{i=1}^{W} a_i \mathrm{ReLU}(W_i^t x + b_i) + c$.

**Part Three:** We prove the Theorem.

For any $\mathcal{D}_{tr} \sim \mathcal{D}_b^N$, we consider the network $\mathcal{F}_{\mathcal{D}_{tr}}$ mentioned in part two. Firstly, we show that $\mathcal{F}_{\mathcal{D}_{tr}}(x) \in \arg\min_{\mathcal{F} \in \mathcal{H}_W(n)} \sum_{(x,y) \in \mathcal{D}_{tr}} L(\mathcal{F}(x), y)$. Because $\mathcal{F}_{\mathcal{D}_{tr}}(x) = \mathcal{F}_t(x_t) = z_1$ when $(x, 1) \in \mathcal{D}_{tr}$ and $\mathcal{F}_{\mathcal{D}_{tr}}(x) = \mathcal{F}_t(x_t) = z_{-1}$ when $(x, -1) \in \mathcal{D}_{tr}$. So $L(\mathcal{F}_{\mathcal{D}_{tr}}(x), y)$ reaches the minimum value for any $(x, y) \in \mathcal{D}_{tr}$, which implies $\mathcal{F}_{\mathcal{D}_{tr}} \in \arg\min_{\mathcal{F} \in \mathcal{H}_W(n)}$.

Secondly, there are $A_{\mathcal{D}}(\mathcal{F}_{\mathcal{D}_{tr}}(x)) = 0.5$. If $A_{\mathcal{D}}(\mathcal{F}_{\mathcal{D}_{tr}}(x)) > 0.5$, then there must be a pair of $(x_1, 1)$ and $(x_2, -1)$ in distribution $\mathcal{D}_b$ such that $(x_1)_t = (x_2)_t$ and $\mathcal{F}_{\mathcal{D}_{tr}}(x)$ give the correct label to $x_1$ and $x_2$. But it is easy to see that $\mathcal{F}_{\mathcal{D}_{tr}}(x) = \mathcal{F}_t(x_t)$ where $\mathcal{F}_t$ is mentioned in part two, so, $\mathcal{F}_{\mathcal{D}_{tr}}(x_1) = \mathcal{F}_t(x)((x_1)_t) = \mathcal{F}_t(x)((x_2)_t) = \mathcal{F}_{\mathcal{D}_{tr}}(x_2)$, which is in contradiction to $\mathcal{F}_{\mathcal{D}_{tr}}(x)$ gives the correct label to $x_1$ and $x_2$. This is what we want.

$\square$

Some required lemmas are given.

**Lemma J.1.** *For any $v \in \mathbb{R}^n$ and $T \geq 1$, let $u \in \mathbb{R}^n$ be uniformly randomly sampled from the hypersphere $S^{n-1}$. Then we have $\mathbb{P}(|\langle u, v \rangle| < \frac{\|v\|_2}{T} \sqrt{\frac{8}{n\pi}}) < \frac{2}{T}$.*

This is Lemma 13 in (Park et al., 2021).

**Lemma J.2.** *For any $N$ points $\{x_i\}_{i=1}^N$ randomly selected in $[0,1]^n$, and any $N$ given point $\{y_i\}_{i=1}^N$ in $[-a, a]$. With probability 0.99 of $\{x_i\}_{i=1}^N$, there is a network $\mathcal{F}$ with width not more than $O(aN^5n^2)$ and $\mathcal{F}(x_i) = y_i$.*

*Proof.* **Part One:** First, we show that with probability 0.99, there is $\|x_i - x_j\|_2 \geq \frac{0.01}{2N^2\sqrt{n}}$ for all pairs $i, j$.

For any $i, j \in \mathbb{N}$ and $\epsilon > 0$, there are:

$$
\begin{aligned}
& P(\|x_i - x_j\|_2 \geq \epsilon) \\
= \; & P(\sum_{k=1}^n ((x_i)_k - (x_j)_k)^2 \geq \epsilon^2) \\
\geq \; & \Pi_{k=1}^n P(((x_i)_k - (x_j)_k)^2 \geq \epsilon^2/n) \\
\geq \; & \Pi_{k=1}^n (1 - \frac{2\epsilon}{\sqrt{n}}) \\
\geq \; & 1 - 2\epsilon\sqrt{n}
\end{aligned}
$$

So $\mathbb{P}(\|x_i - x_j\|_2 \geq \epsilon, \forall(i,j)) \geq 1 - \sum_{i \neq j} P(\|x_i - x_j\|_2 < \epsilon) \geq 1 - 2\epsilon\sqrt{n}N^2$. Take $\epsilon = \frac{0.01}{2\sqrt{n}N^2}$, we get the result.

**Part Two:** There is a $w \in \mathbb{R}^n$ such that $\|w\|_2 = 1$ and $|w(x_i - x_j)| \geq \frac{0.01}{4N^4n} \sqrt{\frac{8}{\pi}}$

By Lemma J.1, for any pair $i, j$, $\mathbb{P}_u(|u(x_i - x_j)| < \frac{\|x_i - x_j\|_2}{2N^2} \sqrt{\frac{8}{n\pi}}) < \frac{1}{N^2}$. So, $\mathbb{P}_u(|u(x_i - x_j)| \geq \frac{\|x_i - x_j\|_2}{2N^2} \sqrt{\frac{8}{n\pi}}, \forall(i,j)) \geq 1 - \sum_{i \neq j} \mathbb{P}_u(|u(x_i - x_j)| < \frac{\|x_i - x_j\|_2}{2N^2} \sqrt{\frac{8}{n\pi}}) > 1 - 1 = 0$, which

implies that there is a $w$ such that $||w||_2 = 1$ and for any pair $(i, j)$, there are $|w(x_i - x_j)| \geq \frac{||x_i - x_j||_2}{2N^2}\sqrt{\frac{8}{n\pi}} \geq \frac{0.01}{4N^4n}\sqrt{\frac{8}{\pi}}$, use the result of part one.

**Part Three:** Prove the result.

Let $w$ be the vector mentioned in part two, and $wx_i < wx_j$ when $i \neq j$. Let $\delta = \frac{0.01}{4N^4n}\sqrt{\frac{8}{\pi}}$. Now, we consider the following network:

$$\mathcal{F}(x) = \sum_{i=1}^{N} \frac{y_i}{\delta}\left(\text{ReLU}(wx - (wx_i + \delta)) + \text{ReLU}(wx - (wx_i - \delta)) - 2\text{ReLU}(wx - wx_i)\right).$$

Easy to verify $\mathcal{F}(x_i) = y_i$. Consider $|wx_i| \leq n$ and $|\frac{y_i}{\delta}| < 400aN^4n$, so $\mathcal{F} \in \mathbf{H}_{O(aN^5n^2)}(n)$. This is what we want. □

We now give the proof of when the loss function $L_b$ satisfies (2) in definition 6.7.

*Proof.* In this proof, we only consider a very simple distribution $\mathcal{D}$: $\mathbb{P}_{(x,y)\sim\mathcal{D}}((x, y) = (\mathbf{0}, -1)) = \mathbb{P}_{(x,y)\sim\mathcal{D}}((x, y) = (\mathbf{1}, 1)) = 0.5$, where $\mathbf{1}$ is a all one vector.

We show that for any $\mathcal{D}_{tr}$ and $W$, let $\mathcal{F} \in \arg\min_{f\in\mathbf{H}_W(n)} \sum_{(x,y)\in\mathcal{D}_{tr}} L_b(f(x), y)$, there are $A_\mathcal{D}(\mathcal{F}) = 0.5$.

**Part one:** When $\mathcal{D}_{tr}$ contains only $(\mathbf{0}, -1)$, then there must be $\mathcal{F} = \sum_{i=1}^{W} -\text{ReLU}(w_i x + 1) - 1$ for some $x_i$, which implies $\mathcal{F}(\mathbf{1}) < 0$, so $A_\mathcal{D}(\mathcal{F}) = 0.5$.

**Part two:** When $\mathcal{D}_{tr}$ contains only $(\mathbf{1}, 1)$, then there must be $\mathcal{F} = \sum_{i=1}^{W} \text{ReLU}(\mathbf{1}x + 1) + 1$, which implies $\mathcal{F}(\mathbf{0}) > 0$, so $A_\mathcal{D}(\mathcal{F}) = 0.5$.

**Part Three:** When $\mathcal{D}_{tr}$ contains $(\mathbf{1}, 1)$ and $(\mathbf{0}, -1)$, we will show that $\mathcal{F} = \sum_{i=1}^{W} \text{ReLU}(\mathbf{1}x + 1) + 1 \in \arg\min_{f\in\mathbf{H}_W(n)} L_b(f(\mathbf{0}), -1) + L_b(f(\mathbf{1}), 1)$. Consider that $A_\mathcal{D}(\mathcal{F}) = 0.5$ for such $\mathcal{F}$, we can prove the Theorem.

If $\mathcal{F} = \sum_{i=1}^{W} \text{ReLU}(\mathbf{1}x + 1) + 1 \notin \arg\min_{f\in\mathbf{H}_W(n)} L_b(f(\mathbf{0}), -1) + L_b(f(\mathbf{1}), 1)$. Let $\mathcal{F}_0(x) = \sum_{i=1}^{W} a_i\text{ReLU}(W_ix + b_i) + c \in \arg\min_{f\in\mathbf{H}_W(n)} L_b(f(\mathbf{0}), -1) + L_b(f(\mathbf{1}), 1)$. Then, let $\mathcal{F}_0(\mathbf{0}) = b$ and $\mathcal{F}_0(\mathbf{1}) = a$.

By $\phi(a) + \phi(-b) = L_b(\mathcal{F}_0(\mathbf{0}), -1) + L_b(\mathcal{F}_0(\mathbf{1}), 1) < L_b(\mathcal{F}(\mathbf{0}), -1) + L_b(\mathcal{F}(\mathbf{1}), 1) = \phi(W(n+1) + 1) + \phi(-W - 1)$, and $\phi$ is a decreasing concave function, there must be $W(n+1) + 1 - a < -b + W + 1$, which implies $|a - b| > Wn$.

Consider $|a - b| = |\sum_{i=1}^{W} a_i\text{ReLU}(b_i) - \sum_{i=1}^{W} a_i\text{ReLU}(W_i\mathbf{1} + b_i)| \leq |\sum_{i=1}^{W} a_i\mathbf{1}W_i| \leq Wn$. This is a contradiction to $|a - b| > Wn$ which was shown above. So, assumption is wrong, so $\mathcal{F} = \sum_{i=1}^{W} \text{ReLU}(\mathbf{1}x + 1) + 1 \in \arg\min_{f\in\mathbf{H}_W(n)} L_b(f(\mathbf{0}), -1) + L_b(f(\mathbf{1}), 1)$, this is what we want. □

## K  EXTEND THE RESULT TO GENERAL NEURAL NETWORK

For multi-layer neural networks, we can show that if there is enough data and the network is large enough, then generalization can also be ensured for the network which can minimum the empirical risk. Unfortunately, due to the complexity of depth networks, we are unable to provide a good generalization bound of such network.

Denote $\mathbf{H}_{W,D}(n)$ to be the set of all neural networks of layers $D$ with input dimension $n$, width $W$ for each hidden layer, activation function ReLU, and all parameters of the transition matrix are in $[-1, 1]$. Then, there are:

**Theorem K.1.** *For any given $n \in \mathbb{N}_+$, if $\mathcal{D} \in \mathcal{D}(n)$ satisfies: there is a network $\mathcal{F} \in \mathbf{H}_{W_0,D_0}(n)$ such that $\mathbb{P}_{(x,y)\sim\mathcal{D}}(y\mathcal{F}(x) > c) = 1$ for a $W_0, D_0 \in \mathbb{N}_+, c > 0$. Then we have that for any $W \geq \Omega(W_0), D \geq \Omega(D_0)$ and $\delta > 0$, with probability at least $1 - \delta$ of $\mathcal{D}_{tr} \sim \mathcal{D}^N$, it holds $A_\mathcal{D}(\mathcal{F}) \geq 1 - O(e^{-W^D/K} + K^n\sqrt{\frac{\ln(K/\delta)}{N}})$ for all $\mathcal{F} \in \mathbf{M}_{W,D}(\mathcal{D}_{tr}, n)$, where $K = (\frac{c}{2^{D_0+2}W_0^{D_0-1}n})^{-1}$.*

However, this bound is relatively loose, and how to obtain a bound that is polynomial in $W_0, D_0, c$ is an important question.

*Proof.* **Part One**. For any given $\mathcal{D}_{tr} \sim \mathcal{D}^N$, we show that there is a network $\mathcal{F} \in \mathbf{H}_{W,D}$ such that $y\mathcal{F}(x) \geq [\frac{W}{W_0}]^{D_0-1} \frac{cW^{D-D_0}}{2}$ for any $(x,y) \in \mathcal{D}_{tr}$.

By the assumption of $\mathcal{D}$ in the theorem, let $\mathcal{F}_1 \in \mathbf{H}_{W_0,D_0}(n)$ satisfy $\mathbb{P}_{(x,y)\sim\mathcal{D}}(y\mathcal{F}_1(x) \geq c) = 1$. And $W_i$ is the $i$-th transition matrix of $\mathcal{F}_1$, $b_i$ is the $i$-th bias vector of $\mathcal{F}_1$.

We will construct $\mathcal{F}$ as $\mathcal{F} = \mathcal{F}_{p2} \circ \mathcal{F}_{p1}$, and we construct the two networks $\mathcal{F}_{p1}$ and $\mathcal{F}_{p2}$ as following:

$\mathcal{F}_{p1} : \mathbb{R}^n \to \mathbb{R}^W$ which has width $W$ and depth $D_0$, and the output layer of $\mathcal{F}_{p1}$ also uses the ReLU activation function.

Let $W$ be a matrix in $\mathbb{R}^{a,b}$ where $a, b \in \mathbb{N}^+$, and $T(W, a_1, b_1)$ is a matrix in $\mathbb{R}^{a_1,b_1}$ defined as: for any $i \in [a], j \in [b], k_1, k_2 \in \mathbb{Z}$, there are $T(W, a_1, b_1)_{k_1[\frac{a_1}{a}]+i,k_2[\frac{b_1}{b}]+j} = W_{i,j}$; other weights of $T(W, a_1, b_1)$ are 0. Then $\mathcal{F}_{p1}$ is defined as:

(1): The first transition matrix is $T(W_1, W, n)$, and the first bias vector is $T(b_1, W, 1)$;

(2): When $i > 2$, the $i$-th transition matrix is $T(W_i, W, W)$, and the $i$-th bias vector is $[\frac{W}{W_0}]^{i-1}T(b_i, W, 1)$.

Then, we have $\mathcal{F}_{p1}(x) = [\frac{W}{W_0}]^{D_0-1}\text{ReLU}(\mathcal{F}_1(x))$.

For $\mathcal{F}_{p2} : \mathbb{R}^W \to \mathbb{R}$, which has width $W$ and depth $D - D_0$, we define it as:

(1): When $i < D - D_0$, the $i$-th transition matrix is $\mathbb{I}_{W,W}$, and the $i$-th bias vector is 0, where $\mathbb{I}$ means all one matrix;

(2): The last transition matrix is $\mathbb{I}(1, W)$, and the last bias vector is $-[\frac{W}{W_0}]^{D_0-1}\frac{cW^{D-D_0}}{2}$.

Then, $\mathcal{F} = \mathcal{F}_2 \circ \mathcal{F}_1$ is what we want.

**Part two**. Similar to the proof of 4.4, there are at most $Ne^{-[\frac{W}{W_0}]^{D_0-1}\frac{cW^{D-D_0}}{4}+2}$ points in $\mathcal{D}_{tr}$ such that $y\mathcal{F}(x) \leq [\frac{W}{W_0}]^{D_0-1}\frac{cW^{D-D_0}}{4}$.

**Part three**. If $y\mathcal{F}(x) \geq [\frac{W}{W_0}]^{D_0-1}\frac{cW^{D-D_0}}{4}$, then $y\mathcal{F}(x') > 0$ for all $||x' - x||_\infty \leq \frac{c}{2^{D_0+1}W_0^{D_0-1}n}$.

As shown in Lemma K.2, there are $y\mathcal{F}(x') \geq [\frac{W}{W_0}]^{D_0-1}\frac{cW^{D-D_0}}{4} - W^{L-1}n||x - x'||_\infty$. So when $||x - x'||_\infty \leq \frac{c[\frac{W}{W_0}]^{L_0-1}}{4nW^{L_0-1}} \leq \frac{c}{2^{D_0+1}W_0^{D_0-1}n}$, there are $y\mathcal{F}(x') > 0$.

**Part four**. Let $r = \frac{c}{2^{D_0+1}W_0^{D_0-1}n}$. we can divide $[0,1]^n$ into $\frac{1}{(r/2)^n}$ disjoint cubes that have side length $r/2$. Then by part three, we know that in a cube, $\mathcal{F}$ gives the same label to every point in such a cube when $|\mathcal{F}(x)| \geq [\frac{W}{W_0}]^{D_0-1}\frac{cW^{D-D_0}}{2}$ for at least one $x$ in such cube.

**Part Five**. Prove the result.

By part four, name such $m$ cubes as $c_1, c_2, \cdots, c_m$, and let $\mathbb{P}_i = \mathbb{P}_{(x,y)\sim\mathcal{D}}(x \in c_i)$ and $\mathbb{P}_i \geq \mathbb{P}_j$ when $i \geq j$.

As shown in part four, let $S = \{i \in [N], \exists (x,y) \in \mathcal{D}_{tr} \cap c_i, y\mathcal{F}(x) \geq [\frac{W}{W_0}]^{D_0-1}\frac{cW^{D-D_0}}{4}\}$, then we have $A_\mathcal{D}(\mathcal{F}) \geq \sum_{i \in S} \mathbb{P}_i$.

For any $i$, by Hoeffding inequality, with probability $1 - e^{-N\mathbb{P}_i^2/2}$, there are at least $N\mathbb{P}_i/2$ points in cube $c_i$. So for any given $\epsilon_0 > 0$, let $\mathbb{P}_{k_0} \geq \epsilon_0$, then, with probability at least $1 - \sum_{i=k_0}^m e^{-N\epsilon_0^2/2}$ of $\mathcal{D}_{tr}$, there are at least $N\mathbb{P}_i/2$ points in $C_i$ for any $i \geq k_0$.

As shown in part two, there are at most $Ne^{-[\frac{W}{W_0}]^{D_0-1}\frac{cW^{D-D_0}}{4}+2}$ points in $\mathcal{D}_{tr}$ such that $y\mathcal{F}(x) \leq [\frac{W}{W_0}]^{D_0-1}\frac{cW^{D-D_0}}{4}$. So, by the above result, let $T = \{k_0, k_0+1, \ldots, N\}/S$ and $N(C_i)$ is the number

of points in $C_i$, with probability at least $1 - \sum_{i=k_0}^{m} e^{-N\epsilon_0^2/2}$ of $\mathcal{D}_{tr}$, there are $\sum_{i\in T} N\mathbb{P}_i/2 \le \sum_{i\in T} N(C_i) \le N e^{-[\frac{W}{W_0}]^{D_0-1}\frac{cW^{D-D_0}}{4}+2}$.

Hence,

$$
\begin{aligned}
&\mathbb{P}_{\mathcal{D}}(\mathcal{F}) \\
\ge\ & \sum_{i\in S} \mathbb{P}_i \\
\ge\ & 1 - \sum_{i\in[k_0]} \mathbb{P}_i - \sum_{i\in T} \mathbb{P}_i \\
\ge\ & 1 - m\epsilon_0 - 2e^{-[\frac{W}{W_0}]^{D_0-1}\frac{cW^{D-D_0}}{4}+2}.
\end{aligned}
$$

Now, we take $\epsilon_0 = \sqrt{\frac{2\ln(m/\delta)}{N}}$. We get the result. $\qquad\square$

A required lemma is given below.

**Lemma K.2.** *If a network with depth $L$ and width $W$, the $L_\infty$ norm of each transition matrix does not exceed 1. Then $|\mathcal{F}(x) - \mathcal{F}(z)| \le nW^{L-1}||x-z||_\infty$.*

*Proof.* It is easy to see that $||Relu(Wx+b) - \mathrm{ReLU}(Wz+b)||_\infty \le ||W(x-z)||_\infty \le ||W||_{1,\infty}||x-z||_\infty$. Let $\mathcal{F}_i$ is the output of i-th layer of $\mathcal{F}$, then

$$
\begin{aligned}
&|\mathcal{F}(x) - \mathcal{F}(z)| \\
\le\ & W||\mathcal{F}_{D-1}(x) - \mathcal{F}_{D-1}(z)||_\infty \\
\le\ & W^2||\mathcal{F}_{D-2}(x) - \mathcal{F}_{D-2}(z)||_\infty \\
\dots\ & \\
\le\ & W^{D-1}||\mathcal{F}_1(x) - \mathcal{F}_1(z)||_\infty \\
\le\ & nW^{D-1}||x-z||_\infty
\end{aligned}
$$

which proves the lemma. $\qquad\square$

## L   EXPERIMENTS

In this section, we give some simple experiments to validate our theoretical conclusions. Our experimental setup is as follows. We used MNIST data set and two-layer networks with ReLU activation function. When training the network, we ensure that the absolute value of each parameter is smaller than 1 by weight-clipping after each gradient descent. Two experiments are considered:

**About size and accuracy:** For networks with widths 100,200,...,900,1000, we observe their accuracy on the test set after training. The results are shown in Figure 1.

**About data and precision:** Using training sets with 10%, 20%, ..., 90%, 100% of the original training set to train a network with widths 200, 400 and 600. The results are shown in Figure 2.

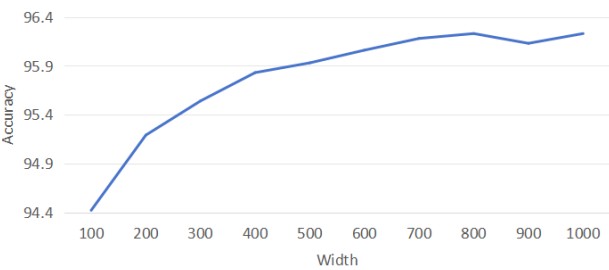

Figure 1: The accuracy on the different width networks.

Based on the experimental results, we have the following conclusions which confirm the correctness of Theorem 4.3.

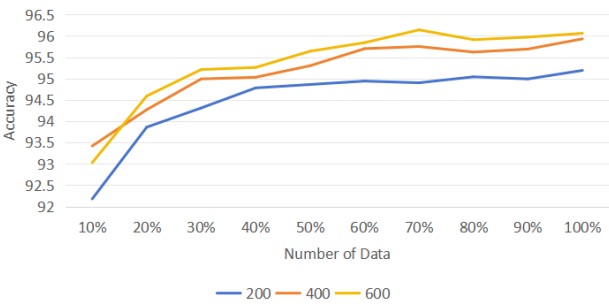

Figure 2: The accuracy on the 200,400,600 width networks with different number of data.

(1) Increasing the amount of data or enlarging the network leads to greater accuracy. Specifically, when there are fewer data (smaller networks), increasing the number of data (width of the network) leads to a greater improvement in accuracy, which is consistent with Theorem 4.3 where the number of data (network size) is located on the denominator.

(2) When the number of data is fixed, increasing the network size has a limitation effect on improving accuracy, as shown in Figure 1, which is consistent with Theorem 4.3, because the number of data cannot affect the item in the generalization bound about network size.

(3) When the network is small, increasing the number of data can only have a limited effect on improving accuracy, as shown in Figure 2. Accuracy on training 200-width network with the entire dataset is almost equivalent to accuracy on training 400-width network with 40% data in the entire dataset. This is consistent with Theorem 4.3, because the size of the network cannot affect the item in the generalization bound about the number of data.

