# OpenReview forum: "Generalizability of Neural Networks Minimizing Empirical Risk Based on Expressive Power"
_ICLR.cc/2025/Conference — ICLR 2025 Poster_

### Official Review · Reviewer_soP8 · 2024-10-28

**Soundness:** 2
**Presentation:** 2
**Contribution:** 2
**Rating:** 5
**Confidence:** 3

**Summary:**

This paper studies the generalization capabilities of two-layer neural networks (NNs) with small empirical error. Based on the expressive power of NNs, the authors derive a lower bound for classification accuracy, or equivalently, an upper bound for classification error, in NNs trained with minimum empirical risk. Their results show that large network width and large sample size can lead to high classification accuracy. Additionally, this conclusion extends to NNs with somewhat higher empirical risk. By their theoretical analysis, the authors provide insights on factors influencing generalization, such as the choice of activation functions, the role of overparameterization, and the impact of loss function selection

**Strengths:**

1. Unlike many previous results that rely on bounded loss functions, this paper analyzes the more practically relevant cross-entropy loss. Additionally, the theoretical results in this paper do not depend on any convexity or smoothness assumptions of the loss function.

2. The derived lower bound on classification accuracy suggests that wider NNs have more potential for high accuracy, which is desirable for deep learning theory.

**Weaknesses:**

1. Although the authors acknowledge that restricting the analysis to two-layer NNs is a limitation, there are some additional constraints in problem setups, such as focusing only on binary classification tasks and constraining the parameter space to $[-1,1]^d$ (where $d$ is the number of total parameters). It seems that these constraints are essential for the theoretical developments, and further relaxing these constraints does not seem straightforward.

2. My another concern arises from the use of Rademacher complexity to derive the lower bound on accuracy (or equivalently, the upper bound on error), which could be loose. For example, Corollary 4.4 in this paper indicates that Theorem 4.3 requires the width and sample size to be sufficiently large for the lower bound to be non-vacuous (i.e., the lower bound itself is non-negative). Thus, outside the large-sample regime, Theorem 4.3 may lack practical relevance, limiting its applicability. In fact, considering that the paper already studies generalization for empirical risk minimizers, it might be more interesting to use bounds based on local Rademacher complexity rather than the original Rademacher complexity, which could give a decay rate of $O(1/N)$ instead of $O(1/\sqrt{N})$ for hypotheses with low risk or variance.

Additional concerns are outlined in the questions below.

**Questions:**

1. Along with the constrained parameter space, the input data space is assumed to lie within $[0,1]^n$, is it possible to relax this requirement? I think normalizing input data to $[-1,1]^n$ is also common in practice.

2. In the proof of Proposition 3.2, it seems that the bounded domain of the parameter space play a critical role in proving the existence of an empirical risk minimizer. How would this apply to practical scenarios with an unbounded parameter space? Moreover, if the cross-entropy loss used in Proposition 3.2 has the reachable upper and lower bounds, does that imply it is also a "bad" loss function as defined in Definition 6.6?

3. In the proof of Proposition 4.2, in Line 723-725, it’s stated that $\mathcal{F}_A$ is a network whose parameter is the corresponding parameter of $\mathcal{F}$ divided by $A$, with $\mathcal{F}_A=\mathcal{F}/{A^2}$. Could you clarify why this equality holds? In addition, if $A<1$, then each parameter of $\mathcal{F}_A$ might exceed the domain $[-1,1]$, it seems that the parameter domain constraint will be violated.

4. In the proof of Theorem 4.3, could you explain how the $L_{1,\infty}$ norm for the three transition matrices in Line 786 were obtained? Additionally, if input data is not restricted to $[0,1]^n$ and the parameter space is unbounded, can these $L_{1,\infty}$ norms still be derived? Furthermore, in Line 838-839, the inequality $|S|< Ne^{-kc/2+2}$ is only meaningful if $kc\geq 4$, as $|S|\leq N$ clearly holds. This is also implied in Line 853, where the lower bound would be vacuous for $kc\leq 4$ since $\mathbb{E}_{(x,y)\sim\mathcal{D}}yg(x)\geq -\frac{kc}{2}$ already holds trivially. Perhaps adding a condition such as $W\geq \frac{4(W_0+1)}{c}$ in the theorem statement might improve clarity.

5. The motivation for the loose results in Section 5.1 is unclear, as the conclusions and insights from these $W$-independent results seem well-known.

6. In your abstract, you mention that the theoretical results in this work can provide insights into robust overfitting, but what you explore in Section 6.1 is not related to the robust overfitting phenomenon, which is proposed in [R1]. Perhaps "robust generalization", as used in the introduction, would be a more accurate term.

[R1] Leslie Rice, Eric Wong, and Zico Kolter. "Overfitting in adversarially robust deep learning." International conference on machine learning. PMLR, 2020.

Minor comments:

1. Some references are missing. For example, stability-based bounds have been extended beyond Hardt et al. (2016) to cover nonsmooth cases (e.g., [R2, R3]), among others. Additionally, PAC-Bayesian and information-theoretic generalization bounds are well-known for being algorithm-dependent and, in some cases, data-dependent. These methods generally do not assume Lipschitz continuity, convexity, or smoothness and some derive fast-rate bounds in the low empirical risk regime. Refer to [R4, R5] for further reading on these types of generalization bounds.

[R2] Raef Bassily, et al. "Stability of stochastic gradient descent on nonsmooth convex losses." Advances in Neural Information Processing Systems 33 (2020): 4381-4391.

[R3] Yunwen Lei. "Stability and generalization of stochastic optimization with nonconvex and nonsmooth problems." The Thirty Sixth Annual Conference on Learning Theory. PMLR, 2023.

[R4] Pierre Alquier. "User-friendly introduction to PAC-Bayes bounds." Foundations and Trends® in Machine Learning 17.2 (2024): 174-303.

[R5] Fredrik Hellström, et al. "Generalization bounds: Perspectives from information theory and PAC-Bayes." arxiv preprint arxiv:2309.04381 (2023).

2. The paper would benefit from substantial proofreading, as there are numerous typos (e.g., Line 092: "reached is minimum" ---> "reached its minimum"; Line 082: "robust memorizing"--->"robustly memorizing", ...) and inconsistencies in notation (e.g., $\mathcal{F}$ vs. $F$; $Z_{2W}(n)$ vs. $\mathbf{H}_{2W}(n)$, ...). Please review the manuscript carefully to identify and fix these issues.

---

> ### Author Response · Authors · 2024-11-19
> **Rebuttal by Authors**
>
> Thank you for acknowledging the novelty of our paper (especially in comparison to previous results) as well as for providing valuable feedback. Below we address the detailed comments, and hope that you can find our response satisfactory.
>
> ***Question 1: Although the authors acknowledge that restricting the analysis to two-layer NNs is a limitation, there are some additional constraints in problem setups, such as focusing only on binary classification tasks and constraining the parameter space to $[-1,1]^{W_d}$ (in the proof of Proposition 3.2). It seems that these constraints are essential for the theoretical developments, and further relaxing these constraints does not seem straightforward.***
>
> Answer: These two constraints can be removed or reasonably relaxed as explained below.
>
> (1) The assumption of binary classification can be removed. We use binary classification in this paper because the description of binary classification problems is very concise, and many previous papers have focused on binary classification. For multi-label classification, we can change the network output dimension and the loss function, and our proof ideas can be transferred to multi-label classification problems.
>
> (2) The parameter domain $[-1,1]^{W_d}$ can be changed to $[-E,E]^{W_d}$ for some fixed $E\in R_+$. In order to ensure the existence of a network that minimizes the empirical risk, the parameter space must be a bounded closed set. Otherwise, the empirical risk can arbitrarily approach 0, but $argmin_{F} \sum_{(x,y)\in D_{tr}} Loss(F(x),y)$ is empty. Here is a short proof of this fact. Assume that $F$ satisfies $yF(x)>0$ for all $(x,y)\in D_{tr}$. Let $F_A(x)=AF(x)$ (only expand the value of parameters in the last layer by $A$ times) for any real number $A>1$. Then it holds that $\sum Loss(F_{A_1}(x),y)<\sum Loss(F_{A_2}(x),y)$ when $A_1<A_2$. Therefore, the empirical risk has no minimum value. On the other hand, in practice, the infinite norm of a network is easy to control and will not infinitely increase with the increase of the network. For example, use CIFAR10 to train ResNet18 with weight decay 0.0005, the $L_\infty$ norm of the parameters is smaller than $0.6$.

---

> ### Author Response · Authors · 2024-11-19
> **Rebuttal by Authors**
>
> ***Question 2: My another concern arises from the use of Rademacher complexity to derive the lower bound on accuracy, which could be loose. Corollary 4.4 in this paper indicates that Theorem 4.3 requires the width and sample size to be sufficiently large for the lower bound to be non-vacuous (i.e., the lower bound itself is non-negative). Thus, outside the large-sample regime, Theorem 4.3 may lack practical relevance, limiting its applicability. In fact, considering that the paper already studies generalization for empirical risk minimizers, it might be more interesting to use bounds based on local Rademacher complexity rather than the original Rademacher complexity, which could give a decay rate of $O(1/N)$ instead of $O(1/\sqrt{N})$ for hypotheses with low risk or variance.***
>
> Answer:
>
> (1) We believe that the statement by the reviewer that **"outside the large-sample regime, Theorem 4.3 may lack practical relevance" is not complete**. Large-sample size is actually a necessary condition for generalization. The main contribution of Theorem 4.3 is that the requirement for sample size $N\ge \Omega(...)$ and the requirement for the size of the neural network $W\ge \Omega(...)$ in Corollary 4.4 are separated and only depend on distribution itself, which is not the case in Theorem 4.8 (based on VC-dimension). In particular, this makes Theorem 4.3 consistent with the theoretical mystery of deep learning: “nice generalization ability of over-parameterized network” [1], whereas the VC-dimension based Theorem 4.8 contradicts this phenomenon, because $VC(H)=O(Wn)>N$ in the over-parameterized regime.
>
> [1] M. Belkina,D. Hsuc, S. Maa, S. Mandala, Reconciling modern machine-learning practice and the classical bias–variance trade-off, 2019 (Fig. 1).
>
> (2): **Local Radermacher complexity** can be used, but there exists other problems. Local Radermacher complexity is a complexity measure of the subspace composed of smooth functions in a hypothesis space, for instance the $f\in H$ satisfying $E(f^2)<r$. If we try to use local Radermacher complexity to prove Theorem 4.3, then we need to find a constant $r$ such that $E(f^2)<r$ for all these $f$ which can minimize the empirical risk based on a dataset i.i.d selected from distribution. The problem is that such an $r$ is quite big. It is easy to see that such an $r$ will depend on $W_0,c,n,W,\sigma$ (still use the symbol in Theorem 4.3). Then by the results shown in [2], the generalization bound is not more than $O(Rad(N,r)+\sqrt{rlog(1/Delta)/N}+log(1/Delta)/N)$. When $r$ is very big, the middle item involving $r$ will be a problem to achieve generalization.
>
> [2]: P.L. Bartlett, O. Bousquet, and S. Mendelson, “Local Rademacher Complexities,” Ann. Statist., 2005.
>
> (3): **Regarding your concerns about the looseness of the bound**, by using the technique from [3], we can obtain a generalization bound with a convergence speed of $1/N$. But there will be a serious problem: the sample complexity derived from this generalization bound still depends on the VC-dimension of the hypothesis space, which is avoided in our Theorem 4.3. Corollary 4.4 has already shown that the sample complexity obtained from our conclusion depends entirely on the distribution itself, **which is an improvement compared to the existing results**. An interesting problem is to find a generalization bound with faster convergence speed whose derived sample complexity only depends on the distribution itself? We currently do not have a particularly good idea for this issue.
>
> [3]: E. Giné, V. Koltchinskii, Concentration inequalities and asymptotic results for ratio type empirical processes, Ann. Probab. 34 (3) (2006) 1143–1216.
>
> ***Question 3: Along with the constrained parameter space, the input data space is assumed to lie within $[0,1]^n$. Is it possible to relax this requirement? I think normalizing input data to $[-1,1]^n$ is also common in practice.***
>
> Answer: This is not a strong assumption, as we can always transform the data to $[0,1]^n$. Actually, we can assume that the data is located in any bounded closed domain $[E,F]^n$ and the theory is still valid. Note that the universal approximation theorem is for compact domains, so the data must be in a bounded area.

---

> ### Author Response · Authors · 2024-11-19
> **Rebuttal by Authors**
>
> ***Question 4: In the proof of Proposition 3.2, it seems that the bounded domain of the parameter space play a critical role in proving the existence of an empirical risk minimizer. How would this apply to practical scenarios with an unbounded parameter space? Moreover, if the cross-entropy loss used in Proposition 3.2 has the reachable upper and lower bounds, does that imply it is also a "bad" loss function as defined in Definition 6.6?***
>
> Answer:
> (1) In order to ensure the existence of a network that minimizes the empirical risk, the parameter space must be a bounded closed set. Otherwise, the empirical risk can arbitrarily approach 0, but $argmin_{F} \sum_{(x,y)\in D_{tr}} Loss(F(x),y)$ is empty.
> Here is a short proof of this fact. Assume that $F$ satisfies $yF(x)>0$ for all $(x,y)\in D_{tr}$. Let $F_A(x)=AF(x)$ for any real number $A>1$. Then it holds  $\sum Loss(F_{A_1}(x),y)<\sum Loss(F_{A_2}(x),y)$ when $A_1<A_2$. Therefore, the empirical risk has no minimum value.
>
> (2) We say that a network $F$ can minimize empirical risk under a dataset when using cross-entropy loss, which means that $argmin_{f\in H} \sum_{L_{CE}(F(x),y)}$ is not empty.
>
> In Definition 6.6, we say that a loss function $L$ is bad in (1) means $argmin_{x\in R}L(x,1)$ and $argmin_{x\in R}L(x,-1)$ have solutions for some $x$, which is not related to the dataset and network, only related to the function itself.
>
> These two concepts are totally different and have no relationship.
>
> ***Question 5: In the proof of Proposition 4.2, in Line 723-725.
> Could you clarify why the equality $F_A=F/A^2$ holds? In addition, if $A<1$, then each parameter of $F_A$ might exceed the domain [-1,1], it seems that the parameter domain constraint will be violated.***
>
> Answer: As said in the proof, $A$ is the maximal value of network $F$, where $F=\sum a_i Relu(W_ix+b_i)$ given by Theorem A.1 (Please note that the network considered in the work [George Cybenko 1989] does not have the last bias value $c$). Then $F_A=\sum (a_i/A)Relu((W_i/A)x+b_i/A)=\sum (a_i/A)Relu(W_ix+b_i)/A=\sum a_iRelu(W_ix+b_i)/(A^2)=F/(A^2)$. Because $|a_i/A|\le 1$, $||W_i/A||_\infty\le1$, $|b_i/A|\le 1$. So $F_A$ is what we want.
>
> If we consider the last bias value $c$, then $F=\sum a_iRelu(W_ix+b_i)+c$. Also, let $A$ be the maximum value of network $F$. Then when $A\le 1$, $F$ is what we want. If not, let $F_A=\sum (a_i/A)Relu((W_i/A)x+b_i/A)+c/A^2=\sum (a_i/A)Relu(W_ix+b_i)/A+c/A^2=\sum (a_iRelu(W_ix+b_i)+c)/(A^2)=F/(A^2)$, and that is what we want.
>
> However, you did remind us to only consider the situation where $A>1$, which we have added in the next version of the article.

---

> ### Author Response · Authors · 2024-11-19
> **Rebuttal by Authors**
>
> ***Question 6. In the proof of Theorem 4.3, could you explain how the $L_{1,\infty}$ norm for the three transition matrices in Line 786 were obtained? Additionally, if input data is not restricted to $[0,1]^n$ and the parameter space is unbounded, can these $L_{1,\infty}$ norms still be derived? Furthermore, in Line 838-839, the inequality $S<NS^{kc/2+2}$ is only meaningful if $kc\ge 4$, as $|S|\ge N$ clearly holds. This is also implied in Line 853, where the lower bound would be vacuous for $kc\le 4$ since already holds trivially. Perhaps adding a condition such as  $W\ge Frac{4(w_0+1)}{c}$ in the theorem statement might improve clarity.***
>
> Answer:
>
> (1): The first transition matrices: this layer is the same as the first layer of $f$. Consider the bound of values of parameters of $f$, its $L_{1,\infty}$ norm is $n+1$.
>
>  The second transition matrices: Let the second transition matrices of $f$ be $W$, bias be $c$. Then the second transition matrices of $F$ is $W/k$, bias is $c/k+a/k$. Using the bound of value of parameters of $f$ and the value of $k$, we get the result.
>
>  The third transition matrix: It is $(1,1,\dots,1,-1,-1,\dots,-1)$, where there are $k$ number of 1 and $k$ number of -1 in it, and we get the result.
>
> (2): As long as the range of data are bounded, it is fine, the difference in the final impact on the conclusion is a constant. If the value of parameters is unbounded, then there are no minimum empirical error, as said in the above questions.
>
> (3): $K$ is directly related to the network width $W$. Because we study all $W$, it naturally includes some simple situations when $W$ is small, but we should focus mainly on the case where $W$ is large. We can add this assumption later.
>
> ***Question 7: The motivation for the loose results in Section 5.1 is unclear.***
>
> Answer: Section 5.1 mainly explains the relationship between the lower bound of the sample complexity for a distribution and the network size required to express the distribution. For example, Corollary 5.3 indicates that for some data distributions requiring networks with width $W_0$ to express, there exists a network requiring at least $\Omega(nW_0)$ training data to ensure generalization, which give a NECESSARY condition for generalization.
>
> However, in general, this relationship is difficult to obtain, so we were unable to provide a very tight conclusion. The given bounds, though relatively loose, are the first results of this type.
>
> Our lower bound on sample complexity is entirely based on the cost of network expressed distribution. Although there have been previous studies on calculating lower bounds, there has been no work based on this, so our results are not well-known.
>
> ***Question 8: In your abstract, you mention that the theoretical results in this work can provide insights into robust overfitting, but what you explore in Section 6.1 is not related to the robust overfitting phenomenon, which is proposed in [R1]. Perhaps "robust generalization", as used in the introduction, would be a more accurate term.***
>
> Answer: This is true, and we have modified these. Theorem 6.2 gives an upper bound for robust accuracy. Thanks.
>
> ***Question 9: Some references are missing.***
>
> Answer: Thanks for pointing out the related works. We will provide a more detailed discussion on these related works in the next revision.
>
> ***Question 10: The paper would benefit from substantial proofreading.***
>
> Answer: Thanks for pointing out these issues. We will correct them in the revision.

---

> > ### Author Response · Authors · 2024-11-25
> >
> > As the discussion phase is concluding soon, we would greatly appreciate your feedback on whether our rebuttal has adequately addressed your concerns. Please feel free to bring up additional discussion if needed.

---

> ### Author Response · Authors · 2024-12-03
>
> We kindly invite you to review our rebuttal as the discussion period comes to an end. We also welcome your opinions on this paper or any  questions we have not yet resolved.

---

> > ### Comment · Reviewer_soP8 · 2024-12-03
> >
> > I would like to thank the authors for their detailed responses and sincerely apologize for not being active during the discussion phase.
> >
> > Regarding the tightness of your results, my concern remains. As you noted, achieving fast-rate bounds would require incorporating the VC-dimension, which this paper aims to avoid. Additionally, regarding local Rademacher complexity, if I understand correctly, you are attempting to lower bound the model’s accuracy, which is equivalent to upper bounding the error (bounded in $[0,1]$). Thus, the value of $r$  in this context would be at most 1. My previous comments stem from the fact that your focus on the ERM solutions may place the resulting hypotheses within a low-variance error regime, making $r$ negligible (see [1, Section 5.3]).
> >
> > Moreover, some studies provide generalization bounds that do not depend on the VC-dimension while still achieving fast-rate results, such as PAC-Bayesian bounds (see [2, Section 4] and [3]). I believe these techniques could provide tighter results than those based on original Rademacher complexity, particularly when focusing on ERM solutions.
> >
> > Once again, I apologize for my last-minute engagement. I will read the feedback from other reviewers and your corresponding responses, and I will remain open to discussing my concerns regarding the tightness of the results during the AC-reviewer discussion period.
> >
> > [1] Stéphane Boucheron, Olivier Bousquet, and Gábor Lugosi. Theory of classification: A survey of some recent advances. ESAIM: probability and statistics, 9:323–375, 2005.
> >
> > [2] Pierre Alquier. "User-friendly introduction to PAC-Bayes bounds." Foundations and Trends® in Machine Learning 17.2 (2024): 174-303.
> >
> > [3] Tolstikhin, I. O. and Seldin, Y. Pac-bayes-empiricalbernstein inequality. Advances in Neural Information Processing Systems, 26, 2013.

---

> ### Author Response · Authors · 2024-12-03
>
> Thank you for your reply. We would like to emphasize a few more points here:
>
> 1: Firstly, there are indeed many studies that have achieved a generalization bound convergence speed of $1/N$, and from this perspective, this is a weakness for us. However, the core purpose of this article is not to improve convergence speed, as mentioned in our article, the advantage of our generalization bound is
>
> ***the number of data and network size is entirely on the denominator in the generalization bound,***
>
> , which was not achieved by previous generalization bounds, based on that, we can explain phenomena such as over-parameterized, and get the number of data and network size which depend only on the distribution required to ensure generalization.
>
> 2: ***Can we get a generalization boundary with a convergence speed of $1/N$, and guarantee the number of data and network size is entirely on the denominator in the generalization bound?*** At present, we do not know how to achieve it. The technique from [3] can not make the number of data and network size to be entirely on the denominator in the generalization bound, bacause VCdim is on the numerator. The Radermacher Complexity calculated during our proof  does not consider the loss function, if the loss function is considered in local Radermacher Complexity, it is still hard to give the upper bound about $1/\sqrt{N}$ for $r$. More over, the resulting generalization bound also cannot  make the number of data and network size to be entirely on the denominator in the generalization bound, bacause Rad(N,r) is on the numerator, such value is obviously influenced by network size. At last, because what we want to prove is a generalization bound for any network, and the PAC-Bayes technique is to prove the bound for most networks, so such techniques may not be applicable to us here.
>
> This idea can be a future work.

---

### Official Review · Reviewer_1rDw · 2024-11-03

**Soundness:** 3
**Presentation:** 2
**Contribution:** 3
**Rating:** 6
**Confidence:** 3

**Summary:**

This paper investigates the generalizability of neural networks trained by empirical-risk-minimization (ERM) algorithms, focusing on understanding the factors that contribute to their ability to generalize well to unseen data. The authors consider two-layer networks and approach generalizability from the perspective of the network's expressive ability, which refers to the network's capacity to represent complex functions and effectively fit the underlying data distribution. The paper establishes a lower bound for the accuracy of neural networks that minimize empirical risk, suggesting that these networks can generalize effectively given sufficiently large training datasets and network sizes. The paper further investigates the lower bound by examining scenarios without enough data. The paper finally provides insights into several observed phenomena in deep learning, including robust overfitting, the importance of over-parameterization, and the impact of loss functions.

**Strengths:**

1. The paper explores generalization from a unique perspective by connecting it to the expressive ability of neural networks, providing a fresh perspective on understanding why neural networks generalize well.
2. The paper do not place strong assumptions on data or loss functions, making the results more applicable to practical scenarios.
3. The paper highlights the importance of choosing appropriate network models and activation functions tailored to the specific data distribution to enhance generalization capabilities.

**Weaknesses:**

1. The focus on two-layer networks might limit the applicability of the findings to more complex and deeper network architectures prevalent in practice.
2. The paper primarily focuses on theoretical analysis and does not include empirical studies to validate its claims and insights.
3. The assumptions on separable data distributions potentially oversimplifies the complexities of real-world deep learning applications.

**Questions:**

1. In Theorem 1.1., please clarify the meanings of "expressing the data distribution with a neural network" and "with high probability of a dataset".
2. Under Theorem 1.2, what are the definitions of "robust memorizing" and "robust fitting"?
3. Why is "positive separation bound" important for the data distributions? How would the results change if the data distribution does not have a positive separation bound?
4. In Section 5, the authors provide upper bound for accuracy without enough data. Could the authors relate the upper bound and the previously derived lower bound and have some discussions?
5.  Under Theorem 6.2, it would be better to elucidate more on the dependency of $c_1$ on $\epsilon$.
6. In Proposition 6.5, how do the numbers "0.9" and "0.6" come out? Similarly for Theorem 6.7.

---

> ### Author Response · Authors · 2024-11-19
> **Rebuttal by Authors**
>
> We thank you for acknowledging the novelty of our paper as well as providing the valuable feedback. Below we address the detailed comments, and hope that you can find our response satisfactory.
>
> ***Question 1: The focus on two-layer networks might limit the applicability of the findings to more complex and deeper network architectures prevalent in practice.***
>
> Answer: First, it should be mentioned that many theoretical papers focus on two-layer networks, although lacking some practical applications, it can help understand and explain some phenomena in neural networks.
>
> Second, extending our results to deep networks is actually a challenge; the real difficulty is how to ensure that sample complexity is independent of network size. At present, we do not have a way to achieve that. Moreover, because most theoretical analyses are difficult on the deep network, (without special assumptions), not just in terms of Radermacher Complexity, but also Gradient Descent Analysis, Robustness Analysis and so on, so it is not practical to directly use these methods, and new methods need to be developed to extend our result to deep network.
>
> ***Question 2: The paper primarily focuses on theoretical analysis and does not include empirical studies to validate its claims and insights.***
>
> Answer: This is indeed a theoretical paper which provides a new and better generalization bound such that BOTH the network size and number of training data are completely in the denominator, as shown in Theorem 4.3.
>
> For experiments in the revised paper, we can consider the following experiments: We training the network and control the absolute values of the parameters not more than 1 during training. Then we observe the relationship curve between the generalization and the network size, number of data. Contrast the relationship between this curve and Theorem 4.3.
>
> ***Question 3: The assumptions on separable data distributions potentially oversimplifies the complexities of real-world deep learning applications. Why is "positive separation bound" important for the data distributions? How would the results change if the data distribution does not have a positive separation bound?***
>
> Answer:  In fact, most classification problems are separable in the real world, such as MNIST, where different numbers cannot be infinitely close to each other.
>
> We use the separability assumption because the distribution that satisfies this assumption must can be expressed by the network, as said in Pro3.2. In fact, the conclusion of Theorem 4.3 applies to all distributions that can be expressed by network even without separability.
>
> If without such assumption, some inseparable data can lead to networks being unable to learn. Here is an example: we construct a distribution $D$ in $[0,1]\times\{-1,1\}$ which can not be learned by network.
>
> The $(x,y)\sim D$ is: (1) Randomly select a number in $\{-1,1\}$ as the label $y$.
> (2) If we get $1$ as the label, then randomly select an irrational number in $[0,1]$ as samples $x$; if we get $-1$ as the label, then randomly select a rational number in $[0,1]$ as samples $x$.
>
> ***Question 4: In Theorem 1.1, please clarify the meanings of "expressing the data distribution with a neural network" and "with high probability of a dataset".***
>
> Answer: Thank you, we have rewrite the th1.1 in the paper. ''expressing the data distribution with a neural network" means the definnition 4.1, ''with high probability of a dataset" means $1-\delta$ mentioned in th4.3.
>
> ***Question 5: Under Theorem 1.2, what are the definitions of "robust memorizing" and "robust fitting"?***
>
> Answer: Robust memorizing means that the network is robust on a given dataset $ D_{tr}$: $F(x+\epsilon)=y$ for all $||\epsilon||<r$ and $(x,y)\in D_{tr}$. Robust fitting is the same as robust memorizing. We will correct this problem in the revision.
>
> ***Question 6: In Section 5, the authors provide an upper bound for accuracy without enough data. Could the authors relate the upper bound and the previously derived lower bound and have some discussions?***
>
> Answer: Our lower and upper bounds depend only on the cost required for the network to express the distribution itself, upper bounds for sufficient conditions for generalization  and lower bounds for necessary conditions for generalization. They reflects the relationship between the amount of data required for generalization and the cost of expressing the distribution. We cannot make them equal yet. In the future, if we can make them equal, we have found the necessary and sufficient conditions for achieving generalization.

---

> ### Author Response · Authors · 2024-11-19
> **Rebuttal by Authors**
>
> ***Question 7: Under Theorem 6.2, it would be better to elucidate more on the dependency of $c_1$ on $\epsilon$.***
>
> Answer: We have add it in remark 6.3.
>
> ***Question 8: In Proposition 6.5, how do the numbers "0.9" and "0.6" come out? Similarly for Theorem 6.7.***
>
> Answer: We use 0.99 to express high accuracy for some $f\in H_{W_0}(n)$, and it can be changed as $1-\delta$ for any $\delta$.
>
> We use 0.6 to express low accuracy for any $f\in M_{W_0}( D_{tr},n)$, it can be changed as $0.5+\delta$ for any $\delta$.
>
> We provide specific numbers only for simple expression and writing. We will add this information in the revised edition of the paper.

---

> > ### Comment · Reviewer_1rDw · 2024-11-26
> >
> > We thank the reviewers for your detailed response.
> >
> > - To A2: What I meant is to plot the theoretical bound and empirical error together to make a comparison.
> >
> > - To A3: So Proposition 3.2 is one case that ensures that the data distribution can be expressed by a neural network. It would be interesting to know other sufficient conditions. In addition, in Definition 4.1, I wonder how the result will change if we let $1$ be $1-\epsilon$, which can possibly include cases where the data are inseparable, such as binary mixture Gaussian dataset.

---

> > > ### Author Response · Authors · 2024-11-26
> > >
> > > ***Question: What I meant is to plot the theoretical bound and empirical error together to make a comparison.***
> > >
> > > Answer: In the new version of the paper, we have included some simple experiments (in Appendix L) to verify our Theorem 4.3, hoping to solve your problem.
> > >
> > > ***Question: It would be interesting to know other sufficient conditions.***
> > >
> > > Answer:
> > > In order for the generalization bound in theorem 4.3 to approach 1 arbitrarily when  the network is large enough and there is enough data, it is necessary for the distribution to be expressed by the network as probability 1.
> > >
> > > If we remove the separable assumptions about the distribution and definition 4.1 only require $1-\epsilon$ accuracy on data distribution (such as binary mixture Gaussian distribution). We can still obtain a conclusion similar to Theorem 4.3, but slightly weaker: when the network is large enough and there is enough data, the generalization bound cannot arbitrarily approach 1, but arbitrarily approach $1-\epsilon$. This is actually reasonable, because the upper limit of the generalization bound is the highest accuracy that a network can achieve in expressing a distribution.

---

> > > > ### Comment · Reviewer_1rDw · 2024-12-02
> > > >
> > > > Thank you for the response. I would like to keep my score unchanged.

---

> > > > > ### Author Response · Authors · 2024-12-02
> > > > >
> > > > > Thank you very much for acknowledging our contributions and proposing helpful suggestions for improving the paper.

---

### Official Review · Reviewer_6ywY · 2024-11-03

**Soundness:** 4
**Presentation:** 3
**Contribution:** 4
**Rating:** 8
**Confidence:** 4

**Summary:**

The authors provide a number of useful generalization results for neural networks. They base their analysis on a definition of expressivity of *distributions* rather than functions (the classic universal approximation theory framework), and focus specifically on the case of distributions that satisfy a strict separability assumption, that implies that the Bayes risk is 0. The expressivity definition gives a natural, architecture-dependent measure of distribution complexity, W0. The authors focus do an algorithm-dependent analysis, focusing of empirical risk minimizers.

The main result is a lower bound on test accuracy of ERM, which depends on the ratio between W0 and the network’s width, W, and the number of training examples, N. Roughly, it implies that having a network width greater than W0 and a number of training examples which exceeds W0*[dimension] is sufficient for generalization.

They provide further analysis to follow-up on the main result, including upper bounds on generalization when the dataset is not big enough, generalization results for when ERM yields a local optimal point, and the explanation for various interesting phenomena (robustness, overparametrization, etc). The authors also provide discussion of implications of their results, and comparison to other existing results in literature.

**Strengths:**

The explicit dependence of the bounds on both training set size, but also the width of the network is not common in statistical learning theory bounds and is very important to explain various width-related generalization phenomena.

There is thorough analytical follow-up of the main result, including upper bounds on generalization when the dataset is not big enough, generalization results for when ERM yields a local optimal point, and the explanation for various interesting phenomena (robustness, overparametrization, etc). The authors also provide good discussion of implications of their results, and comparison to other existing results in literature.

The literature review, and comparison to relevant literature is complete.

**Weaknesses:**

1. Many grammatical errors and typos. Most of them are inconsequential for comprehension, but some actually make check the validity difficult:  Line 723: “Miximum” is that a maximum or a minimum?

2. While the authors state their positive separation assumption in Definition 3.1, it would improve clarity if they repeated in the statements of subsequent theorems that they only apply to distributions that satisfy the separation. Same for Lin 304-305. The results are discussed as if they hold for any distribution, which is not true.

3. There are some minor issues with rigour:
- Line 721: “for all (x,y)~D” is an odd statement. It is not clear whether the authors mean “surely” or “almost surely” with respect to distribution D.
- The infimum in Definition 3.1 is not well defined. Is (z,-1)~D meant to signify any z in the support of D conditional on the label being -1? This needs to be clarified.

4. [minor] The pervasive use of passive voice can be confusing. The authors use passive voice interchangeably for their own contributions, and for existing results (by other authors) in literature.

**Questions:**

Please look at weaknesses section for some comments that might require a response.

---

> ### Author Response · Authors · 2024-11-19
> **Rebuttal by Authors**
>
> Thank you for appreciating our new contributions as well as providing the valuable feedback. Below we address the detailed comments, and hope that you can find our response satisfactory.
>
> ***Question 1: Many grammatical errors and typos. Most of them are inconsequential for comprehension, but some actually make check the validity difficult: Line 723: “Miximum” is that a maximum or a minimum?***
>
> Answer: It is maximum. Sorry for that. We have modified these typos in the revised paper.
>
> ***Question 2: Although the authors state their positive separation assumption in Definition 3.1, it would improve clarity if they repeated in the statements of subsequent theorems that they only apply to distributions that satisfy the separation. Same for Lin 304-305. The results are discussed as if they hold for any distribution, which is not true.***
>
> Answer: Thank you, we have already explained it in the new revised paper.
>
> ***Question 3: Line 721: “for all $(x,y)$~$D$” is an odd statement. It is not clear whether the authors mean “surely” or “almost surely” with respect to distribution D.***
>
> Answer: ‘for all $(x,y)$~$D$’ has been replaced with 'with probability 1 under distribution D', we have modified these typos in the revised paper.
>
> ***Question 4: The infimum in Definition 3.1 is not well defined. Is (z,-1)~D meant to signify any z in the support of D conditional on the label being -1? This needs to be clarified.***
>
> Answer:  Yes, this represents any sample z with the label -1 selected from the distribution.
>
> ***Question 5: The pervasive use of passive voice can be confusing. The authors use passive voice interchangeably for their own contributions, and for existing results (by other authors) in literature.***
>
> Answer: Thanks for pointing out the potential confusion caused by the frequent use of passive voice, such as lines 19–21. We will address this issue in the revised version.

---

### Official Review · Reviewer_WTwE · 2024-11-04

**Soundness:** 4
**Presentation:** 3
**Contribution:** 3
**Rating:** 8
**Confidence:** 3

**Summary:**

The paper addresses the generalization of neural networks from the perspective of their expressive power. The authors provide new generalization bounds based on a network’s expressive capacity and without strong assumptions. The paper also provide a lower bound of generalizability. Additionally, the paper explores implications for over-parameterized networks, robustness, and the impact of different loss functions on generalizability.

**Strengths:**

This paper provide a novel generalization bound based on expressive power of the network, which is different to traditional bounds. The assumption that there exists a network separates the distribution is more natural in practice than convexity or NTK. With rigorous analysis to both sample comlexity and lower bound of generalizability, the paper shows integrity of this research topic. Moreover, this work also provide some insights to the phenomena in deep learning such as overparameterization, robustness and the effect of different loss functions. Additionally, the paper is well organized and easy to understand.

**Weaknesses:**

This is a good paper in general. But I have some concerns about the contribution. The main results of the paper is showing the generalizability of shallow ReLU networks on a positive separated distribution that can be expressed by a smaller network. This maybe not a siginificant contribution since it seems really natural. Intuitively, if the data distribution can be separated by a network, there must exist functions in the class of larger networks that can also separate the data distribution. With large enough sample size, the ERM solution certainly can generalize with high probability. And techinically, there are Rademacher complexity type bounds with sample complexity $O(1/\sqrt{N})$ [1]. The only difficult of the main theorem is to estimate the Rademacher complexity of the class of larger ReLU networks.
So I doubt a little bit about the contribution of this paper.

[1] Shai Shalev-Shwartz & Shai Ben-David. Understanding Machine Learning: From Theory to Algorithms. Cambridge University Press, 2014.

**Questions:**

See weakness part.

---

> ### Author Response · Authors · 2024-11-19
> **Rebuttal by Authors**
>
> Thank you for acknowledging the novelty of our paper as well as providing the valuable feedback. Below we address the detailed comments, and hope that you can find our response satisfactory.
>
> We divide these questions into two sub-questions.
>
> ***Question 1: This maybe not a significant contribution since it seems really natural. Intuitively, if the data distribution can be separated by a network, there must exist functions in the class of larger networks that can also separate the data distribution.***
>
> Answer: It is well known that the more data there is, the better the generalization, which is obvious. However, the specific amount of data required to achieve generalization is indeed of great research significance.
>
> Similar, it is reasonable that distribution which can be expressed by small networks can be expressed by large networks naturally. However, the corresponding generalization bound and sample complexity need to be calculated to help us better understand generalization.
>
>
>
>
> ***Question 2:  And techinically, there are Rademacher complexity type bounds with sample complexity [1]. The only difficult of the main theorem is to estimate the Rademacher complexity of the class of larger ReLU networks.***
>
> Answer:
>
> (1) The Rademacher complexity type bounds with sample complexity [1] are uniform bounds, that is, bounds are correct for all networks in the hypothesis space. But, our bounds are not uniform bounds and they are valid for those networks that minimize the empirical risks. The main difficulty of Theorem 4.3 is to use this condition to give a better generalization bound.
>
> (2) We give a new and better generalization bound such that BOTH the network size and number of training data are completely in the denominator, as shown in Theorem 4.3. The sample complexity derived based on this generalization bound only depends on the distribution itself, which has not been achieved before, as shown in Corollary 4.4.
>
> For example, the generalization bound based on VC-dimension in Theorem 4.8 contains the term $\frac{VC(H)}{N}$. This implies that over-parameterized models do not generalize, contradicting the experimental results that over-parameterized models generalize well [2]. However, our generalization bounds can be used to explain this important phenomenon.
>
> (3) Extending our conclusions to deep networks is actually a challenge, the real difficulty is how to ensure that sample complexity is independent of network size. At present, we do not have a way to achieve that. Moreover, because most theoretical analyses are difficult on the deep network, (without special assumptions), not just in terms of Radermacher Complexity, but also Gradient Descent Analysis, Robustness Analysis, and so on, so it is not practical to directly use these methods, and new methods need to be developed to extend our result to deep network.
>
> [2] M. Belkina,D. Hsuc, S. Maa, S. Mandala, Reconciling modern machine-learning practice and the classical bias–variance trade-off, 2019 (Fig. 1).

---

> ### Comment · Reviewer_WTwE · 2024-11-26
>
> We thank the authors for the detailed responses. I will remain the score.

---

> > ### Author Response · Authors · 2024-11-27
> >
> > Thank you very much for your insightful comments which have helped us to improve the paper.

---

### Official Review · Reviewer_LSKf · 2024-11-04

**Soundness:** 3
**Presentation:** 1
**Contribution:** 3
**Rating:** 6
**Confidence:** 3

**Summary:**

This paper studies generalization error of 2 layer neural networks that minimize empirical risk in a binary classification problem. The authors present a lower bound for accuracy based on the expressiveness of these networks, indicating that, with a sufficiently large training sample and network size, these networks can generalize. They offer an extension of the result to approximate empirical risk minimizers. They consider several other implications (relationship between the size of the neural network needed to represent the target distribution, and the quantity of data required to ensure generalization, robustness, etc.).

**Strengths:**

One strength is that the work is studying an important problem: explaining deep learning generalization.

Another strength is that they are using unconventional hypotheses, namely relying on the size of a network that exactly matches the data. This hypothesis was considered by Buzaglo et al in recent work (ICML 2024).

**Weaknesses:**

There are many weaknesses.

Perhaps the biggest issue is that there's no new insight here. We have old school uniform convergence analyses, coming together with universal approximation arguments, but what have we learned? Negative results are for "some distribution" and don't explain practice. And there's no evidence the accuracy lower bounds (use error not accuracy) are strong enough to explain practice.

The hypothesis that some network exactly labels the data with a margin c is too strong in practice. This hypothesis rules out situations where there is label noise. It even rules out situations where there is no noise, but the decision boundary cannot be exactly represented by a neural network. (Approximation theorems don't help here.)

The results are uniform convergence results and so it's not clear how they get around the roadblock identified by Nagarajan and Kolter (2022)'s work on uniform convergence not explaining deep learning.

**Questions:**

How do the results relate to Buzaglo et al (ICML 2024)?

How do the results relate to Nagarajan and Kolter (NeurIPS best paper: arXiv:1902.04742)? How do you sidestep the issues they raise?

What would empirical validation of these theories look like?



## FOLLOW UP QUESTIONS - PLEASE RESPOND IF YOU CAN ##

I have a question that would help me move to quickly resolve my concerns. The questions/remarks below ("Other questions / comments.") are less important and you should simply aim to address these in your own revisions. They are likely small typos or minor points that would confuse readers.

Key questions:

1.
In the proof of Lemma B.5, in (3) you write "The L1,Inf norm of the three transition matrices...". It would seem that there is assumption hidden here about the L1,Inf norms of the networks in H_W(n). I don't see any assumptions about the L1,Inf norms in definition of H_W(n) at the top of page 4. Can you maybe offer a bit more detail on the arguments arriving at these three norm bounds?



Other questions / comments.

1. It seems that the (W0,c) delivered by Proposition 4.2 are rather important in practice. These terms appear in the final bound as (W0 + c)/ cN and so, in particular, the tradeoff between W0 and c is essential. It may be the case that the minimum W0 is W0** but for that width W0**, the corresponding c** might be 2^{-100}, and maybe each increase in W0 only brings you a small improvement in c. Of course, these are just constant, but they would make the bounds impractical (and thus not explain practice).

2. Defn 3.1. There is no standard notion of inf over a pair of random variables. You should make a probability one statement over the two samples: y_1 != y_2 ==> ||x_1 - x_2||_2 > 0. In particular, this implies no noise and a zero Bayes error rate. These are strong assumptions that should be highlighted with a remark. There is no role for the L2 norm here. The assumption is simply that H(y|x) = 0. for (x,y) ~ D, IINM.

3. Proposition 3.2 is written in an odd way. M_W \subset H_W and so you are simply arguing that M_W is non-empty, using uniform continuity (compactness + continuity).

4. Proof of Proposition 4.2. There is a claim that I do not believe is true, starting "Then, because D has a positive separation distance, [there exists a continuous function that f(x)=y with probability one under D]" You would likely need a uniform gap ||x_1 - x_2||_2 >= gap  for some constant gap > 0. Regardless, it seems the only use of this assumption is to guarantee this continuous function f, and so just make that your assumption in the first place, which is then the weakest assumption that makes your argument go through, and is also the clearest explanation of your assumption.

5. Theorem A.1. Missing quantification over x.

6. Lemma B.4. b_i is a vector and so it doesn't have an L1,Inf norm. Do you mean L1. Wen et al. talk about the L1,inf of the combined bias and weights, so I believe you want L1?  And what is the justification for the claim that the L1,Inf norms at layer i are bounded by c_i? Or is this mean to be a definition? (If so, remove "Then" and write "We also assume...".

Wen et al. (Statistica Sinica 31 (2021), 1397-1414 doi:10.5705/ss.202018.0468) On CIFAR, c >= 15 in their experiments to


Notation:

7. Using W for the width and W_i for weight matrices is rather nonstandard. You elsewhere use w_i for whole matrices. Would be nice to have the notation consistent throughout the work.



## UPDATE TO REVIEW

Thank you to the authors for answering my last minute questions.


I'm finding it very hard to get comfortable with these results, but after considerable effort studying some key proofs in detail, I cannot find any errors, and so I will upgrade my score. I will lower my confidence, however, to reflect the fact that my intuition is feeling off.

Theorem 4.3 is, in many ways, exactly what we would want to prove. But I'm finding it very difficult to believe that the ingredients assembled here are what has achieved it. I suspect that one of the assumptions is doing a lot more heavy lifting than is evident. Even so, this would be progress.

In terms of key assumptions, the existence of a finite width (W0) network that has confidence c with probability 1 is essential to the current proof. The proof also relies on a Rademacher bound for L1,Inf bounded networks. I have had to assume that this bound is correct. It is published in a reputable journal (Statistical Sinica), and so I think this is a fair assumption.

I'm skeptical of ERM on the cross entropy as a model of standard deep learning algorithms, and so this is another source of my unease. The local minimum result alleviates this concern somewhat, but I've no idea how big the multiplicative factor (q) would be in practice and so I don't know if this is realistic for standard amounts of overparameterization. (And overparameterization likely affects q, so it is not solved by making W bigger.)

I'm somewhat skeptical that H_W(n) has weights in [-1,1]. Standard overparameterized networks will have most weights MUCH smaller than this at initialization. It seems like a large space for ERM to operate over.

Typos/comments:

The paper (especially the appendix) is full of English grammar errors.

I'm still confused by the statement of Lemma B.4. I think the last sentence of the 1st paragraph ("Then the L1,Inf norm of wi plus the L1,inf norm of bi is not more than ci.") should start "Assume the ..." because there is no way to deduce these bounds from anything stated earlier. Indeed, the word "then" and "there" are misused throughout the paper and generally there are many language issues, but they are relatively easy to ignore.

---

> ### Author Response · Authors · 2024-11-19
> **Rebuttal by Author**
>
> Thank you for acknowledging the importance of the problem studied in our paper as well as providing the valuable feedback. Below we address the detailed comments, and hope that you can find our response satisfactory.
>
> ***Question 1: Perhaps the biggest issue is that there is no new insight here. The results are uniform convergence results and so it's not clear how they get around the roadblock identified by Nagarajan and Kolter (2022)'s work on uniform convergence not explaining deep learning. How do the results relate to Nagarajan and Kolter? How do you sidestep the issues they raise?***
>
> Answer:  We believe that **our results provide new insights** compared to existing results in that we provide a better generalization bound that can be used to explain the over-parameterized models generalize well to some extent.
>
> (1): In our generalization bound, BOTH the network size and number of training data are completely in the denominator, as shown in Theorem 4.3. And the sample complexity for generalization, the network size required for generalization got by our generalization only depended on the distribution itself(Corollary 4.4). **Previous conclusions have been unable to achieve this point**, such as the results mentioned in Nagarajan and Kolter (2022). This implies that when the size of the network and the number of training samples are both large, we have shown the generalizable of network, consistent with the theoretical mystery that over-parameterized models generalize well [1].
>
> Our generalization bound is derived under the assumption that the $L_\infty$ norm of parameters $\le1$, which is reasonable in the sense explained below. By increasing the size of the network, it is easy to control the $L_\infty$ norm of the parameters after training not more than 1 and ensure the accuracy of the training set. For example, use CIFAR10 to train ResNet18 with weight decay 0.0005, after training, the $L_\infty$ norm of the parameters is smaller than $0.6$, and the network has accuracy 99$\%$ on the training set.
>
> (2): **The core contradiction pointed out in Nagarajan and Kolter (2022)'s paper** is that: under big dataset, to get a better performance on a training set, training need to increase the norm of network weights, which in turn leads to an increase of the Radermacher Complexity, and thus will increase the generalization bound.
>
> In fact, our bound is not UNIFORM in that it holds for neural networks that minimize empirical risk, and **such problem does not bother us**.
> Since the network size $W$ and number of training data $N$ are in the denominator of the generalization bound, so training the network will not lead to an increase in our generalization bound, so **our results can truly explain practice**.
>
> Specifically, we use Theorem 4.3 to explain **our results how to avoid Nagarajan and Kolter (2022)'s paper’s problem**: We can use training to find a network with minimal empirical error, and we set the size of the network to be very large and limit the absolute values of the parameters during training such as weight decay. After training, we can obtain a large network with the $L_\infty$ norm of its parameters no more than $1$. Consider our generalization bound: the network size is completely on the denominator, so no matter how large the network, even if it far more the number of data in training set, it can only provide positive effects on the generalization. Therefore, our conclusion can fully demonstrate that large-scale but bounded-value networks must have good generalization performance.
>
> [1] Belkina.et, Reconciling modern machine-learning practice and the classical bias–variance trade-off, 2019 (Fig.1).
>
> ***Question 2: How do the results relate to Buzaglo et al (ICML 2024)?***
>
> Answer: In our opinion, our work has some similarities and many major differences with Buzaglo (ICML 2024).
>
> **Similarity:** We both assume that distribution can be expressed by a network; we both try to find the sample complexity based on the expressive of distribution.
>
> **Differences:**
>
> (1): Our samples complexity only depends on data distribution and does not depend on the hypothesis space of networks (Corollary 4.4). This is the main contribution of our paper. But the sample complexity obtained by Buzaglo still depends on hypothesis space of network.
>
> (2): Our paper focuses on the networks that make the empirical risk minimum (use cross-entropy loss), which does not imply INTERPOLATION. Buzaglo et al focus on INTERPOLATION network obtained by an algorithm designed by themselves, and such networks make the accuracy on training set to be 1 and they do not consider the empirical risk defined by cross-entropy loss.
> Therefore, our generalization bound is superior in terms of its applicability.
>
> (3): We give the samples complexity to make $A_D(F)\ge1-\epsilon$ for all ERM networks, but Buzaglo et al just find the samples complexity to make their algorithm give a high accuracy network with high probability.

---

> ### Author Response · Authors · 2024-11-19
> **Rebuttal by Author**
>
> ***Question 3: What would empirical validation of these theories look like?***
>
> Answer: We can consider the following experiments: We training the network and control the absolute values of the parameters not more than 1 during training. Then we observe the relationship curve between the generalization and the network size, number of data. Contrast the relationship between this curve and Theorem 4.3.
>
> ***Question 4: The hypothesis that some network exactly labels the data with a margin c is too strong in practice. This hypothesis rules out situations where there is label noise. It even rules out situations where there is no noise, but the decision boundary cannot be exactly represented by a neural network. (Approximation theorems do not help here.)***
>
> Answer: Firstly, we point out that in Proposition 4.2, we have already shown that all distributions with positive separation distances satisfy this assumption, so this assumption is not that strong as it appears.
>
> Secondly, the confidence $c$ in Definition 4.1 must be incurred when considering minimizing cross-entropy empirical risks, because among all distributions that can be expressed by networks of the same size, some are easily learned by networks, while others are not. If we do not introduce $c$, it will lead to the same generalization bound for the distribution of those that are easy to learn and those that are difficult to learn, which is obviously unreasonable.
>
> We give a simple example. Let $D$ be defined in $B_2(x_1,1)\cup B_2(x_2,1)$. The points in $B_2(x_1,1)$ have label 1. The points in $B_2(x_2,1)$ have label -1. When $||x_1-x_2||_2>2$, it is linearly separable.
>
> For situations $||x_1-x_2||_2>>2$ and $||x_1-x_2||_2=2.01$, it is obvious that the first situation is easier to learn. Because for the first situation, minimizing the empirical error for any two points with different labels must lead to the linear function being able to separate these two spheres, because the dividing line created by the linear function is inevitably far away from these points and it will not divide a ball into two.
>
> But when $||x_1-x_2||_2=2.01$, consider that these two balls are too close, so if the points are taken unevenly, it may lead to inaccurate delineation of the boundary between these two balls during minimize the empirical error and lead to bad generalization.

---

> > ### Author Response · Authors · 2024-11-25
> >
> > As the discussion phase is concluding soon, we would greatly appreciate your feedback on whether our rebuttal has adequately addressed your concerns. Please feel free to bring up additional discussion if needed.

---

> ### Author Response · Authors · 2024-12-03
>
> We kindly invite you to review our rebuttal as the discussion period comes to an end. We also welcome your opinions on this paper or any  questions we have not yet resolved.

---

> > ### Comment · Reviewer_LSKf · 2024-12-03
> > **Response forthcoming**
> >
> > Apologies for the delayed response.
> >
> > I am still processing your comments, and will respond in more detail tomorrow. It sounds like you did some sort of localization argument which I did not see the first time around, which would be a uniform convergence argument in some small “ball”. (Where’s this localization argument?) The separation assumption is strong because even positive separation is strong in my opinion.

---

> > > ### Author Response · Authors · 2024-12-03
> > >
> > > Thank you for your reply. We look forward to your further response, but there is one thing I need to remind that the discussion period will end at December 2nd AOE. Please pay attention to the time.
> > > In addition, for the separability in definition 4.1, we need to point out the following fact that:
> > >
> > > If we simplify the assumption of separability to $P(yF(x)>c)\ge1-\epsilon$, then
> > >
> > > 1. We can still derive Theorem 4.3, but the resulting generalization bound will weaken: when the number of data and network width are large enough, the accuracy is not close to 1 but rather 1-epsilon.
> > >
> > > 2. Under such premise, it is impossible to obtain a generalization bound close to 1 when the data and network are large enough.
> > >
> > > If we simplify the assumption of separability to $P(yF(x)>0)\ge1-\epsilon$, then
> > >
> > > 1. We can not derive Theorem 4.3. In other words, under this premise, we cannot make the number data and network size completely on the denominator in the generalization bound.

---

> ### Author Response · Authors · 2024-12-04
> **About The Follow Up Questions**
>
> Key Question 1:
>
> Please note that in lines 168-170 (revision), we have declared that the $L_\infty$ norm of network in H_W(n) is not more than 1. Based on this, we show how we get such three $L_{1,\infty}$ norm:
>
> The first transition matrices: this layer is the same as the first layer of $f$. Consider the $L_{\infty}$ bound of values of parameters of $f$ is not more than 1, and the first transition matrix of $f$ has $n$ weights in each row, and with the  bias added, there are a total of n+1 weights, so its $L_{1,\infty}$ norm is $n+1$.
>
>  The second transition matrices: Let the second transition matrices of $f$ be $W_f$, bias be $c$. Then the second transition matrices of $F$ is $W_f/k$, bias is $c/k+a/k$ or $c/k-a/k$. Using the bound of value of parameters of $f$ and the value of $k$, and $W_f$ has width $W$, so we get the result.
>
>  The third transition matrix: It is $(1,1,\dots,1,-1,-1,\dots,-1)$, where there are $k$ number of 1 and $k$ number of -1 in it, and we get the result.
>
> Others Question: About the proof of Proposition 4.2.
>
> Please note the Definition 3.1, we want distribution $D$ to satisfies: $inf _{(x_1,y_1),(x_2,y_2),y_1\ne y_2}||x_1-x_2||>0$.  This  actually implies that the distance between different label samples in distribution $D$ cannot be arbitrarily close to 0 (which means the gap you mentioned does exist), or there will be $inf _{(x_1,y_1),(x_2,y_2),y_1\ne y_2}||x_1-x_2||=0$. We will clarify this in the next version.
>
> About Lemma B4.  $b_i$ is a vector and so it doesn't have an $L_{1,\infty}$ norm.
>
> In the whole proof, we see vector as a matrix with one column. More specific, we write one layer of a neural network is $Wx+b$, see $W,x,b$ as matrix, from the perspective of matrix multiplication, there are $W\in[-1,1]^{m,n}$, $x\in[0,1]^{n,1}$ and $b\in[0,1]^{m,1}$. So $L_{1,\infty}$ norm of $b$ is the maximum weights of $|b|$.  We will clarify this in the next version.
>
> Thank you for raising these questions. We will make modifications in future versions.

---

### Meta-Review · Area_Chair_Pyzg · 2024-12-18

**Metareview:**

This paper studies the generalization properties of two-layer neural networks that minimize empirical risk in a binary classification problem. The authors present a lower bound for accuracy based on the expressiveness of these networks in Proposition 4.2 and Thoerem 4.3, indicating that, a distribution (or data) can be well seperated with some probability by a certain two-layer neural network and estimate the performance of the network on the dataset. It offers a new view of generalization and considers several other implications, e.g., robustness. The AC suggests the author to include the comparison with results on PAC-Bayes.

**Additional Comments On Reviewer Discussion:**

After the discussion, most of the issues have been addressed.

---

### Decision · Program_Chairs · 2025-01-22

Accept (Poster)